# Deleting the mitochondrial respiration negative regulator MCJ enhances the efficacy of CD8+ T cell adoptive therapies in pre-clinical studies

Meng-Han Wu[1], Felipe Valenca-Pereira[1], Francesca Cendali [2], Emily L. Giddings[3], Catherine Pham-Danis[4], Michael C. Yarnell [4], Amanda J. Novak [4], Tonya M. Brunetti[1], Scott B. Thompson[1], Jorge Henao-Mejia [5,6,7], Richard A. Flavell [8,9], Angelo D'Alessandro [2], M. Eric Kohler [4,10] ✉ & Mercedes Rincon [1,3] ✉

Mitochondrial respiration is essential for the survival and function of T cells used in adoptive cellular therapies. However, strategies that specifically enhance mitochondrial respiration to promote T cell function remain limited. Here, we investigate methylation-controlled J protein (MCJ), an endogenous negative regulator of mitochondrial complex I expressed in CD8 cells, as a target for improving the efficacy of adoptive T cell therapies. We demonstrate that MCJ inhibits mitochondrial respiration in murine CD8+ CAR-T cells and that deletion of MCJ increases their in vitro and in vivo efficacy against murine B cell leukaemia. Similarly, MCJ deletion in ovalbumin (OVA)-specific CD8+ T cells also increases their efficacy against established OVA-expressing melanoma tumors in vivo. Furthermore, we show for the first time that MCJ is expressed in human CD8 cells and that the level of MCJ expression correlates with the functional activity of CD8+ CAR-T cells. Silencing MCJ expression in human CD8 CAR-T cells increases their mitochondrial metabolism and enhances their anti-tumor activity. Thus, targeting MCJ may represent a potential therapeutic strategy to increase mitochondrial metabolism and improve the efficacy of adoptive T cell therapies.

Adoptive T-cell therapy is an emerging immunotherapy for treating hematologic and solid malignancies. The three main modalities of adoptive T-cell therapies are undergoing clinical testing and/or being used as proven therapies in patients with cancer. Tumor-infiltrating lymphocyte (TIL) therapy use T cells isolated from solid tumors. Genetically engineered T-cell receptor (TCR) therapy uses autologous peripheral T cells expressing a specific TCR recognizing a tumor antigen presented by MHC/HLA. Chimeric antigen receptor (CAR) T cell therapy uses autologous peripheral T cells redirected by the CAR molecule to recognize tumor antigens independent of MHC molecules[1-4]. Among the three, currently only CAR-T cell therapy has been approved by FDA for hematological cancers[4-7]. TIL therapy is in a number of clinical trials for different types of solid tumors[8-10], and has recently been approved by FDA to treat patients with unresectable or metastatic melanoma. Similarly, several clinical trials are currently testing engineered TCR therapy, with promising results[11-14]. While these therapies differ in their approach to generating tumor-responsive T cells, each has historically relied upon extensive ex vivo

expansion of T cells, most commonly with IL-2. Concerns surrounding the impact of such manufacturing practices on the ex vivo differentiation of T cells has led to exploration of alternative cytokines and truncated expansion during manufacturing[15]. However, the optimal conditions to generate adoptive T-cell products with high levels of efficacy, particularly against solid tumors, remains an area of investigation.

As of December of 2023, over 27,000 doses of FDA-approved CAR-T cells have been administered to patients products[16]. In CAR-T cell therapy, T cells (including both CD4 and CD8 T cells) from cancer patients are transduced ex vivo with retroviral/lentiviral vectors expressing an antibody-based receptor specific for an antigen expressed on the cancer cells. Through the expression of the CAR, T cells are "redirected" to the target antigen. After multiple in vitro cell expansions (in the presence of exogenous cytokine), transduced cells are reinfused into the patients for tumor destruction[17]. There are four FDA-approved CAR-T cell therapies targeting CD19 for relapsed and/or refractory (r/r) B-lineage leukemias and lymphomas. CAR-T cell therapy demonstrated the ability to induce complete remissions in up to 60–90% of patients with r/r-B-ALL (B cell acute lymphocytic leukemia), 50–65% with r/r-NHL (Non-Hodgkin Lymphoma) and 20–50% with CLL (Chronic lymphocytic Leukemia)[18–22]. However, treatment failures and post-CAR-T cell therapy relapses have been found to be common and remain a major concern of the therapy[18,23]. The underlying mechanisms of poor response of CAR-T cell therapy include poor initial expansions, sub-optimal persistence of functional CAR-T cells, and loss or down-regulation of the targeted antigen by tumor cells. In addition, CAR-T cell-mediated cytokine release syndrome (CRS) and immune effector cell-associated neurotoxicity syndrome (ICANS) can lead to severe life-threatening conditions in patients[24]. CRS, marked by fevers, malaise, and—in severe cases—hemodynamic instability, is driven by high systemic levels of inflammatory cytokines such as IL-1 and IL-6. While CRS usually precedes ICANS, which manifests as neurological symptoms ranging from word-finding difficulties to seizures, the pathophysiology has been more elusive with elevated IL-1, disruptions to the blood brain barrier and endothelial dysfunction likely contributing to this toxicity components[24,25]. While monocytes and macrophages have been shown to produce high levels of IL-6 during CRS[26,27], CD4 CAR-T cells have been implicated as drivers of CRS through the activation of monocytes and the direct production of IL-6[26–30]; however, CD8 CAR-T cells are more effective when supported by CD4 cell help[29,31], and CAR-T cell product with defined CD4:CD8 ratio has been manufactured and showed reduced toxicity and improved disease-free survival[32].

Metabolism is reemerging as a major factor that regulates the function of immune cells and influences the course of an immune response[33–35]. Both CD4 and CD8 cells undergo a reprogramming of their metabolic pathways throughout differentiation[33–35]. Naïve CD8 cells primarily use mitochondrial oxidative phosphorylation (OXPHOS)[36,37]. Activation of naïve CD8 cells causes a metabolic switch towards glycolysis for ATP synthesis and biosynthesis of nucleotides. Glycolysis is the main pathway used for T cell proliferation. However, the production of cytokines and cytotoxicity by effector CD8 cells does not depend on glycolysis, but relies on mitochondrial respiration[38,39]. Memory CD8 cells use free fatty acid (FFA) oxidation in mitochondria as the main energy pathway for long-term persistence[36,40,41]. As immunometabolism plays an essential role in T cell fate, manipulating T cell metabolism is emerging as an additional approach to improve CAR-T cell therapy and other adoptive T-cell therapies[42–44]. While glycolysis is known to be essential for cell expansion of T cells, it has also been shown to promote differentiation and reduce the self-renewal capacity of CAR-T cells primarily during the manufacturing of CAR-T cells[42,43]. Inhibition of glycolysis during manufacturing seems to maintain CAR-T cells in a more undifferentiated state and can improve CAR-T cell activity in vivo[45–47]. Mitochondrial respiration is essential for survival and function of CAR-T

cells[44], and increasing mitochondrial respiration while sustaining glycolytic rates that allow for proliferation would be predicted to improve adoptive T-cell therapy efficacy. Increasing mitochondrial biogenesis and mitochondrial mass is currently considered a potential strategy through overexpression of PGC1α[44,48–50]. However, mitochondrial biogenesis, much like other organelles biogenesis, is an energetically costly process due to the need of lipid synthesis for the inner and outer membranes. A number of pharmacological inhibitors have been developed to inhibit mitochondrial respiration, however, strategies to enhance specifically mitochondrial respiration without the need to increasing mitochondrial biogenesis are yet to be discovered.

We have shown that MCJ (Methylation-Controlled J protein, encoded by the nuclear *DnaJC15* gene) is a transmembrane protein localized in the inner membrane of mitochondria, and interacts with Complex I through the N-terminal domain where it acts as an endogenous negative regulator of Complex I of the electron transport chain (ETC)[51–55]. Loss of MCJ results in increased Complex I activity and mitochondrial membrane potential, without increasing the production of reactive oxygen species (ROS)[51–55]. Loss of MCJ also promotes the formation of respiratory supercomplexes and reduces electron leak, although the mechanism underlying this is not yet defined[51,52,55]. In mice, MCJ is abundantly expressed in CD8 cells relative to CD4 cells and other immune cells[52]. According to the function of MCJ as a negative regulator of mitochondria, MCJ-deficient mouse CD8 cells have increased mitochondrial membrane potential, Complex I activity, mitochondrial respiration and production of mitochondrial ATP, without impairing glycolysis[51]. Loss of MCJ does not affect CD8 cell proliferation, but the increased mitochondrial ATP production in the absence of MCJ enhances secretion of cytokines as well as cytotoxic activity of effector CD8 cells[51]. Moreover, while WT memory CD8 cells from influenza virus-infected mice fail to provide protection in the absence of memory CD4 cells, MCJ-deficient memory CD8 cells protect against a lethal dose of influenza virus without the help of CD4 cells[51]. Thus, boosting mitochondrial respiration in CD8 cells by the loss of MCJ enhances the effector function and memory activity of CD8 cells, while reducing the need of CD4 cell help.

No studies have investigated the effect of loss of MCJ in CD8-mediated anti-tumor immune responses. Here, we show that loss of MCJ in mouse CD8 cells enhances TCR-antigen specific anti-tumor immune responses in vitro and in vivo. Loss of MCJ in mouse CD8 CAR-T cells also enhances their efficacy against B cell leukemia in vitro and in vivo. Moreover, we describe for the first time the presence of MCJ in primary human CD8 cells, and its role on mitochondrial metabolism and cytokine secretion by these cells. We have developed a novel CD19-CAR construct designed to silence MCJ in human CD8 CAR-T cells, and have shown that this approach also increases the anti-tumor killing activity of human CD8 CAR-T cells.

## Results

### Loss of MCJ increases TCR-antigen specific anti-tumor response of CD8 cells in vitro

MCJ (encoded by the *DnaJC15* gene) is a negative regulator of Complex I and acts as an endogenous break for mitochondrial respiration[51,52,54]. Within the immune system, MCJ is more abundantly expressed in CD8 cells and MCJ KO mice display enhanced CD8 cell response against influenza virus infection[51,52]. No studies have previously investigated the role of MCJ in the anti-tumor immune response. Thus, we first examined the ability of antigen specific tumor cells to grow in MCJ KO mice. We implanted B16 melanoma cells expressing ovalbumin (OVA) as a foreign antigen in WT or MCJ KO mice as hosts. B16-OVA melanoma cells are poorly immunogenic and they rapidly developed large aggressive tumors in vivo[56]. Accordingly, rapidly growing tumors developed in WT hosts implanted with B16-OVA cells (Fig. 1a). In contrast, B16-OVA tumors grew slower in MCJ KO mice and did not reach the larger size of the tumors in WT mice (Fig. 1a). Survival

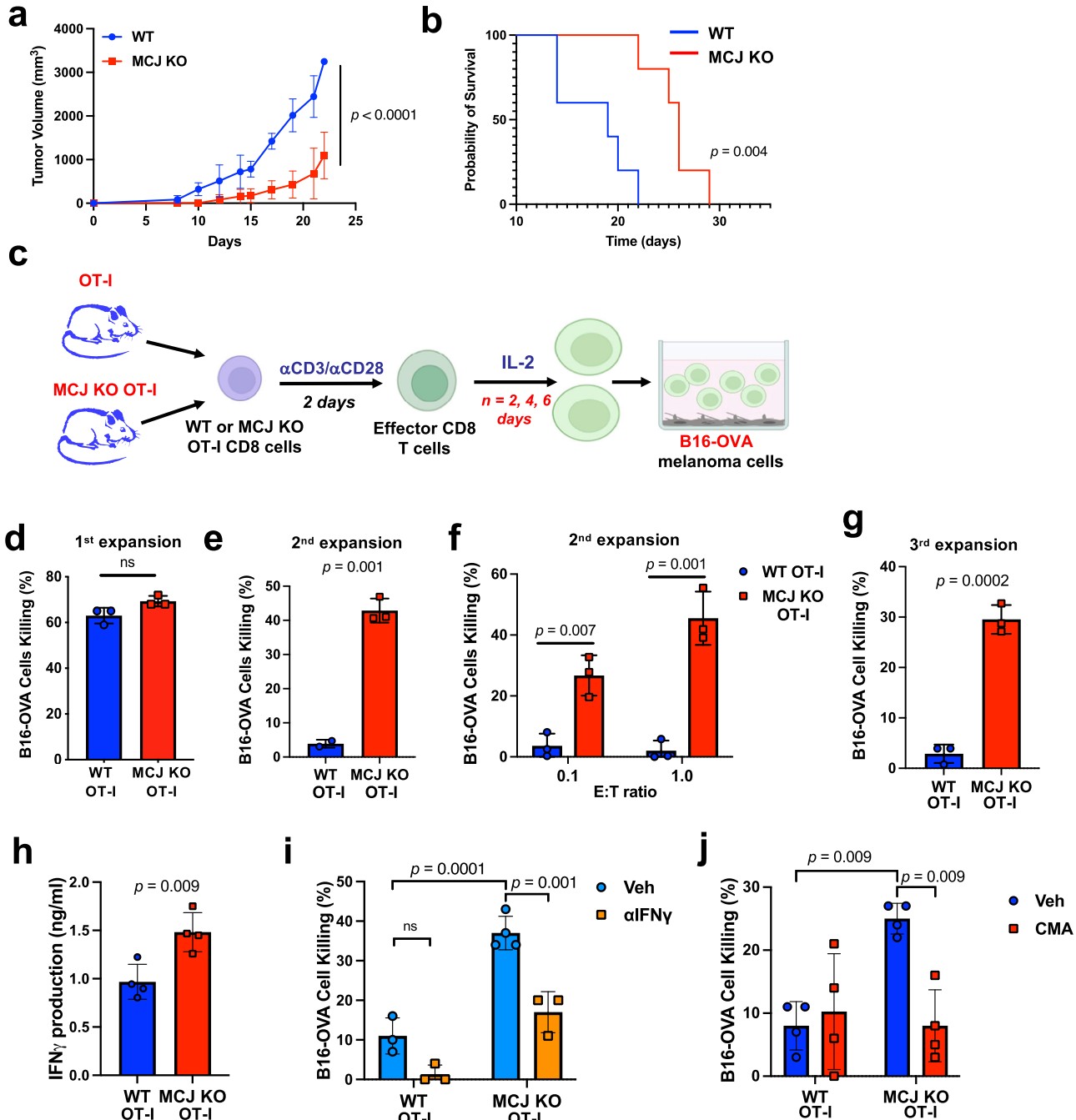

**Fig. 1 | Loss of MCJ in TCR-antigen specific CD8 cells enhances their anti-tumor killing activity in vitro. a**, **b** B16-OVA melanoma cells ($4 \times 10^5$/mouse) were subcutaneously (s.c.) injected on the flank of WT or MCJ KO host mice. Tumor volume over time (**a**) and survival (**b**) were followed ($n = 5$ biologically independent animals). **c**–**f** WT or MCJ KO OT-I CD8 cells were activated for 2 days with anti-CD3/anti-CD28 Abs, washed, and then expanded with IL-2 (40 IU/ml) for 2 days (1st expansion) and expanded again with fresh medium and IL-2 (2nd expansion) (**c**) (created using BioRender.com). After the 1st expansion with IL-2, WT and MCJ KO OT-I cells were co-cultured with B16-OVA cells for killing assay at an E:T = 10, and live B16-OVA cells were counted after 24 h ($n = 3$ biologically independent cells) (**d**). After the 2nd expansion with IL-2, WT and MCJ KO OT-I cells were co-cultured with B16-OVA cells for killing assay at an E:T = 10 and live B16-OVA cells were counted after 24 h ($n = 3$ biologically independent cells) (**e**), or an E:T = 1 and E:T = 0.1 and live B16-OVA cells were counted after 46 h co-culture ($n = 3$ biologically

independent cells) (**f**). **g** WT and MCJ KO OT-I cells were activated as in (**c**) and after the 3rd expansion with IL-2 were co-cultured with B16-OVA cells at an E:T = 5. Live B16-OVA cells were counted after 24 h co-culture ($n = 3$ biologically independent cells). **h** ELISA of IFNγ levels in supernatant of co-cultures of B16-OVA cells with either WT or MCJ KO CD8 OT-I cells after the 2nd expansion ($n = 4$ biologically independent samples). **i** WT and MCJ KO OT-I cells were activated, expanded with IL-2 for two expansions and co-cultured with B16-OVA cells at an E:T = 10, with or without an anti-IFNγ blocking Ab. Live B16-OVA cells were counted after 24 h ($n = 3$ biologically independent cells). **j** WT and MCJ KO OT-I cells were activated and expanded as in (**i**), pretreated with CMA or vehicle for 2 h, washed, and co-cultured with B16-OVA cells (E:T = 2). Live B16-OVA cells were counted after 24 h ($n = 4$ biologically independent cells). $p$ was determined by mixed-effect analysis (**a**), Mantel-Cox test (**b**), two-sided unpaired $t$ test (**d**–**h**), and 2-way ANOVA multiple comparisons (**i**, **j**). Mean ± SD is shown for (**a**, **d**–**j**).

analyses also showed significantly prolonged survival of MCJ KO mice (Fig. 1b), indicating superior protection of MCJ KO mice against melanoma.

To investigate whether the superior protection of MCJ KO hosts against the B16-OVA melanoma could be due to the absence of MCJ in T cells, we used T cell conditional MCJ KO mice (hereinafter referred to as T-cMCJ KO mice) generated by crossing MCJ[fl/fl] mice with CD4-Cre mice. Analysis of MCJ in freshly isolated CD8 cells from WT mice (CD4-Cre[−] mice) and T-cMCJ KO mice (CD4-Cre[+] mice) by Western blot analysis showed reduced levels of MCJ in T-cMCJ KO CD8 cells (Supplementary Fig. 1a). As described for MCJ KO mice[51], reduced levels of MCJ in T-cMCJ KO mice did not affect T cell development in the thymus (Supplementary Fig. 1b, c). Similarly, no difference in the presence of CD4 and CD8 cells in the lymph nodes (Supplementary Fig. 1d) or the spleen (Supplementary Fig. 1e) could be observed between WT and T-cMCJ KO mice. We have previously shown that proliferation and cell surface activation markers upon activation were not affected, but IFNγ secretion was increased in CD8 cells from MCJ KO mice, due to a higher mitochondrial ATP production[51]. Analyses of CD44, CD25, CD69 and CD62L upon activation with anti-CD3/anti-CD28 antibodies (Abs) showed no difference between WT and T-cMCJ KO CD8 cells (Supplementary Fig. 1f). Similarly, no difference in the expression of activation markers was observed between activated WT and T-cMCJ KO CD4 cells (Supplementary Fig. 1g). However, similar to MCJ KO CD8 cells, T-cMCJ KO CD8 cells secreted more IFNγ after activation compared to WT CD8 cells (Supplementary Fig. 1h). In contrast, IFNγ production by activated T-cMCJ KO CD4 cells was not increased relative to WT CD4 cells (Supplementary Fig. 1i), consistent with the low baseline expression of MCJ in CD4 cells, minimizing the impact of MCJ loss on the mitochondrial membrane potential of CD4 cells[51,52]. We then examined the overall survival of these mice after implantation of B16-OVA melanoma cells. Similar to MCJ KO mice, T-cMCJ KO mice showed a prolonged survival compared to WT mice (Supplementary Fig. 1j, k), indicating the loss of MCJ in the T cells enhanced T cell-mediated anti-tumor immune response against poorly immunogenic B16-OVA melanoma tumors.

To directly examine the anti-tumor killing activity of antigen-specific CD8 cells lacking MCJ, we used OT-I transgenic mice expressing the TCR that recognizes explicitly the SIINFEKL peptide presented by MHC-I[57]. CD8 cells were activated with anti-CD3/anti-CD28 Abs for 2 days, and expanded with IL-2 for specific periods of time (Fig. 1c), as it is clinically done for the use of adoptive T cell therapy prior to the cell infusion to patient. After expansion with IL-2, we tested the tumor killing activity of the expanded OT-I CD8 cells against B16-OVA cells in vitro. After the 1st expansion (2 days with IL-2), both WT and MCJ KO OT-I CD8 cells were highly efficient in killing B16-OVA cells (Fig. 1d). Since the continuous expansion of effector T cells with IL-2 ex vivo has been reported to impair their function after activation[58], we also examined CD8 cells expanded with IL-2 for 2 days, split and expanded with IL-2 for two more days (2nd expansion) for their killing activity against B16-OVA cells. MCJ expression was maintained in CD8 cells through multiple expansions with IL-2 (Supplementary Fig. 2), as previously described[59]. While WT OT-I CD8 cells lost the killing activity compared with WT CD8 cells from the 1st expansion, MCJ KO OT-I CD8 cells retained their killing activity after the 2nd expansion (Fig. 1e). Even at a low effector to target (E:T) ratio, MCJ KO OT-I CD8 cells showed superior in vitro anti-tumor killing activity (Fig. 1f). We also examined the anti-tumor killing activity of CD8 cells after three expansions with IL-2, and MCJ KO OT-I CD8 cells also showed enhanced killing activity compared to the WT OT-I CD8 cells (Fig. 1g). Thus, expansion of effector CD8 cells with IL-2 causes a progressive loss of anti-tumor killing activity, but the lack of MCJ sustains their killing activity.

We then investigated the mechanism for the superior killing activity of effector MCJ KO CD8 cells after expansion with IL-2. Analysis of IFNγ levels in the supernatant of IL-2-expanded effector OT-I CD8 cells (two expansions) co-cultured with B16-OVA melanoma cells showed higher levels of IFNγ by MCJ OK OT-I CD8 cells than WT OT-I cells (Fig. 1h). To determine whether this augmented IFNγ production could contribute to the superior killing activity of MCJ KO OT-I CD8 cells, we assessed B16-OVA melanoma cell killing in the presence or absence of an anti-IFNγ blocking Ab. Blocking IFNγ significantly reduced killing by MCJ KO OT-I CD8 cells to levels comparable to WT OT-I cells (Fig. 1i). Similar results were found when we tested the effect of blocking IFNγ during the killing assay with MCJ KO OT-I CD8 cell after the 3rd expansion with IL-2 (Supplementary Fig. 3). Thus, enhanced IFNγ secretion by effector CD8 cells lacking MCJ contributes to the superior anti-tumor killing activity of these cells.

Since exocytosis is highly dependent on ATP, we also examined the role of perforin-based cytotoxicity in the superior killing activity of MCJ KO OT-I CD8 cells. WT and MCJ KO OT-I CD8 cells were activated as described above, underwent two rounds of expansion with IL-2, and were pre-treated for 2 h with vehicle or concanamycin A (CMA), which is known to inhibit perforin-based cytotoxicity through accelerated degradation of perforin by increasing the pH of lytic granules[60,61]. OT-I cells were then extensively washed to eliminate the CMA and co-cultured with B16-OVA cells. CMA pretreatment blocked the enhanced killing activity of MCJ KO OT-I CD8 cells (Fig. 1j).

Together, these results indicate that loss of MCJ in IL2-expanded TCR-specific CD8 cells can enhance their antigen-specific anti-tumor-killing activity in vitro by promoting the secretion of IFNγ and their cytotoxic activity.

## MCJ deficiency improves the efficacy of adoptive TCR-specific CD8 cell therapy against solid tumors in vivo

Effector or memory CD8 cells have limited anti-tumor efficacy in the absence of CD4 cells in vivo[62]. We investigated whether loss of MCJ could provide a competitive advantage to TCR-specific adoptive CD8 cell therapy in vivo. B16-OVA cells were implanted in WT mice and when the tumors were palpable (treatment instead of prevention), mice were administered with WT or MCJ KO OT-I CD8 cells that had undergone activation with anti-CD3/anti-CD28 Abs and expansion with IL-2. Cell surface analysis of activation markers by flow cytometry prior to their administration into the mice showed no difference between WT and MCJ KO OT-I CD8 cells (Supplementary Fig. 4a). We measured tumor growth over time after the adoptive transfer of OT-I CD8 cells. WT OT-I CD8 cells failed to control rapid tumor growth (Fig. 2a). In contrast, mice that had received MCJ KO OT-I CD8 cells displayed slower tumor growth (Fig. 2a and Supplementary Fig. 4b) and reduced tumor size at the time of analysis (Supplementary Fig. 4c). Thus, lack of MCJ in TCR-specific CD8 cells results in a superior anti-tumor efficacy of the adoptive CD8 cell therapy.

Analysis of CD8 cells infiltrated in the tumor by immunostaining of tissue sections at the time of euthanasia suggested the tumors from mice that had received MCJ KO OT-I CD8 cells have increased number of CD8 cells relative to tumors in mice with WT OT-I CD8 cells (Fig. 2b). Thus, we performed additional experiments where tumors were harvested 8 days after administration of WT or MCJ KO OTI CD8 cells, and the presence of CD8 cells was determined by flow cytometry using total tumor cell homogenate. The frequency of total CD8 cells within the leukocyte population (CD45[+] cells) in tumors from mice that had received MCJ KO OT-I cells was higher than in the mice with WT OT-I cells (Fig. 2c). Similarly, the frequency of adoptively transferred OT-I CD8 cells (CD8 TCR Vα2[+]) within the leukocyte population in tumors from mice that had received MCJ KO OT-I CD8 cells was higher than in tumors from mice receiving WT OT-I cells (Fig. 2d). Thus, correlating with the enhanced anti-tumor efficacy described above, loss of MCJ in the adoptive CD8 cells results in a higher accumulation of these cells within solid tumors.

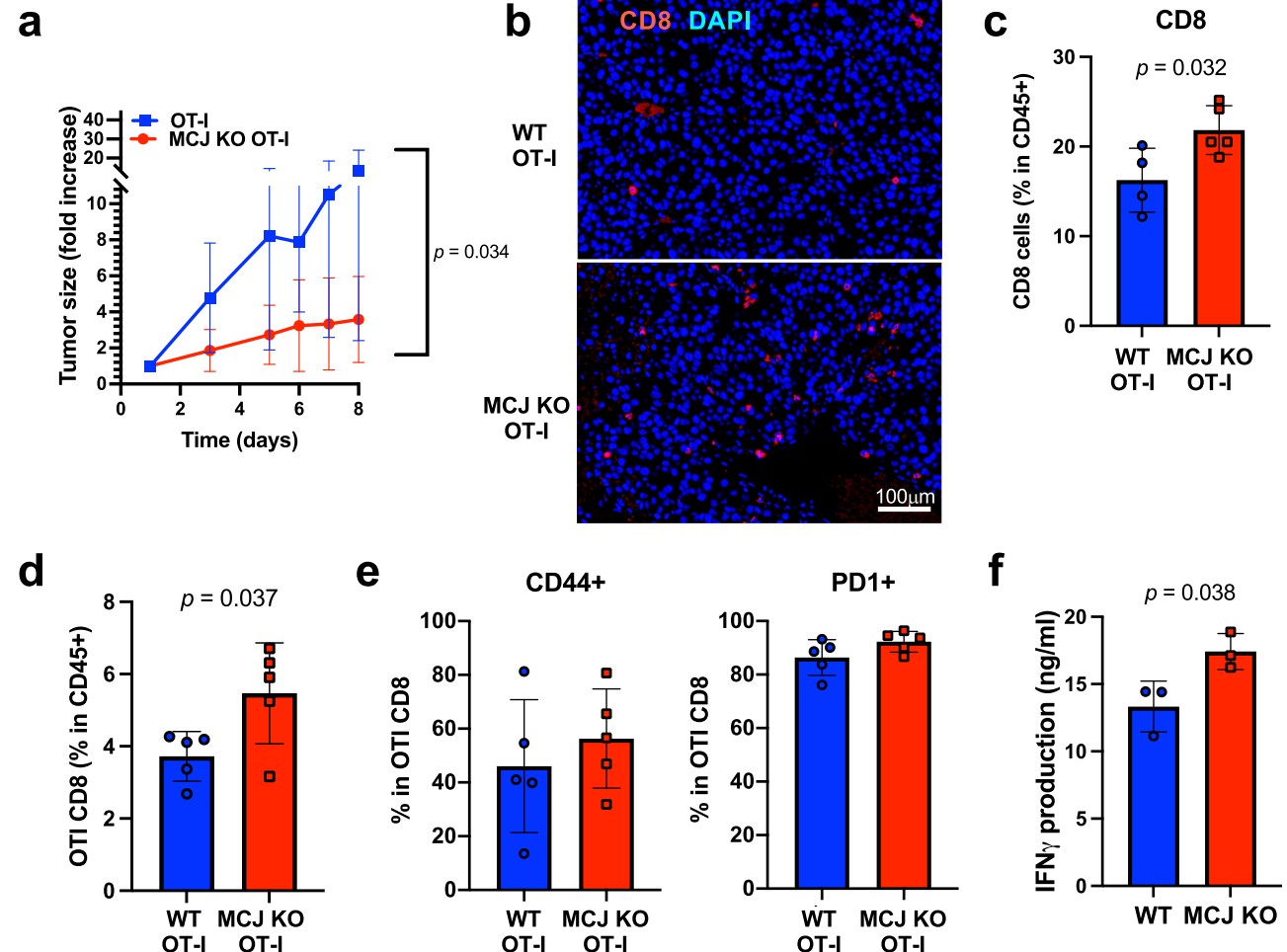

**Fig. 2 | Superior efficacy of MCJ-deficient TCR-antigen specific CD8 cells for the treatment of melanoma in vivo. a** B16-OVA tumor cells ($4 \times 10^5$/mouse) were s.c. injected on the flank to WT mice (n = 7 biologically independent animals per group). WT and MCJ KO OT-I CD8 cells were activated with anti-CD3/anti-CD28 Abs as in Fig. 1, expanded with IL-2 for two expansions, and ($5 \times 10^5$ cells/mouse) i.v. administered to the B16-OVA tumor-bearing mice 10 days post-implantation when the tumors were palpable. Tumor size was followed for 8 days (when a large fraction of mice needed to be euthanized based on tumor size), and it is presented as fold-increase relative to the size of the tumor at the time of the CD8 cell adoptive transfer. **b** Immunostaining for CD8 (red) and DAPI as nuclear marker (blue) of histological sections from B16-OVA tumors harvested 8 days after the adoptive transfer of WT or MCJ KO OT-I CD8 cells, as in (**a**). **c–e** WT and MCJ KO OT-I CD8 cells were activated and expanded as in (**a**), and injected into WT mice (n = 5 biologically independent animals per group) bearing B16-OVA melanoma tumors. Tumors were harvested 10 days after CD8 cell adoptive transfer and tumor cell homogenate was examined by flow cytometry for the presence of total CD8 cells (**c**) or OT-I CD8 cells (**d**) within the leukocyte (CD45+) population, as well as the expression of CD44 and PD1 on OT-I CD8 cells within the OT-I CD8 cells (**e**). **f** WT and MCJ KO OT-I CD8 cells were activated and expanded as in (**a**), and injected into WT mice (n = 4 biologically independent animals per group) bearing B16-OVA tumors. 7 days after CD8 cell adoptive transfer tumors were harvested to isolate tumor infiltrating CD8 cells. Pooled infiltrated CD8 cells were and activated ex vivo with anti-CD3/anti-CD28 Abs for 24 h and IFNγ levels in supernatant was examined by ELISA (n = 3 replicate wells per group). *p* was determined by 2-way ANOVA analysis (**a**) and two-sided unpaired *t* test (**c–f**). Mean ± SD is shown for (**a, c–f**).

We have previously shown that loss of MCJ has no effect on the expression of cell surface markers on CD8 cells post-activation[51]. Consistently, flow cytometry analysis of activation markers (e.g., CD44, PD1) on tumor infiltrated CD8 cells showed no difference between infiltrated WT OT-I and MCJ KO OT-I CD8 cells (Fig. 2e). We then examined the ex vivo ability of tumor infiltrated CD8 cells to produce IFNγ, since loss of MCJ enhances IFNγ secretion but not synthesis[51]. We isolated tumor infiltrated CD8 cells and stimulated with anti-CD3/anti-CD28 Abs ex vivo only for a short period of time to stimulate only those cells that were actively producing IFNγ in vivo. IFNγ in the supernatant was examined by ELISA. Higher levels of IFNγ were detected in infiltrated CD8 cells from tumors that had received MCJ KO OTI CD8 cells (Fig. 2f).

Together, these results show that disrupting MCJ expression in CD8 cells improves the efficacy of adoptive TCR-specific CD8 cell therapy on the treatment of solid tumors, by facilitating the accumulation of tumor-infiltrating transferred CD8 cells as well as boosting their effector function.

### Loss of MCJ enhances in vitro killing activity of CD8 CAR-T cells against of B cell leukemia

CAR-T cell therapy requires the in vitro activation of autologous T cells from patients, transduction of a specific CAR construct and multiple cell expansions to obtain the needed cell number. We therefore investigated whether increasing mitochondrial metabolism by the loss of MCJ could improve the cytotoxicity of CD8 CAR-T cells against leukemia cells. We used the CD19/4-1BB CAR that contains the anti-mouse CD19 single-chain variable fragment (scFv), the CD3ζ signaling domain (CD19-BBz CAR) and the mouse 4-1BB costimulatory domain (Supplementary Fig. 5a). CD8 cells were isolated from WT or MCJ KO mice, activated with anti-CD3/anti-CD28-coated beads and transduced with the CD19-BBz CAR retroviral construct. Analysis of cell surface

expression of the CAR by flow cytometry showed comparable levels of expression of CAR in transduced WT and MCJ KO CD8 cells (Supplementary Fig. 5b). After activation, CD8 CAR-T cells were expanded with IL-2, mimicking the clinical protocol for CAR-T cells manufacturing, and were examined for their in vitro killing activity through co-cultures with E2a B-cell leukemia cells (expressing CD19). We first examined the killing activity of WT and MCJ KO CD8 CAR-T cells after 1st expansion (2 days) cultured with IL-2. While both WT and MCJ KO CD8 CAR-T cells were highly efficient in killing E2a cells, increased killing activity was found for MCJ KO CD8 CAR-T cells (Fig. 3a). We also performed similar killing studies with CD8 CAR-T cells after 3rd expansion. While the killing efficiency of WT CD8 CAR-T cells was lower after the 3rd expansion relative to the 1st expansion, MCJ KO CD8 CAR-T cells retained their high killing efficiency at this time point (Fig. 3b and Supplementary Fig. 5c). Thus, MCJ deficiency can preserve the cytotoxic function of CD8 CAR-T cells during in vitro expansion with IL-2.

After manufacturing CAR-T cells with high dose of IL-2 in vitro, these cells are reinfused into the patients in an IL-2-deprived environment. Since withdrawal of IL-2 has been shown to affect T cell metabolism and function markedly[63,64], we investigated whether MCJ deficiency would give a functional advantage to CD8 CAR-T cells after cytokine withdrawal. WT and MCJ KO CD8 CAR-T cells were expanded with IL-2 for three expansions, extensively washed and incubated in medium alone for 24 h prior to setting the co-cultures with E2a leukemia cells for a killing assay. After cytokine withdraw, MCJ KO CD8 CAR-T cells showed superior killing of E2a target cells compared to WT CD8 CAR-T cells (Fig. 3c). Thus, even in a disadvantage environment such as low IL-2, loss of MCJ preserves the anti-tumor efficacy of CD8 CAR-T cells.

In addition to expansion of CAR-T cells with IL-2, there is an emerging interest in testing other potential cytokines that may better sustain the persistence of these cells[65–68]. IL-7 in combination with IL-15 for expansion of CAR-T cells is currently being tested in trials (NCT04544592, NCT05535855, NCT05098613). We therefore examined whether the loss of MCJ could also provide fitness advantage to CAR-T cells after expanded with IL-7 plus IL-15. WT and MCJ KO CD8 CAR-T cells were generated and grown either with IL-2 or IL-7/IL-15 for three expansions. As expected, we did not find any difference in cell surface markers between WT and MCJ KO CD8 CAR-T cells expanded with IL-2 or with IL-7/IL-15 (Supplementary Fig. 6a, b). We also examined the expression of transcriptional regulators (T-bet, Tox, Foxo1 and Tcf1). As previously described[69–71], there was an increase in the expression of Foxo1 and Tcf1 (associated with increased anti-tumor CD8 cell response) in IL-7/IL-15 CAR-T cells compared with the IL2-expanded CAR-T cells. There was a slight upregulation of Foxo1 and Tcf1 in IL2-expanded MCJ KO CAR-T cells relative to WT CAR-T cells, but there was no difference between WT and MCJ KO CAR-T cells expanded with IL-7/IL15 (Supplementary Fig. 6c). The expression of Tox (marker for exhausted CD8 cells but also effector memory CD8 cells[72,73]) was high in all cells (Supplementary Fig. 6c). In line with our results using IL-2 for the expansion, MCJ KO CD8 CAR-T cells expanded with IL-7/IL-15 retained their superior killing activity relative to WT CD8 CAR-T cells (Supplementary Fig. 6d).

We investigated whether the superior tumor cell killing achieved by MCJ KO CD8 CAR-T cells could be due to increasing IFNγ production or cytotoxicity as observed with the OT-I CD8 cells. We examined the levels of IFNγ released in the supernatant during the killing assay with either WT or MCJ KO CD8 CAR-T cells after the 3rd expansion with IL-2. Higher levels of IFNγ were present in the co-cultures of E2a cells with MCJ KO CD8 CAR-T cells relative to those with WT CD8 CAR-T cells (Fig. 3d). Similarly, increased levels of IFNγ were produced by MCJ KO CD8 CAR-T cells that underwent IL-2 withdrawal prior to co-culture with E2a, relative to the levels of IFNγ produced by WT CD8 CAR-T cells (Fig. 3e). Although CAR-T cells killing of leukemia cells has been shown to be independent on the IFNγ pathway[74], we investigate whether the increased production of IFNγ could contribute to the superior killing activity of MCJ KO CD8 CAR-T cells, we performed killing assays in the presence or absence of anti-IFNγ Ab. However, blocking IFNγ had no effect on the killing activity of either WT or MCJ KO CD8 CAR-T cells (Fig. 3f). These results indicated that the superior killing of leukemia cells by MCJ KO CD8 CAR-T cells is independent of IFNγ, consistent with previous studies showing IFNγ−independent CAR-T cell activity against some hematologic cancers, although IFNγ has also been reported to contribute to CAR-T cell activity in some solid tumors and some hematologic malignancies[74–77].

Although Fas-FasL interaction can also mediate apoptosis of cancer cells, this does not seem to be the mechanism for killing of E2a cells by CAR-T cells since no Fas expression was detected on E2a cells when analyzed by flow cytometry (Supplementary Fig. 7). Degranulation of perforin and granzyme has been shown to be one of the main mechanisms CAR-T cells utilize to kill their target cells[78,79]. We examined the expression of CD107a, a lysosomal surface protein used as marker of cytotoxic degranulation[80], in WT and MCJ KO CD8 CAR-T cells (3rd expansion with IL-2) upon co-culture with E2a cells. Increased CD107a expression was detected in MCJ KO CD8 CAR-T cells (Fig. 3g). To examine whether superior killing of MCJ KO CD8 CAR-T cells was due to an enhanced cytotoxic degranulation, we performed a killing assay in the presence of EGTA, a chelating agent for $Ca^{2+}$ needed for perforin-based cytotoxicity[81]. EGTA treatment suppressed the superior killing activity of MCJ KO CD8 CAR-T cells (Fig. 3h). In addition, we also performed a killing assay in the presence or absence of CMA. CMA significantly inhibited the killing of E2a leukemia cells by MCJ KO and WT CD8 CAR-T cells (Fig. 3i).

Together, these results demonstrated that MCJ-deficient CD8 CAR-T cells are superior in killing leukemia cells in vitro, due to their enhanced cytotoxicity activity.

## Superior efficacy of MCJ-deficient CD8 CAR-T cells in eradicating B cell leukemia in vivo

To investigate the in vivo efficacy of the MCJ-deficient CD8 CAR-T cells for the treatment of leukemia, E2a leukemia cells were intravenously (i.v.) injected to the WT host mice, and 3 days later mice received sub-lethal irradiation for lymphodepletion, as previously described[79,82,83]. CD19-BBz CD8 CAR-T cells were generated using WT and MCJ KO CD8 cells, and expanded with IL-2. As described above, after the 3rd expansion with IL-2, there was no difference in activation markers expression between WT or MCJ KO CD8 CAR-T cells (Supplementary Fig. 8a). These cells were administered to leukemia-bearing mice by i.v. injection one day after lymphodepletion. Similar to the in vitro expansion of the CAR-T cells, no difference could be detected in the expression of cell surface markers (CD44, CD62L, PD1 and Tim3) between MCJ KO and WT CAR-T cells in vivo after 7, 21, or 28 days post-transfer (Supplementary Fig. 8b–d).

We then followed the survival of the mice over 80 days to account for potential relapse. As control, a cohort of mice did not receive T cell therapy, and all untreated mice succumbed rapidly to leukemia progression, as expected (Fig. 4a). While only about 35% of the leukemia bearing mice treated with WT CD8 CAR-T cells survived, all the mice that had received MCJ KO CD8 CAR-T cells survived for the duration of the study (Fig. 4a). Thus, lack of MCJ in CD8 CAR-T cells markedly improves their in vivo anti-tumor efficacy.

To further show that the superior in vivo efficacy of MCJ-deficient CD8 CAR-T cells was due to a better effector function of these cells we performed ex vivo killing assay with CAR-T cells recovered from the tumor site (i.e., bone marrow). Following the same protocol described above, we generated CD8 CAR-T cells using WT and MCJ KO CD8 cells, expanded with IL-2, and after the 3rd expansion, they were administered i.v. to WT leukemia-bearing mice. After 30 days, CD8 cells were isolated from the bone marrow of CAR-T cell treated mice and equal

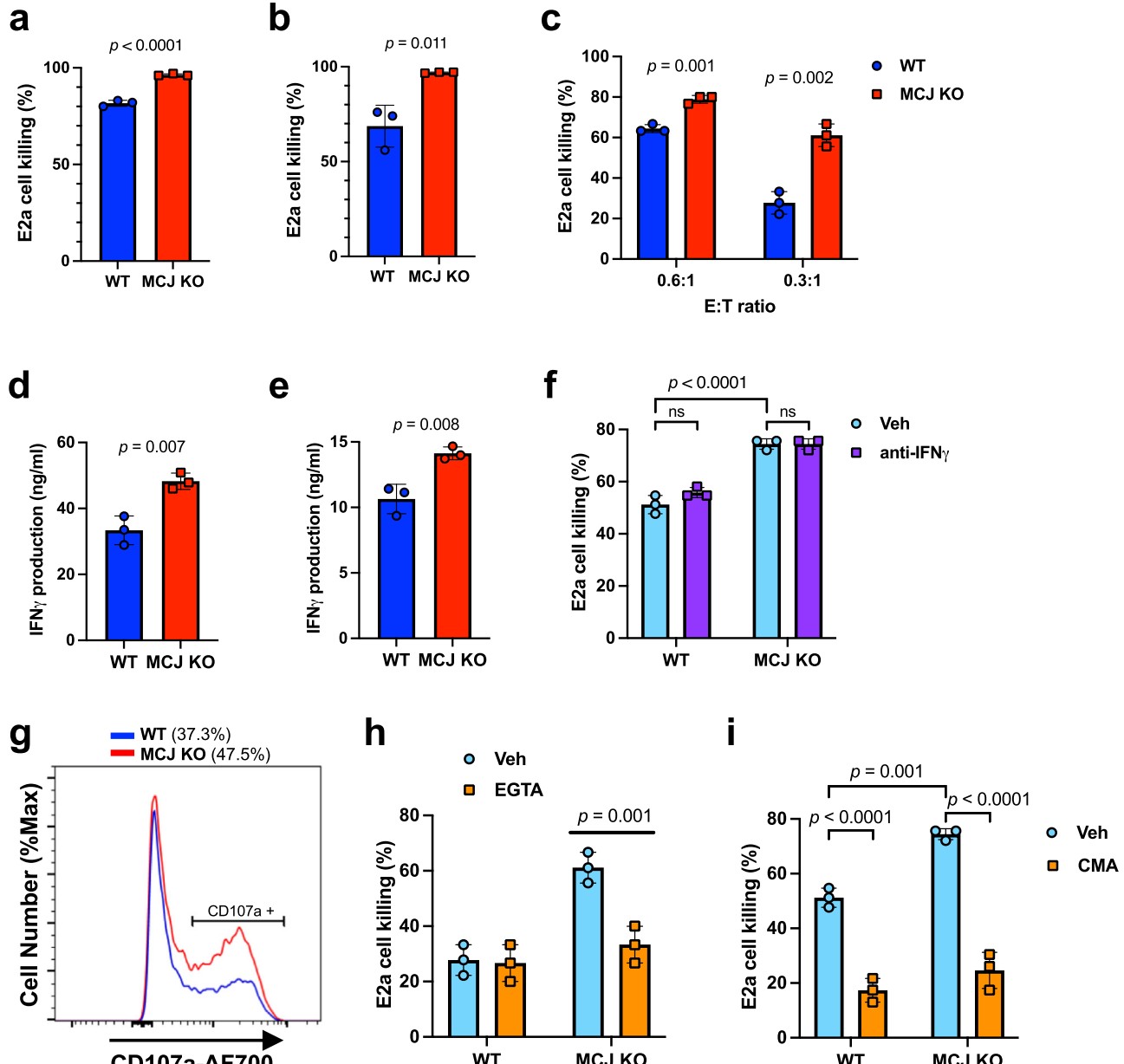

**Fig. 3 | MCJ-deficient CD8 CAR-T cells are superior in perforin-mediated killing of leukemia B cells in vitro. a** WT or MCJ KO CD8 cells were activated with anti-CD3/anti-CD28 Abs for 1 day, retrovirally transduced with CD19-BBz CAR, expanded with IL-2 (60 IU/ml) for 2 days (1st expansion), and co-cultured with E2a leukemia cells (E:T = 0.5) for killing assay. Live E2a cells were examined by flow cytometry after 15 h ($n = 3$ biologically independent cells). **b** WT or MCJ KO CD8 cells were activated and transduced with CD19-BBz CAR as in (**a**). After three expansions with IL-2, CD8 CAR-T cells were co-cultured with E2a leukemia cells (E:T = 0.5) for killing assay ($n = 3$ biologically independent cells). **c** WT or MCJ KO CD8 cells were activated and transduced with CD19-BBz CAR. After the 3rd expansion with IL-2, cells were washed, and incubated in medium along for 24 h. CAR-T cells were then co-cultured with E2a leukemia cells at the given E:T ratio for a 5 h killing assay ($n = 3$ biologically independent cells). **d** IFNγ production during the killing assay with WT and MCJ KO CD8 CAR-T cells generated as in (**b**) and E2a leukemia cells (E:T = 0.5), as determined by ELISA ($n = 3$ biologically independent cells). (**e**) IFNγ production during the killing assay with WT and MCJ KO CD8 CAR-T cells generated as in (**c**)

and E2a leukemia cells (E:T = 0.3) ($n = 3$ biologically independent cells). **f** WT or MCJ KO CD8 cells were activated and transduced with CD19-BBz CAR as in (**a**). After the 3rd expansion with IL-2, cells were washed, and incubated in medium alone for 24 h. CAR-T cells were then co-cultured with E2a leukemia cells (E:T = 0.3) in the presence or absence of an anti-IFNγ blocking Ab for a killing assay ($n = 3$ biologically independent cells). **g** CD107a expression on 3rd-expansion WT and MCJ KO CD8 CAR-T cells after 4 h of co-culture with E2a cells, as determined by flow cytometry. % of CD107a+ cells is shown. **h** After the 3rd expansion with IL-2 followed by 24 h resting in medium, WT and MCJ KO CD8 CAR-T cells were co-cultured with E2a cells (E:T = 0.3) in the presence or the absence of EGTA (3 mM)/MgCl₂ (2 mM). Live E2a cells were counted after 5 h ($n = 3$ biologically independent cells). **i** After the 3rd expansion with IL-2 followed by 24 h resting in medium, WT and MCJ KO CD8 CAR-T cells were pre-treated with CMA (100 nM) or vehicle for 2 h, washed and co-cultured with E2a cells (E:T = 0.4) for 5 h. Live E2a cells were counted ($n = 3$ biologically independent cells). $p$ was determined by two-sided unpaired $t$ test (**a**–**e**) or 2-way ANOVA (**f**, **h**, **i**). Mean ± SD is shown for all figures with bar graph.

number of CAR-T cells were co-cultured with E2a cells ex vivo to assess cytotoxic function. Increased E2a cell killing was observed in CD8 cells isolated from bone marrow of mice that had received MCJ KO CD8 CAR-T cells (Fig. 4b). Thus, MCJ-deficient CD8 CAR-T cells retained

their cytotoxic effector function in vivo, correlating with their superior efficacy in increasing overall survival.

Since our in vitro data suggest that expansion with IL-2 over time compromises the anti-tumor activity of WT CD8 CAR-T cells

## a

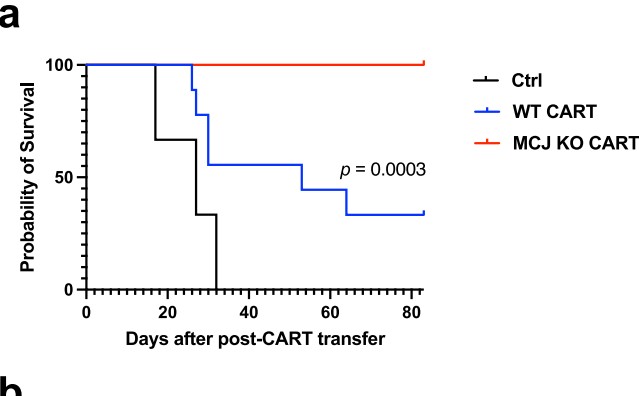

## b

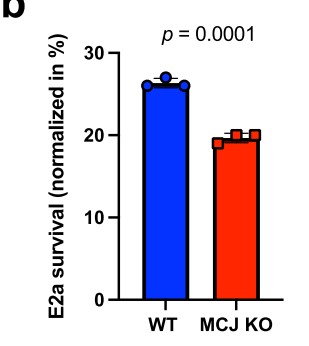

## c

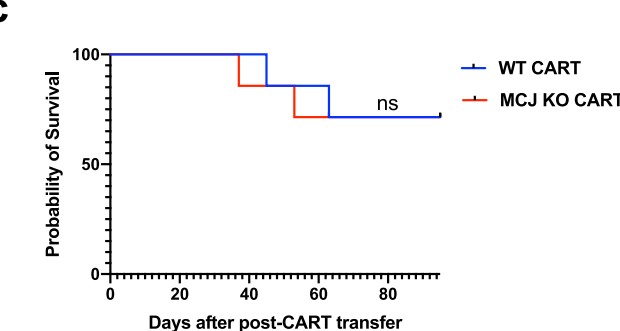

**Fig. 4 | Lack of MCJ in CD8 CAR-T cells improves their efficacy against the leukemia B cells in vivo. a** E2a cells ($10^6$/mouse) were injected i.v. into WT host mice (*n* = 9 biologically independent animals per group). 3 days later, mice were sublethally irradiated (500 cGy), and the next day they received an i.v. administration of PBS alone as vehicle (Ctrl), WT CD19-BBz CD8 CAR-T cells or MCJ KO CD19-BBz CD8 CAR-T cells ($10^5$ CAR-T cells/mouse) that were expanded with IL-2 for 3 expansions. Survival of the mice post-treatment with CAR-T cells was recorded over time. Kaplan-Meier survival analysis is shown. **b** WT host mice were administered with E2a cells, irradiated and treated with WT or MCJ KO CD19-BBz CD8 CAR-T cells ($10^6$ CAR-T cells/mouse) as described in (**a**). 21 days after the treatment with CAR-T cells, bone marrow was harvested to isolate CD8 cells that were then used for an ex vivo killing assay with E2a cells as targets (E:T = 0.05, based on the percentage of the CAR + CD8 cells). Same number of CAR + CD8 cells were co-cultured with E2a cells. Live E2a cells were counted after 20 h co-culture (*n* = 3 replicate wells per group). **c** Kaplan–Meier survival analysis of leukemia-bearing mice treated as in (**a**) with WT or MCJ KO CD19-BBz CD8 CAR-T cells after the 1st expansion with IL-2 (*n* = 7 biologically independent animals per group). *p* was determined by log-rank (Mantel–Cox) test (**a, c**) or two-sided unpaired *t* test (**b**). Mean ± SD is shown for (**b**).

(Fig. 3a, b), we tested the in vivo anti-tumor efficacy of WT and MCJ KO CD8 CAR-T cells that were expanded with IL-2 only for one expansion. No difference was observed in the expression of activation markers between WT and MCJ KO CD8 CAR-T cells after the 1st expansion, prior to the infusion into host mice (Supplementary Fig. 9). These cells were then administered to leukemia-bearing mice and survival of the host mice was followed. Interestingly, after a single round of expansion with

IL-2, WT and MCJ KO CD8 CAR-T cells equally prolonged the survival of mice with leukemia (Fig. 4c).

Together, these results show that progressive expansion of CD8 CAR-T cells with IL-2 (normally performed in the clinic) compromises their fitness and attenuates their anti-tumor efficacy, but the enhanced metabolic fitness provided by the loss of MCJ preserves their anti-tumor activity and overall in vivo efficacy.

### MCJ deficiency enhances the mitochondrial metabolism of CD8 CAR-T cells during the expansion with IL-2

MCJ acts as an endogenous negative regulator of Complex I[51–55]. IL-2 has been shown to upregulate MCJ during the expansion of CD8 cells[59]. It is well known that IL-2 favors glycolysis over mitochondrial respiration during the expansion of effector CD8 cells[35,36,59]. Therefore, we investigated whether loss of MCJ in IL-2 expanded CD8 CAR-T cells could improve mitochondrial metabolism. We first examined mitochondrial membrane potential (MMP) by TMRE staining and flow cytometry, as a measure of ETC activity. WT and MCJ KO CD8 CAR-T cells were generated as described above, and were cultured with IL-2. There was no difference in MMP between WT and MCJ KO CD8 CAR-T cells after the 1st expansion in IL-2 post-activation, with high MMP observed in both groups (Fig. 5a). Similarly, there was no difference in mitochondrial mass, as determined by Mitotracker staining, between WT and MCJ KO CD8 CAR-T cells after the 1st IL-2 expansion (Fig. 5b). However, after the 3rd expansion with IL-2, WT CD8 CAR-T cells showed a marked loss of MMP relative to the MMP after the 1st expansion (Fig. 5a, c). In contrast, MCJ KO CD8 CAR-T cells retained high MMP after the 3rd expansion (Fig. 5a, c), and showed higher MMP than the WT CD8 CAR-T cells (Fig. 5c). The increased MMP in MCJ KO CD8 CAR-T cells was not caused by an increase in mitochondrial mass, as determined by Mitotracker staining (Fig. 5d), but represented increased ETC activity in the absence of the inhibitory effect of MCJ on Complex I. In line with our previous studies on CD8 cells[51,52], the increased MMP present in MCJ KO CD8 CAR-T cells did not result in increased, but decreased, mitochondrial ROS production (Supplementary Fig. 10a, b). Thus, over time during the expansion with IL-2, WT CD8 CAR-T cells lose their mitochondrial membrane potential, but the absence of MCJ helps to sustain their MMP and mitochondrial metabolism.

To examine whether the increased MMP found in MCJ KO CD8 CAR-T cells also results in enhanced mitochondrial respiration and ATP production, we performed Seahorse MitoStress assay in purified WT and MCJ KO CD8 CAR-T cells after the 3rd expansion with IL-2. MCJ KO CD8 CAR-T cells had higher maximal respiration compared to WT CD8 CAR-T cells (Fig. 5e, f). Spare respiratory capacity was also higher in MCJ KO CD8 CAR-T cells (Fig. 5e, f). Thus, according to its role as a negative regulator of mitochondrial respiration, the absence of MCJ increases the mitochondrial respiration of IL-2-expanded CD8 CAR-T cells. In addition, using the Seahorse ATP production rate assay, we found increased mitochondrial and total ATP production rate in MCJ KO CD8 CAR-T cells (Supplementary Fig. 11).

To interrogate the impact of MCJ on CD8 CAR-T cell metabolism, we performed unbiased metabolomics analysis on purified WT and MCJ KO CD8 CAR-T cells after three expansions with IL-2. Supporting higher mitochondrial respiration, the levels of ATP and ADP were elevated in MCJ KO CD8 CAR-T cells relative to WT CD8 CAR-T cells (Fig. 5g and Supplementary Fig. 12a, b), indicative of a higher energy state (Supplementary Fig. 12c). In addition, the levels of $NAD^+$, which is in part generated by Complex I activity, were also elevated in MCJ KO CD8 CAR-T cells (Fig. 5g and Supplementary Fig. 12a, b). Heat map analysis also show an accumulation of TCA metabolites (α-ketoglutarate/2-oxoglutarate, succinate) in MCJ KO CD8 CAR-T cells (Fig. 5g and Supplementary Fig. 12a, b), supporting the enhanced mitochondrial metabolism in CD8 CAR-T cells in the absence of MCJ. Principal component analysis (PCA) of the metabolomics data further showed the

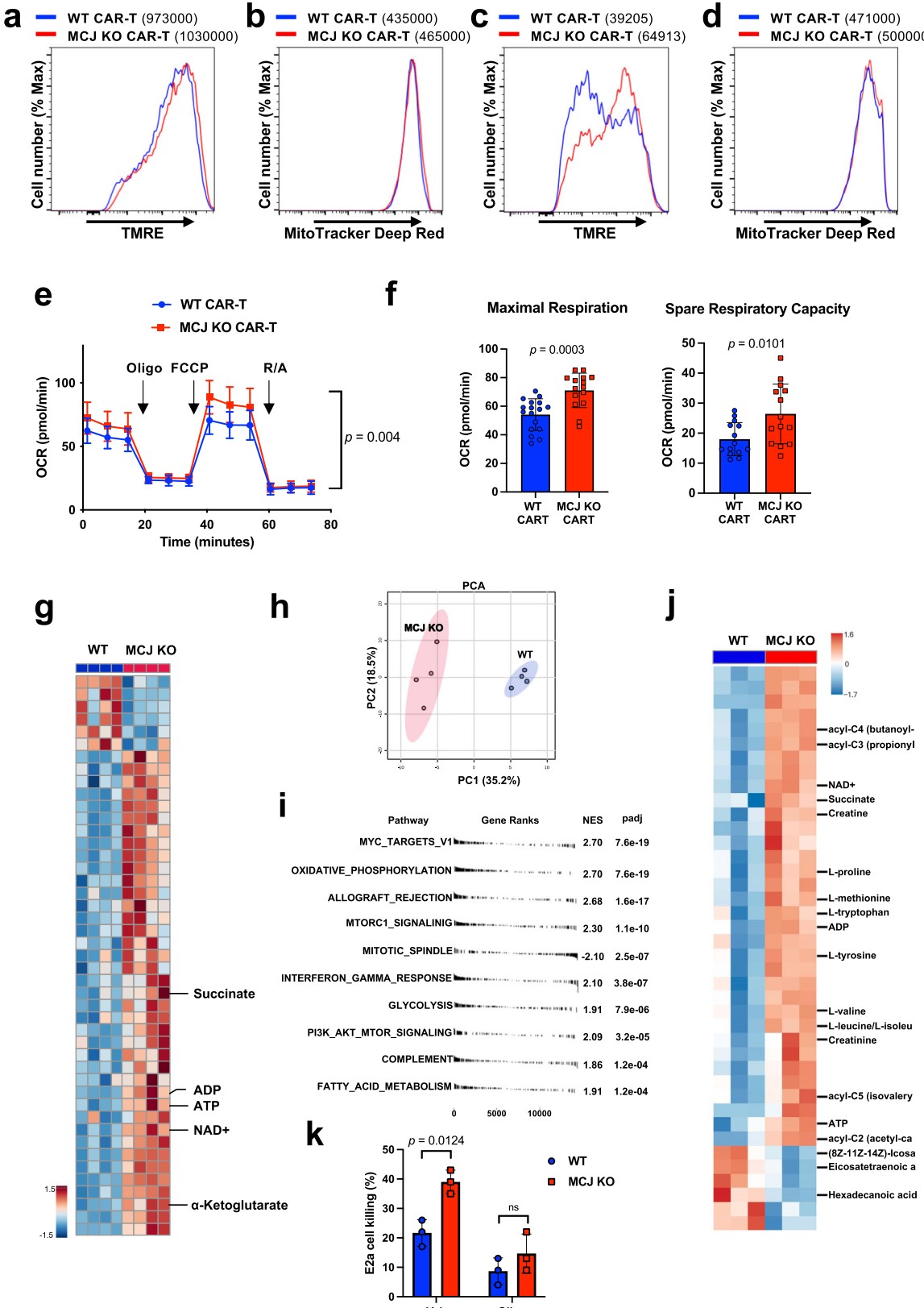

distinct metabolic profile between WT and MCJ KO CD8 CAR-T cells (Fig. 5h).

To further demonstrate a selective effect of MCJ deficiency on the metabolism of CD8 CAR-T cells, we also performed bulk RNAseq using isolated WT and MCJ KO CD8 CAR-T cells after 3 expansions with IL-2. PCA defined WT and MCJ KO CAR-T cells as displaying a distinct gene

expression profile (Supplementary Fig. 13a, b). We then identified the top 10 gene set enrichment analysis (GSEA) pathways that distinguished WT and MCJ KO CD8 CAR-T cells based on normalized enrichment score (NES) and adjust p-value. Interesting, most of these pathways are associated to metabolism (Fig. 5i). Consistent with the metabolomics results, GSEA showed upregulated oxidative

**Fig. 5 | MCJ deficiency enhances the mitochondrial fitness of CD8 CAR-T cells.** **a**–**d** WT or MCJ KO CD8 cells were activated with anti-CD3/anti-CD28 Abs, transduced with CD19-BBz CAR, and expanded with IL-2. After the 1st expansion (**a** and **b**) or the 3rd expansion (**c** and **d**) with IL-2, cells were stained with TMRE (**a** and **c**) or Mitotracker (**b** and **d**) and analyzed by flow cytometry. Numbers in parenthesis show the geometric mean fluorescence intensity. **e, f** After three expansions with IL-2, WT and MCJ KO CAR-T cells (as in (**a**)) were purified and used for Seahorse MitoStress assay. OCR was measured at baseline and in response to oligomycin (Oligo), FCCP and rotenone with antimycin (R/A) (*n* = 16 replicate wells per group) (**e**). Maximal respiration and spare respiratory capacity (**f**) were determined. **g, h** Mass spectrometry-based metabolomics analysis of purified WT and MCJ KO CD8 CAR-T cells after three expansions with IL-2 (*n* = 4 replicate cells per group). Hierarchical clustering analysis of the relative abundance of top 50 distinct metabolites shown in heat map from two groups. Metabolites of interests are highlighted (**g**). PCA of two groups (**h**). **i** WT or MCJ KO CD8 cells were activated and transduced to generate CD19-BBz CAR-T cells as in (**a**). After 3 expansions with IL-2, WT and MCJ KO CAR-T cells are purified and used for RNAseq. The top 10 distinctly differential pathways in GSEA HALLMARK pathways are shown as water plot. **j** Mass spectrometry-based metabolomics analysis of purified WT and MCJ KO CD8 CAR-T cells after three expansions with IL-2 followed by 48 h in cytokine-free medium (*n* = 3 replicate cells per group). Hierarchical clustering analysis of the relative abundance of top 40 distinct metabolites between two groups shown in heat map. Metabolites of interests are highlighted. **k** After 3 expansions with IL-2 followed by 24 h in cytokine-free medium, WT and MCJ KO CD8 CAR-T cells were pretreated with oligomycin (5 μM) or DMSO (Veh) for 3 h, washed and co-cultured with E2a cells for 4 h for killing assay (*n* = 3 biologically independent cells). *p* was determined by two-way ANOVA (**e, k**), two-sided unpaired *t* test (**f, g, j**). **i** *p*-adj was calculated using Benjamini–Hochberg procedure using 10,000 permutations. Mean ± SD is shown (**e, f, k**).

phosphorylation pathways in MCJ KO CD8 CAR-T cells. Moreover, glycolysis and fatty acid metabolism were upregulated pathways in MCJ KO CAR-T cells (Supplementary Fig. 13c), suggesting that MCJ KO CD8 CAR-T cells may also use fatty acid oxidation as an additional carbon source to glucose to promote mitochondrial respiration. Interestingly, the results from these RNAseq analyses also revealed higher expression of *Tcf7* (encoding Tcf1*), Ifitm3, Itgae* (encoding CD103), and *Sell* (encoding CD62L) genes in MCJ KO CD8 CAR-T cells, all of which are involved in the generation or survival of central memory or tissue resident memory CD8 cells[84–87] (Supplementary Fig. 13d). In contrast, the expression of *Havcr2* (encoding Tim3), involved in loss of T cell function[88], was decreased in MCJ KO CAR-T cells (Supplementary Fig. 13d). Thus, correlating with their enhanced metabolism, IL-2 expanded MCJ KO CAR-T cells display more memory-like properties than WT CAR-T cells.

We examined the metabolic profile of WT and MCJ KO CD8 CAR-T cells after three expansions with IL-2 followed by incubation in cytokine-free media to mimic the cytokine withdraw that CAR-T cells undergo when transferred in vivo (i.e., into patients). Similar to the results obtained after the 3 expansions with IL-2, there was a distinct metabolic profile between WT and MCJ KO CAR-T cells following cytokine withdrawal as determined by PCA (Supplementary Fig. 14a). Hierarchical clustering analysis of the top 40 metabolites by t-test also showed increased levels of ATP/ADP and NAD⁺ in MCJ KO CAR-T cells (heat map in Fig. 5j and Supplementary Fig. 14b). Interestingly, the levels of short-chain acyl-carnitines (e.g., acyl-C2, -C3, -C4, -C5) were also higher in MCJ KO CD8 CAR-T cells (Fig. 5j and Supplementary Fig. 14b). These results correlated with the increased levels in MCJ KO CAR-T cells of valine, leucine and isoleucine (Fig. 5j and Supplementary Fig. 14b), branched-chain amino acids whose catabolism generates ketoacids moieties that supply short-chain acyl-carnitine pools[89]. Multiple amino acids beyond Leu/Ile were elevated in MCJ KO cells, including methionine, proline, tryptophan, tyrosine, valine, as well as arginine metabolites creatine and creatinine (Fig. 5j), similar to previous observation in MCJ KO CD8⁺ T cells[51]. Elevation in acyl-carnitine pools were also accompanied by inverse trends in very long chain highly unsaturated fatty acids (FA 20:4; 20:5) and palmitate (FA 16:0), which is interested in light of the established role of acyl-carnitine in FA shuttling into the mitochondria. Altogether these results suggest that - under cytokine withdraw (i.e., starvation conditions) - MCJ deficiency may facilitate both protein catabolism and enhance fatty acid oxidation in the mitochondria through the TCA cycle.

We then investigated whether the enhanced cancer cell killing activity of the MCJ deficient CAR-T cells was due to enhanced mitochondrial respiration. CD8 CAR-T cells from WT and MCJ KO mice were generated and expanded with IL-2 as described above, pretreated with oligomycin (inhibitor of Complex V/ATP-synthase), washed and assayed for their ability to kill E2a cells. The superior killing activity of MCJ KO CD8 CAR-T cells was abrogated by the treatment with

oligomycin (Fig. 5k). Thus, the superior anti-tumor activity of MCJ KO CD8 CAR-T cells is due to the enhanced mitochondrial respiration in the absence of MCJ.

We also examined the metabolism of CD8 CAR-T cells in vivo upon infusion in mice bearing leukemia cells. WT and MCJ KO CD8 CAR-T cells were generated, and after three expansions with IL-2 they were transferred to mice with E2a leukemia cells, as described above. Four days later, CAR-T cells were isolated from bone marrow and underwent metabolomic analysis. While there was a limited number of different metabolites between the recovered WT and MCJ KO CAR-T cells, distinct metabolic profiles were observed (Supplementary Fig. 15a). The most significant changes include a clear accumulation of short/medium-chain FA (e.g., hexanoic acid, heptanoic acid, tetradecanoid acid) and depletion in long chain FA (20:4) and acyl-carnitines (C20; C20:4) in MCJ KO CAR-T cells (Supplementary Fig. 15b). Pathway enrichment analysis indicated likely alterations in fatty acid metabolism and beta oxidation in MCJ KO CAR-T cells (Supplementary Fig. 15c). Short-chain FA have been shown to enhance memory potential of activated CD8 cells[90], according to the role of fatty acids β-oxidation in memory CD8 cells[90,91]. Of note, kynurenine levels were also elevated in MCJ KO CD8 cells, while its precursor (tryptophan, formyl-kynurenine) and downstream metabolites (anthranilate, hydroxykynurenic acid) were depleted, suggesting a rewiring in kynurenine metabolism (Supplementary Fig. 15b). Although the production of kynurenine by tumor cells is considered immunosuppressive for anti-tumor immune response[92], kynurenine may be imported from the bloodstream via large amino acid transporter 1 (LAT1 – also known as SLC7A1)[93] and may promote β-oxidation inside CD8 T cells[94], further exacerbating the MCJ KO metabolic effect on this pathway. In contrast to kynurenine, the levels of hydroxykynurenic acid, a kynurenine metabolite also known to be immunosuppressor[95], were higher in WT CD8 CAR-T cells (Supplementary Fig. 15b). Thus, silencing MCJ also modulates the metabolism of CD8 CAR-T cells in vivo, and impacts mitochondrial fatty acid oxidation and kynurenine metabolism.

Together, these data show that the expansion of CD8 CAR-T cells in IL-2 attenuated mitochondrial respiration in these cells compromising their ability to kill cancer cells in vivo, and that loss of MCJ in CAR-T cells enhances mitochondrial metabolism and overall metabolic fitness, resulting in an improved anti-tumor efficacy in vivo.

## MCJ regulates mitochondrial metabolism and effector function in human CD8 cells

Despite the known role of MCJ as a metabolic regulator of mouse CD8 cells[51], no studies have investigated the presence and role of MCJ in primary human CD8 cells. Thus, we first examined the MCJ protein expression in CD4 and CD8 cells purified from peripheral blood mononuclear cells (PBMC) of healthy volunteers, by Western blot analysis. Similar to MCJ expression in mouse T cells[52], MCJ was more abundantly expressed in human CD8 cells than in CD4 cells (Fig. 6a).

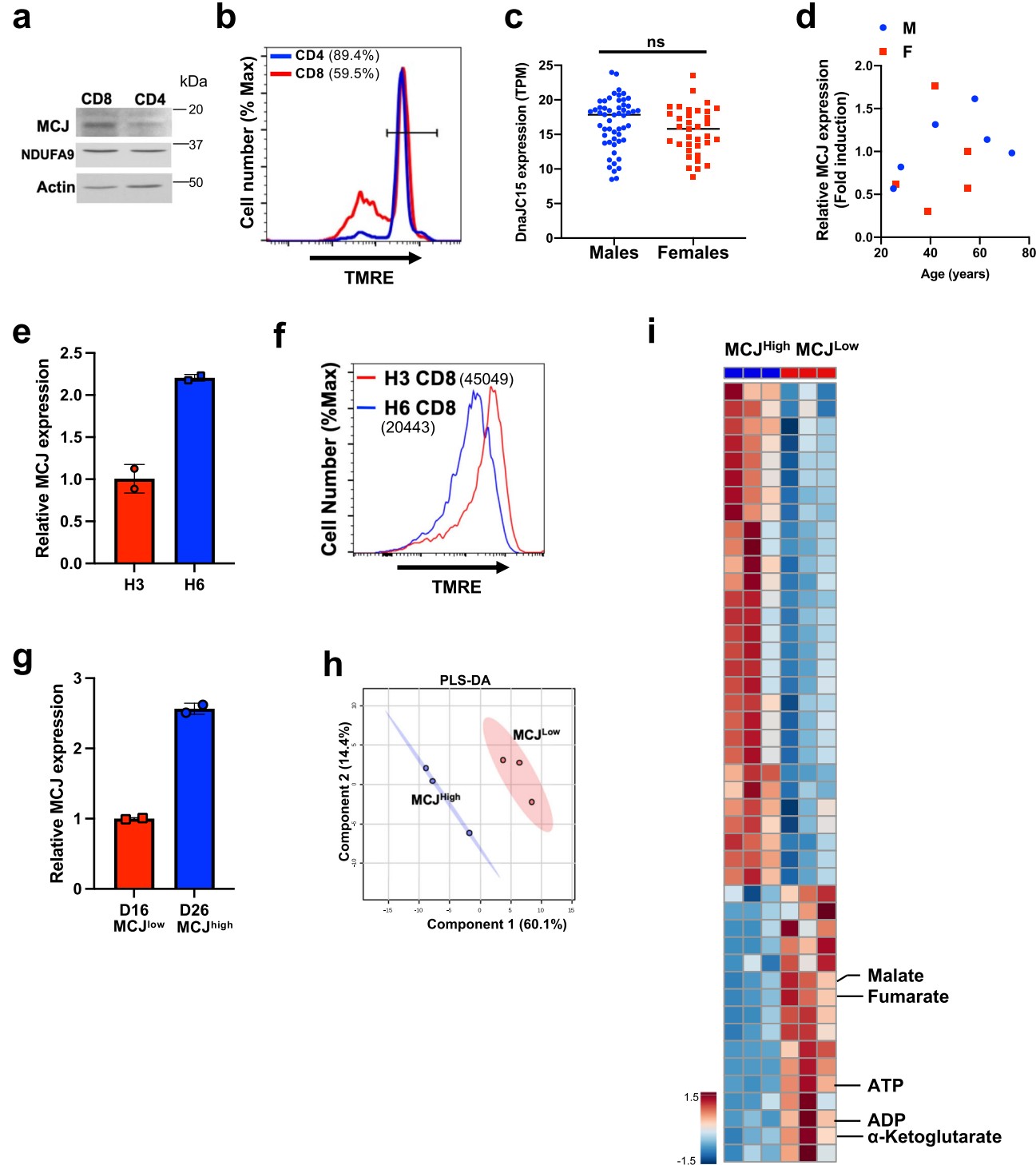

**Fig. 6 | MCJ regulates mitochondrial metabolism and effector function in human CD8 cells. a** MCJ expression by Western blot analysis using whole cell extracts from CD4 and CD8 cells freshly isolated from peripheral blood of a healthy volunteer. NDUFA9 subunit of Complex I was examined as a mitochondria content normalization control. **b** MMP by TMRE staining in combination with staining for CD4 and CD8 of freshly obtained PBMC by flow cytometry. The percentage of TMRE[high] cells is shown in parenthesis. **c** Data from normalized MCJ/*DnaJC15* mRNA levels (transcript per million, TPM) in human naive CD8 cells from healthy subjects (54 males and 35 females) were obtained from DICE database. **d** MCJ/*DnaJC15* mRNA levels in freshly isolated CD8 cells from 6 male (M) and 5 female (F) healthy volunteers assayed by real-time RT-PCR. β2-microglobulin was used as house-keeping gene for normalization. **e, f** MCJ mRNA levels in freshly isolated CD8 cells from healthy donors H3 and H6 were examined by real-time RT-PCR using HPRT as housekeeping gene for normalization (*n* = 2 replicates for each donor) (**e**). CD8 cells

from both H3 and H6 donors were activated with anti-human CD3 (10 μg/ml) and anti-human CD28 (2 μg/ml) Abs for 2 days, expanded with IL-2 (100 IU/ml) and after two expansions MMP was examined by TMRE staining. Numbers in parenthesis show the mean fluorescence intensity (**f**). **g–i** CD8 cells were isolated from healthy donors D16 and D26 and MCJ mRNA levels were assayed by real time RT-PCR, using HPRT as housekeeping gene (*n* = 2 replicates for each donor) (**g**). CD8 cells from both donors were activated with anti-CD3/anti-CD28 Abs, transduced with a human CD19-BBz CAR, and expanded with IL-2. After three expansions, purified CAR-T cells were used for metabolomics analysis (*n* = 3 replicate cells per group). Partial least squares-discriminant analysis (PLS-DA) of D16 (MCJ[low]) and D26 (MCJ[high]) CAR-T cells (**h**). Hierarchical clustering analysis of the relative abundance of top 45 distinct metabolites shown in heat map from D16 (MCJ[low]) and D26 (MCJ[high]) CAR-T cells. Metabolites of interests are highlighted (**i**). *p* was determined by two-sided unpaired *t* test (**c, i**). Mean is shown for (**c**). Mean ± SD is shown for (**e, g**).

We examined MMP in CD4 and CD8 cells from healthy donor PBMC. Correlating with higher levels of MCJ, human CD8 cells have lower MMP compared with CD4 cells (Fig. 6b), consistent with what has previously been shown in mouse CD8 cells[51]. To further examine the expression of MCJ in CD8 cells in the healthy human population, we performed data mining in the DICE (Database of Immune Cell Expression/eQTLs/epigenomics) on human naïve CD8 cells[96]. We found variable expression of MCJ (up to 3-fold difference) in CD8 cells among healthy individuals regardless of sex (Fig. 6c). To further investigate whether MCJ expression varies depending on sex and age, we examined MCJ expression by real time RT-PCR on CD8 cells freshly isolated from PBMC of 11 healthy volunteers with a large range of age and both sexes. The results confirmed the heterogeneity of MCJ expression in CD8 cells in the human population that did not segregate with sex or age (Fig. 6d). We also examined the expression of MCJ in naive CD8 cells, effector memory (Tem) CD8 cells and effector memory cells re-expressing CD45RA CD8 cells (Temra) isolated by cell sorting from a healthy donor and found no differences in the levels of MCJ expression (Supplementary Fig. 16). Thus, the heterogeneity of MCJ expression in CD8 cells among the human healthy population does not seem to be due to a selective overrepresentation of a CD8 cell subset.

To investigate whether low MCJ levels in human CD8 cells also correlate with enhanced mitochondrial metabolism as it does in mouse CD8 cells, we isolated CD8 cells from the PBMC of a healthy donor with relatively lower levels of MCJ (H3) and a donor with relative higher levels of MCJ (H6), as determined by real-time RT-PCR (Fig. 6e). CD8 cells were activated with anti-CD3/anti-CD28 Abs for 2 days, followed by two expansions with IL-2. We then examined MMP by TMRE staining. Consistent with mouse CD8 cells, MMP was higher in CD8 cells with lower MCJ levels (healthy donor H3) relative to the CD8 cells with higher MCJ expression (healthy donor H6) (Fig. 6f), suggesting that MCJ also acts as a negative regulator of mitochondrial metabolism in human CD8 cells.

We performed unbiased metabolomics studies in human CD8 CAR-T cells, transduced with a CD19-BBz CAR, from a donor with relatively low MCJ (healthy donor D16) and a donor with high MCJ expression (healthy donor D26) (Fig. 6g). Metabolomics analyses were performed on isolated CD8 CAR-T cells after three expansions with IL-2, as described in mouse CAR-T cells. Partial least squares-discriminant analysis (PLS-DA) revealed a distinct metabolic profile between MCJ$^{low}$ (D16) and MCJ$^{high}$ (D26) human CD8 CAR-T cells (Fig. 6h). Similar to the metabolomic differences between MCJ KO and WT murine CD8 CAR-T cells, the level of ATP in MCJ$^{low}$ CAR-T cells was increased relative to MCJ$^{high}$ CD8 CAR-T cells (Fig. 6i and Supplementary Fig. 17a, b). There was also an enrichment of TCA metabolites (aKG/2-oxoglutarate, fumarate, malate) in MCJ$^{low}$ CD8 CAR-T cells (Fig. 6i and Supplementary Fig. 17a). Together, these data further support that low levels of MCJ in human CD8 CAR-T cells enhances mitochondrial metabolism, as seen in the murine model.

## MCJ deficiency improves human CD8 CD19-BBz CAR-T cell efficacy

To examine whether the role of MCJ as a negative regulator of mitochondrial respiration impacts the function of human CD8 CAR-T cells, we knocked down MCJ expression in CD8 cells isolated from healthy donor PBMC by transfection with an MCJ-specific siRNA (siMCJ). Analysis of MCJ protein by Western blot showed a substantial reduction of MCJ in human CD8 cells transfected with siMCJ compared with CD8 cells transfected with a control siRNA (c-siRNA) (Fig. 7a). To examine the effect of MCJ silencing on MMP in CD8 cells, after transfection with the siRNA, CD8 cells were activated with anti-CD3/anti-CD28 Abs for 2 days, expanded with IL-2, and after two expansions we performed TMRE staining. Higher MMP was detected in siMCJ-transfected human CD8 cells (Fig. 7b), demonstrating that silencing MCJ expression is a

feasible strategy to sustain MMP in human CD8 cells during in vivo expansion.

We also investigated whether reducing MCJ expression in human CD8 cells impacted IFNγ production. CD8 cells isolated from PBMC were transfected with c-siRNA or siMCJ, activated with anti-CD3/antti-CD28 Abs, washed, placed in medium alone, and human IFNγ in the supernatant was determined by ELISA. Higher IFNγ levels were produced by siMCJ-transfected CD8 cells compared with c-siRNA-transfected CD8 cells (Fig. 7c). Thus, the function of MCJ in human CD8 cells mimics its function in mouse CD8 cells.

We therefore developed an efficient strategy to reduce MCJ levels in human CD8 CAR-T cells with the goal of boosting mitochondrial respiration and effector function. We generated a unique lentiviral construct that includes the human CD19-BBz CAR as well as an shRNA for MCJ (shMCJ-1) under its own specific promoter (CD19-BBz/shMCJ CAR) (Fig. 7d). This strategy ensures that all CD8 cells expressing the CD19-BBz CAR will also express the shMCJ, attenuating MCJ expression specifically in CAR-T cells. As a control, we also generated a similar vector with a control (scramble) shRNA (c-shRNA). We tested this new CD19-BBz/shMCJ-1 CAR for its ability to reduce MCJ expression. CD8 cells were isolated from healthy donor PBMC and activated with anti-CD3/anti-CD28 for 48 h, followed by lentiviral transduction with CD19-BBz/shMCJ-1 CAR or CD19-BBz/c-shRNA CAR vectors. Transduced cells were then expanded with IL-2. Transduction efficiency was comparable for both CARs (Supplementary Fig. 18). After three expansions with IL-2, CD8 CAR-T cells were isolated to examine MCJ expression by Western blot analysis. MCJ levels were substantially reduced in CD19-BBz/shMCJ-1 CD8 CAR-T cells relative to CD19-BBz/c-shRNA CAR-T cells (Fig. 7e). We also examined whether this level of reduction of MCJ expression was sufficient to enhance mitochondrial respiration using the Mitostress assay in purified CAR-T cells. Basal mitochondrial respiration, spare respiratory capacity, maximal respiratory capacity and ATP production were higher in CD19-BBz/shMCJ-1 CD8 CAR-T cells relative to CD19-BBz/c-shRNA CAR-T cells (Fig. 7f and Supplementary Fig. 19a). Thus, consistent with mouse CD8 CAR-T cells, loss of MCJ increases mitochondrial respiration in human CD8 CAR-T cells.

To determine the efficacy of human CD19-BBz/shMCJ-1 CD8 CAR-T cells in killing human leukemia cells expressing CD19, we isolated CD8 cells from 2 healthy donors (H16 and H19) with high MCJ expression. CD8 cells were activated, transduced with the CD19-BBz/shMCJ-1 CAR or CD19-BBz/c-shRNA CAR, and expanded with IL-2. After the 3rd expansion, no obvious difference in the cell surface phenotype of the CD8 CAR-T cells was observed, with similar frequency of central memory (Tcm) and stem cell memory (Tscm) subsets between CD19-BBz/shMCJ CD8 CAR-T cells and CD19-BBz/c-shRNA CD8 CAR-T cells (Supplementary Fig. 20a, b). We next tested their killing activity. After the 3rd expansion, CD8 CAR-T cells were co-cultured with the human B-ALL cell line, Nalm6, and the survival of Nalm6 cells was examined 20 h later. CD19-BBz/shMCJ-1 CD8 CAR-T cells showed enhanced killing activity compared to CD19-BBz/c-shRNA CD8 CAR-T cells from donor H16 (Fig. 7g). Similar results were obtained with CD19-BBz/shMCJ-1 CD8 CAR-T cells from donor H19 (Fig. 7g). In addition, analysis of CD107a expression showed increased cytotoxic degranulation in CD19-BBz/shMCJ-1 CD8 CAR-T cells relative to CD19-BBz/c-shRNA CD8 CAR-T cells, after co-culture with Nalm6 leukemia cells (Fig. 7h).

To further demonstrate specific effect of silencing MCJ expression in human CAR-T cells on the efficacy of CAR-T cells, we generated another CD19-BBz/shMCJ construct with an independent shMCJ sequence (shMCJ-2). Similar to the CD19-BBz/shMCJ-1 CD8 CAR-T cells, MCJ levels were markedly reduced in CD19-BBz/shMCJ-2 CD8 CAR-T cells (Fig. 7i). We also examined the effect of MCJ knockdown in CD19-BBz/shMCJ-2 CD8 CAR-T cells on mitochondrial respiration. Basal mitochondrial respiration, maximal respiration, spare respiratory capacity and ATP production were markedly increased in CD19-BBz/shMCJ-2 CD8 CAR-T cells relative to CD19-BBz/c-shRNA CAR-T cells

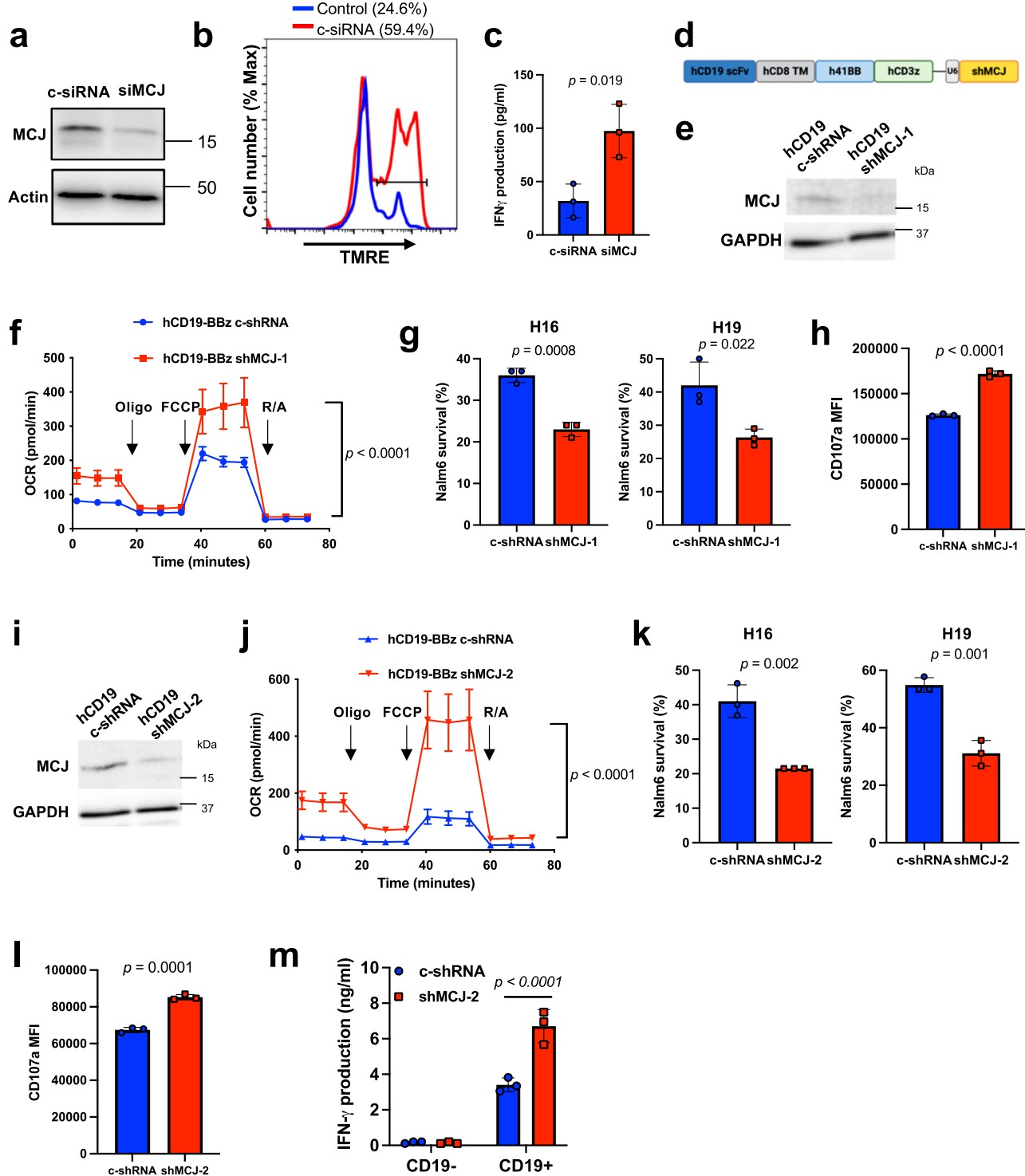

(Fig. 7j and Supplementary Fig. 19b). To test the killing activity of the CD19-BBz/shMCJ-2 CD8 CAR-T cells, CD8 cells from two donors were transduced with the CD19-BBz/shMCJ-2 CAR or CD19-BBz/c-shRNA CAR and after three expansions with IL-2, they were co-cultured with Nalm6 cells. CD19-BBz/shMCJ-2 CD8 CAR-T cells from both donors also showed increased killing activity compared to the control CD19-BBz/c-shRNA CD8 CAR-T cells (Fig. 7k). Furthermore, CD107a expression was higher in CD19-BBz/shMCJ-2 CD8 CAR-T cells (Fig. 7l). In addition, we measured the production of IFNγ by CD19-BBz/shMCJ-2 CD8 CAR-T cells during co-culture with Nalm6 cells, by ELISA. The levels of IFNγ were higher in CD19-BBz/shMCJ-2 CD8 CAR-T cells

relative to CD19-BBz/c-shRNA CD8 CAR-T cells (Fig. 7m). As a control, neither shMCJ-2 or shRNA control CD8 CAR-T cells produced IFNγ when CD19-negative Nalm6 cells were used for the co-culture (Fig. 7m). As an additional control for the specific effect of shMCJ in CAR-T cells, we found no difference in CD107a expression between shMCJ-2 CD8 CAR-T cells and c-shRNA CD8 CAR-T cells when we used CD8 cells from a MCJ$^{low}$ donor (Supplementary Fig. 21a, b).

Together, these results reveal a novel strategy to improve mitochondrial metabolism in human CD8 CAR-T cells by silencing MCJ, and they show that CD19-BBz/shMCJ CD8 CAR-T cells outperformed the standard CD8 CAR-T cells against human Nalm6 B-ALL cells.

**Fig. 7 | MCJ deficiency improves human CD19-BBz CAR-T cell efficacy. a** CD8 cells isolated from the PBMC of a healthy volunteer were transfected by nucleofection with a control siRNA or siMCJ and activated with coated anti-CD3/anti-CD28 for 2 days. MCJ levels were analyzed by Western blot analysis. **b** CD8 cells from a healthy donor were transfected with siMCJ or control siRNA as in (**a**), activated for 2 days, expanded with IL-2 for 2 days. MMP of the cells was examined by TMRE staining. The percentage of TMRE^high cells is shown in parenthesis. **c** CD8 cells from a healthy donor were transfected and activated as in (**a**), washed, and incubated in medium alone for 48 h. IFNγ levels in the supernatant was determined by ELISA (*n* = 3 biologically independent samples). **d** Scheme (created using BioRender.com) showing the human CD19-BBz/shMCJ CAR construct containing (a) the scFv part of human CD19 (hCD19 scFv), (b) the transmembrane domain (TM) of human CD8 (hCD8 TM), (c) the cytoplasmic costimulatory domain of human 4-1BB (h41BB), (d) the cytoplasmic domain of human CD3z (hCD3z) and the shRNA for human MCJ cassette (including the U6 promoter). As a control we used the same construct but with a scramble shRNA. **e** CD8 cells were isolated from a healthy donor, activated with anti-CD3/anti-CD28 Abs for 2 days, transduced with the lentiviral CD19-BBz/shMCJ-1 CAR or CD19-BBz/shRNA CAR construct, and expanded with IL-2 (100 IU/ml) for 3 expansions. MCJ expression in purified CD19-BBz/shMCJ-1 or CD19-BBz/c-shRNA CAR-T cells after 3 expansions, by Western blot analysis. **f** After 3 expansions with IL-2, purified CD19-BBz/shMCJ-1 or CD19-BBz/c-shRNA CD8 CAR-T cells were used for Seahorse MitoStress assay (*n* = 10 replicate well per group). **g** CD8 cells

isolated from two healthy donors (H16 and H19), were activated, transduced and expanded as (**e**). After the 3rd expansion with IL-2, CD19-BBz/shMCJ-1 or CD19-BBz/c-shRNA CAR-T cells were co-cultured with the human B cell leukemia Nalm6 cells (E:T = 0.125). After 20 h, live Nalm6 cells were counted (*n* = 3 biologically independent cells). **h** CD107a expression (MFI) on CD19-BBz/shMCJ or CD19-BBz/c-shRNA CAR-T cells after 4 h of co-culture with Nalm6 cells, as determined by flow cytometry (*n* = 3 biological independent cells). **i** Human CD8 cells were activated, transduced with lentiviral construct contained CD19-BBz/shMCJ-2 CAR or CD19-BBz/c-shRNA CAR and expanded with IL-2 for 3 expansions. MCJ expression in purified CAR-T cells, by Western blot analysis. **j** After 3 expansions with IL-2, purified human CD19-BBz/shMCJ-2 or CD19-BBz/c-shRNA CD8 CAR-T cells were used for Seahorse MitoStress assay (*n* = 10 replicate wells per group). **k** CD8 cells isolated from two healthy donors (H16 and H19), were activated, transduced and expanded as (**h**), and CD19-BBz/shMCJ-2 or CD19-BBz/c-shRNA CAR-T cells were co-cultured with the human B cell leukemia Nalm6 cells (E:T = 0.125). After 20 h, live Nalm6 cells were counted (*n* = 3 biologically independent cells). **l** CD107a expression (MFI) on CD19-BBz/shMCJ-2 or CD19-BBz/c-shRNA CAR-T cells after 4 h of co-culture with Nalm6 cells (*n* = 3 biologically independent cells). **m** IFNγ production of CD19-BBz/shMCJ-2 or CD19-BBz/c-shRNA CAR-T cells during co-culture with Nalm6 cells at E:T = 0.25, as determined by ELISA (*n* = 3 biologically independent samples). *p* was determined by two-sided unpaired *t* test (**c**, **g**, **h**, **k**, **l**) and by two-way ANOVA (**f**, **j**, **m**). Mean ± SD are shown for (**c**, **f**–**h**, **j**–**m**).

## Discussion

While CAR-T cell therapy has produced remarkable upfront responses in patients with relapsed/refractory B cell malignancies, the majority of patients will not achieve durable remissions[22]. Multiple mechanisms of CAR-T cell failure have been identified, including tumor-intrinsic mechanisms such as the escape of malignant cells which lose or down-regulate antigen expression and CAR T cell-intrinsic properties relating to the activity of the CAR product within the patient. Poor in vivo expansion of CAR-T cells has been linked to the inability to achieve upfront complete remissions across multiple diseases, while short persistence of CAR-T cells has been associated with increased risk of relapse in leukemia patients[7,18–20,23,97,98]. Such data highlight the clinical promise of improving CAR-T cell outcomes by improving the fitness of the CAR-T cell product. Manipulating the metabolism of the CAR-T cells has recently emerged as an appealing strategy to improve the function and survival of CAR-T cells in vivo. Several studies have revealed the metabolic state of CAR-T cells is critical in regulating their functionality and persistence[42–44]. There is a growing interest in boosting mitochondrial metabolism to enhance CAR-T cell efficacy, since it is well established that memory CD8 cells rely on mitochondrial respiration for survival as well as for their rapid recall function. We have previously identified MCJ/*DnaJC15* as an endogenous negative regulator of complex I of ETC and mitochondrial respiration[51,52]. We have shown that loss of MCJ increases mitochondrial respiration in mouse CD8 cells, and increases their memory function against influenza virus[51]. Here, we show for the first time the superior efficacy of MCJ-deficient CAR-T CD8 cells against B cell leukemia in vitro and in vivo, as well as the superior efficacy of adoptively transferred TCR-specific MCJ-deficient CD8 cells against a solid tumor, demonstrating the impact of MCJ on both CAR and TCR-specific T cell therapies. In addition, we have identified MCJ as a negative regulator of mitochondrial metabolism in human CD8 cells and as a novel target for enhancing the potency of CAR and other adoptive T cell therapies in patients.

Metabolism is emerging as a key area for improvement of CAR-T cell therapy, since the source of ATP production (i.e., glycolysis versus mitochondrial respiration, glucose versus fatty acids) during cell manufacturing and expansion can impact the in vivo function and persistence[42,43,99]. After activation of a naïve T cell population that predominantly uses fatty acid oxidation to obtain ATP, the metabolism reprograms to favor glycolysis as the primary method of ATP production during the rapid proliferation of T cells. Glycolysis triggered by IL-2 during the expansion of CAR-T cells is associated to

a more terminally differentiated phenotype of short-lived CAR-T cells, and reduced persistence, while mitochondrial respiration is associated with a less differentiated and more "stem-like" phenotype of CAR-T cells[35,66,100]. Recent studies have shown that inhibiting glycolysis during expansion can help to maintain an undifferentiated state of CAR-T cells and promote the memory potential of the product[45–47]. Approaches to inhibit glycolysis (e.g., Akt inhibition) are currently being tested[45,46], but limiting glycolysis has the potential to compromise the expansion of CAR-T cells as glycolysis not only produces ATP, but is essential for nucleotide synthesis for proliferation. As mentioned above, mitochondrial respiration is important for CAR-T cell function and survival, and impaired mitochondrial metabolism has been associated with poor efficacy of CAR-T cells[44]. Boosting mitochondrial respiration in CAR-T cells is not simple. Recently, overexpression of PGC1a was shown to increase mitochondrial biomass in CAR-T cells and subsequently to increase CAR-T cell control of tumors in a xenograft model of lung carcinoma[48]. However, increasing mitochondrial biomass is a metabolic costly process for CAR-T cells with the potential to compromise other functions. In addition, increasing mitochondrial biogenesis can lead to increased ROS levels which could be detrimental in a long term for CAR-T cells. Here, we propose a novel and feasible strategy to specifically increase mitochondrial respiration, ATP and NAD production without increasing mitochondrial mass by silencing the expression of MCJ in CD8 CAR-T cells. The absence of MCJ does not interfere with glycolysis or the proliferation of CD8 cells, and we have shown that increased mitochondrial ATP production in the absence of MCJ enhances CD8 cell secretion of cytokines and the release of cytotoxic granules by exocytosis, two ATP-dependent processes[101,102]. We have also shown that loss of MCJ increases Complex I activity without increasing ROS production because it promotes the formation of respiratory supercomplexes[51,52]. Thus, silencing MCJ in CAR-T cells is a promising strategy to specifically increase mitochondrial respiration and metabolic fitness of CAR T cells.

CAR-T cell manufacturing commonly involves ex vivo expansions with IL-2, which promotes rapid proliferation of T cells via upregulation of glycolysis[103,104]. However, prolonged expansion of CD8 cells with IL-2 often results in loss of T cell function and reduced persistence in vivo[58]. IL-2 has previously been shown to upregulate MCJ expression during the expansion of CD8 cells[59] and thus, could be a mechanism by which CD8 CAR-T cells lose function over time. In these studies, we show that expansion of CD8 CAR-T cells with IL-2 leads to loss of MMP and mitochondrial respiration. In addition, the in vivo efficacy of WT

CD8 CAR-T cells decreases between the first and the third expansion with IL-2. In contrast, MCJ-deficient CD8 CAR-T cells retain their efficacy even after three expansions in IL-2. While the use of alternative common-γ-chain cytokines can allow CD8 CAR-T cells to retain greater in vivo function[67], we found that MCJ-deficiency was able to improve the cytotoxic capacity of CAR-T cells manufactured with IL-7 and IL-15 (Supplementery Fig. 6d), suggesting that the benefits of MCJ-deletion extend beyond mitigating the impact of IL-2 during manufacturing.

Currently, commercial CAR-T cell therapy includes both CD4 and CD8 CAR-T cells, at both fixed and unfixed CD4:CD8 ratios. It is well-known that effector CD8 cells need the help of CD4 cells in vivo in immune responses to infectious diseases as well as in TCR-driven anti-tumor immune responses. Similar results have been found for CAR-T cell therapy based on preclinical studies, with CD4 CAR-T cells needed in addition to CD8 CAR-T cells for maximal response[29,31]; however, CD4 CAR-T cells have been implicated in driving cytokine release syndrome (CRS)[26–30]. We have recently shown that human CD4 cells, unlike mouse CD4 cells, produce high levels of IL-6 after activation[105]. In contrast to human CD4 cells, human CD8 cells do not produce IL-6 upon activation (Supplementary Fig. 22a). Interestingly, human CD4 cells were found to produce IL-6 during the expansion with IL-2 in the absence of TCR signals (Supplementary Fig. 22b). Thus, although it is generally believed that the high levels of IL-6 in patients treated with CAR-T cells comes from monocytes[26], activated CD4 CAR-T cells may represent another major source of IL-6. Since IL-6 is a major driver of CRS, reducing the need of CD4 cells for CD8 CAR-T cells efficacy could be potentially beneficial. Keeping a balance between efficacy and safety remains a major challenge for CAR-T cell therapy and other T cell adoptive therapies. In our study, we demonstrate that MCJ-deficient CD8 CAR-T cells without CD4 CAR-T cells are highly efficient in vivo against a B cell leukemia mouse model. In contrast, WT CD8 CAR-T cells alone were not able to control the disease, consistent with previous studies[29,31]. We have previously shown that memory WT CD8 cells are not protective against influenza virus in the absence of memory CD4 cells, but MCJ-KO CD8 cells alone were highly efficient in providing protection against lethal dosage of influenza virus[51]. Thus, enhancing mitochondrial metabolism by silencing MCJ expression in effector/memory CD8 cells reduces their need for CD4 cell help to achieve maximal activity. This could be a clinical strategy to improve CD8 CAR-T cell therapy efficacy with minimal need of CD4 cells, thereby minimizing the risk of severe CRS.

Overall, our study reveals MCJ as a novel target to enhance CD8 CAR-T cell efficacy by enhancing mitochondrial function. We also provide a strategy to attenuate MCJ expression CAR-T cells by incorporating shRNA targeting MCJ into the CAR vector. These data demonstrate the proof-of-principle that manipulating MCJ can also improve human CAR-T cell function. Such strategy has promising therapeutic potential to improve the efficacy of CAR-T cell therapies for B cell leukemias and lymphomas, as focused on in these pre-clinical studies, as well as in solid tumors in which the metabolic barriers of the tumor microenvironment present an obstacle to effective immune responses. Additionally, our studies using OT-I T cells demonstrate the generalizability of MCJ silencing or ablation to improve upon TCR-based adoptive T cell therapies as well, further broadening the potential benefit of this strategy to all T cell-based immunotherapies.

## Methods
### Mice
WT C57BL/6 J mice (RRID:IMSR_JAX:000664), OT-I transgenic mice (RRID:IMSR_JAX:003831) and CD4-Cre mice (RRID:IMSR_JAX:022071) were purchased from Jackson Laboratories. MCJ KO mice were previously described[52]. MCJ$^{f/f}$ mice were generated by CRISPR/Cas as described[106] using sgRNAs and Cas9 in C57Bl/6 single-cell embryos to insert loxP sites flanking the exon I of mouse MCJ/*DnaJC15*. One line of

MCJ$^{f/+}$ mice was identified, loxP sites insertion was verified and the mice were backcrossed with C57Bl/6 mice first and intercrossed to generate homozygous MCJ$^{f/f}$ mice. These mice were then crossed with CD4-Cre transgenic mice. All mice were co-housed in specific pathogen free animal facilities, maintained at 72$^{0}$F (+/− 2$^{0}$F), 35% humidity, and a 14 h light/10 h dark cycle (6 am-8 pm). All the mice were back-crossed in the C57Bl/6 J background for over 10–12 generations. Both male and female mice were used for the studies. The average age of the mice was between 12 to 24 weeks. $CO_2$ followed by cervical dislocation as an approved secondary method was used for euthanasia. Animal studies were approved by the Institutional Animal Care and Use Committee (IACUC) of the University of Vermont and University of Colorado, and performed following the guidelines.

### Cell lines
The murine B16-OVA melanoma cell line expressing ovalbumin[107] was cultured in Bruff's media (complete Click's (EHAA) medium plus β-mercaptoethanol). The B16-OVA cells were not passed more than 4 passages per experiment. The murine E2a pre-B ALL cell line (derived from E2a:PBX1 transgenic mice in the C57Bl/6 background)[108] was provided by Janetta Bijl (Université de Montréal, Montreal, Canada). As previously described, the E2a cells express pre-ALL markers, such as CD19[82]. The E2a cell lines were transduced with lentivirus encoding GFP, and a single cell clone was established by dilutional cloning to generate the E2a GFP+ cell line. The human Nalm6 pre-B ALL cell line expressing CD19 was initiated from an adolescent pre-B ALL patient[17].

### Purification and activation of CD8 cells
Murine CD8 cells were isolated from lymph nodes and spleen using negative selection with CD8α$^{+}$ T cell isolation kit (for generation of CAR-T cells) (Miltenyi, order# 130-104-075) or positive selection (for OT-I CD8 cells) with CD8α (Ly-2) Microbeads (Miltenyi, order# 130-117-044). Mouse CD8 cells were activated with plate-bound anti-CD3 (5 µg/ml) and soluble anti-CD28 mAbs (1 µg/ml) for 48 h for WT and cMCJ KO CD8 studies and OT-I CD8 cells studies. The detailed information of antibodies is listed in Supplementary Table 1.

Human CD8 cells were isolated: (1) from de-identified PBMC obtained from LRS (leukoreduction system) chambers (discard material) at the Colorado Children's Hospital Blood Bank (considered no human subjects), or (2) from PBMC obtained from healthy volunteers after signing an informed consent forms at the University of Vermont following the guidelines of the University of Vermont Institutional Review Board (IRB # 04-393). Positive selection using CD8 Microbeads, human (Miltenyi, order# 130-045-201), was utilized to isolate human CD8 cells. The human CD8 cells were activated with anti-CD3 (10 µg/ml) and anti-CD28 (4 µg/ml) Abs for 48 h, as previously described[105].

### Generation of mouse anti-CD19 CAR-T cells
The mouse CD19-BBz CAR retroviral construct expressing truncated hEGFR was kindly provided by Drs. Shivani Srivastava and Stanley Riddell[109]. Retroviral supernatant was produced in platinum-E (Plat-E) (ATCC) retroviral packaging cell line as previously described[82], and collected after 42 h post-transfection.

Mouse CD8 cells were isolated with negative selection as described above, activated with anti-CD3/anti-CD28 beads (Dynabeads™ Mouse T-Activator CD3/CD28 for T-Cell Expansion and Activation, Gibco, Cat#11453D) with addition of recombinant human IL-2 (rhIL-2) (60 IU/ml) and human recombinant IL-7 (10 ng/ml). On the second and third day of activation, the supernatant containing the CD19-BBz CAR/hEGFR retrovirus was spun on a plate coated with retronectin (Takara Bio Inc.) (2000 × *g* for 2.5 h at 32 °C). The activated cells were added to the viral-coated plate for transduction. The transduced CD8 CAR-T cells were removed from the beads on the fourth day and expanded with rhIL-2 (60 IU/ml) or rhIL-7 (10 ng/ml) and rhL-15 (100 ng/ml). After

2 days (1st expansion) cells were split 1/3 with fresh cytokine-containing medium. After 2 more days (2nd expansion), cells were split again with fresh cytokine-containing medium. Two days later (3rd expansion) cells were used for experiment or incubated in cytokine free-medium.

For CAR+ cells enrichment, the CAR-T cells expanded with IL-2 for 3 expansions were incubated with an anti-hEGFR-PE (BioLegend, Cat#352904) antibody, followed by incubation with anti-PE microbeads (Miltenyi, order#130-048-801), and purified using the Miltenyi LS columns as recommended by the manufacturer (Miltenyi). The enrichment resulted in a 98–100% CAR+ population, as verified by flow cytometry.

### Generation of human anti-CD19 CAR-T cells
The human CD19-BBz shRNA CAR lentiviral constructs were based on a previously described CD19-BBz CAR containing the human CD19-binding scFV FMC63, CD8 hinge domain, 4-1BB costimulatory domain and CD3ζ chain[110]. Using CD19-BBz CAR plasmid as a cloning vector, we generated multiple vectors where we incorporated the RNA polymerase III U6 promotor (on the 3′ of the CD3ζ chain domain) followed by an shRNA: (1) a CD19-BBz/shMCJ-1 CAR construct containing the shMCJ-1 5′-GAAGATTTCAACTCCTAGC-3′ sequence[11], (2) a CD19-BBz/shMCJ-2 CAR construct containing the shMCJ-2; 5′-AACCTCTAGAACAAGTTATC-3′, and (3) a CD19-BBz/c-shRNA CAR vectors expressing the shRNA encoding scramble sequences. Lentiviral supernatant was produced in the LentiX-293T packaging cell line (Clonetech) as previously described[112]. Lentiviral supernatants were collected after 48 h post-transfection.

Human CD8 cells were isolated with positive selection as described. The CD8 cells were activated with anti-CD3/anti-CD28 beads (Dynabeads™ Human T-Expander CD3/CD28, Gibco), as previously described[112]. After 48 h activation, the CD8 cells were spun (1000 × $g$ for 2 h at 32 °C) with lentiviral supernatant containing CD19-BBz/shMCJ-1 CAR, CD19-BBz/shMCJ-2 CAR or CD19-BBz/c-shRNA CAR construct-packing virus with rhIL-2 (40 IU/ml) and protamine sulfate. After transduction, the anti-CD3/anti-CD28 beads were removed and CD8 cells were expanded with rhIL-2 (100 IU/ml) for the specified number of days.

To enrich in CAR + CD8 cells post-transduction, CAR-T cells expanded with IL-2 for 3 expansions were stained with a CD19-Fc/AF647 conjugated peptide followed by anti-AF647 beads (Miltenyi) as recommended by the manufacturer (Miltenyi). The enrichment also resulted in a 98–100% CAR+ population, as determined by flow cytometry.

### In vivo B16-OVA tumor model
The B16-OVA cells were cultured in Bruff's media and washed with PBS before subcutaneous injection. $4 \times 10^5$ cells were injected into WT host mice. For the adoptive transfer OT-I experiment, the OT-I CD8 cells were isolated and activated as described above. The OT-I CD8 cells were then cultured with rhIL-2 (40 IU/ml) for 2 expansions. Cells were then washed with PBS and administered i.v. to B16-OVA-bearing mice after 10 days of tumor implantation. Tumor measurements were done using a caliper to calculate the tumor volumes using the formula Vol = length × width × height. The tumor size was followed over time. As a human endpoint, mice were euthanized when the tumor reached 20 mm in any one dimension, tumors were ulcerated or infected, or if there was a major sign of discomfort as determined by the institutional veterinarian. Mice were monitored every other day during the first week or until the tumor was palpated and daily after until the mice needed to be euthanized. Veterinary technicians in the institutional facility monitored mice daily.

The harvested tumors were homogenized through the mesh, and the red blood cells were removed through red blood cell lysis solution. The homogenized tumor cells were used for staining to identify infiltrated immune cells through flow cytometry.

To isolate the infiltrated CD8 cells from the tumor, CD8a (Ly-2) Microbeads, mouse (Miltneyi) was utilized. We followed the protocol recommended by the manufacturer for CD8 cell isolation. Briefly, the homogenized tumor cells were washed with MACS, and incubated with anti-CD8 beads at 4 °C for 15 min. The cells were washed and applied to the LS column and washed with 3 ml of MACS buffer for 3 times. CD8 cells were eluted from the column.

### In vivo mouse CAR-T cell model
The E2a cells ($10^6$) were cultured in CMM for no more than 2 passages before intravenous tail vail injection to B6 mice 3 days before CAR-T cell injection. The E2a-bearing mice were irradiated with 500 cGy sublethal dosage the day before CAR-T cell injection. The murine CD8 CAR-T cells were generated and expanded with rhIL-2 (60 IU/ml) as described above. The expanded CD8 CAR-T cells were washed with PBS, and $10^6$ CAR+ cells were injected into the E2a-bearing mice via intravenous tail vail injection. The mice were followed for survival with the humane end point of development of hind limb paralysis, or weight loss of more than 20% of the original weight, or signs of major discomfort as determined by the institutional veterinarian. Mice were monitored every other day during the first week and daily after until the mice needed to be euthanized. Veterinary technicians in the institutional facility monitored mice daily.

### Mitochondrial respiration and ATP production
Oxygen consumption rate (OCR) was analyzed using the Seahorse MitoStress Test Kit with XF96 Extracellular Flux analyzer (Seahorse Bioscience). The murine CD8 CAR-T cells were expanded in rhIL-2 (60 IU/ml) and were enriched utilized anti-PE beads (Miltenyi, order#130-048-801) by staining CAR+ cells with anti-hEGFR-PE antibody (BioLegend, Cat#352904). The human CD8 CAR-T cells were expanded in rhIL-2 (100 IU/ml) and were purified with hCD19-Fc/AF647 conjugated peptide followed by anti-AF647 beads as described above. The enriched CAR-T cells ($2 \times 10^5$) were then plated to a 96-well Seahorse plate coated with Cell-Tak (Corning) as previously described[51]. The plate was incubated at 37 °C with no $CO_2$ for 1 h before loading to the XF96 Extracellular Flux analyzer. OCR was measured at baseline and after the sequential injections of Oligomycin (1.5 μM), FCCP (2 μM) and Rotenone/antimycin A (0.5 μM) as recommended by the manufacturer. Real-time ATP production was analyzed using Seahorse XF Real-Time ATP Rate Assay Kit with XF96 Extracellular Flux analyzer (Seahorse Bioscience). The murine CD8 CAR-T cells were expanded in rhIL-2 and purified using anti-hEGFR-PE and anti-PE beads as mentioned above. The purified CAR-T cells ($2 \times 10^5$) were plated to a 96-well Seahorse plate coated with Cell-Tak (Corning) as described above. The plate was incubated at 37 °C with no $CO_2$ for 1 h before loading to the XF96 Extracellular Flux analyzer. OCR and ECAR were measured at baseline and after the sequential injections of Oligomycin (1.5 μM) and Rotenone/antimycin A (0.5 μM) as recommended by the manufacturer. The ATP production rate was calculated with the Wave software (Seahorse Bioscience). The equations used for calculation are provided in XF Real-Time ATP Rate Assay Kit User Guide.

### Multispectral Immunohistochemistry (IHC)
We performed multispectral IHC imaging using the Phenocycler Fusion instrument (formerly Vectra Polaris, Akoya Biosciences) by collaborating with the Human Immune Monitoring Shared Resource (HIMSR) at the University of Colorado School of Medicine. To quantify levels of infiltrated CD8 cells in the melanoma tumor, formalin-fixed paraffin-embedded tissue sections were stained with specific primary antibodies for mouse CD8 (Cell Signaling Technology, Cat#98941). The detailed information of antibodies is listed in Supplementary Table 1. The slides were stained with primary antibodies followed by HRP-conjugated secondary antibody, and HRP-reactive OPAL fluorescent reagents, as previously described[113].

20x objective was used for visualization of the images using the inForm software V2.6 (Akoya Biosciences).

## Cytokine production analysis

The murine IFNγ levels in culture supernatants were examined by ELISA as previously described[114] using anti-mouse IFNγ capture and biotinylated Abs (BioLegend). The detailed information of antibodies is listed in Supplementary Table 1. The human IFNγ level in culture supernatants was examined by ELISA using the Human IFNγ ELISA MAX Standard Set (BioLegend, Cat# 430102) followed by the instructions provided by the manufacturers.

## Flow cytometry analysis

For mitochondrial membrane potential (MMP) analysis, the assay was performed by staining CD8 CAR-T cells with TMRE as previously described[52]. For mitochondrial mass analysis, the CD8 CAR-T cells were stained with Mitotracker 633 (Thermo Fisher Scientific) for 30 min at 37 °C. For mitochondrial reactive oxygen species (ROS) analysis, the assay was performed by staining CD8 CAR-T cells with MitoSOX Red (Thermo Fisher Scientific, Cat#M36008) for 10 min at 37 °C. The anti-hEGFR (BioLegend, Cat#352904) or anti-G4S staining antibody (Cell Signaling Technology, Cat#38907 S) was added at the last 5–10 min of the incubation with the corresponding dye to identify the CAR+ population. The cells were washed and examined on the flow cytometer (Northern Lights and Aurora, Cytek Boisciences; LSRFortessa, BD Biosciences). We gated on the live cell population based on the FSC and SSC for these analyses.

The surface markers of the murine cells were examined through flow cytometry by staining the cells with antibodies recognizing G4S (Cell Signaling Technology), CD45 (BioLegend), CD8 (BioLegend, BD Biosciences), CD4 (BioLegend, BD Biosciences), CD11b (BioLegend), B220 (BioLegend), CD44 (BioLegend), CD25 (BioLegend), CD69 (BioLegend), CD62L (BioLegend), PD-1 (BioLegend), Fas (BioLegend), Tim3 (BioLegend) and TCR Vα2+ (BD Bioscience).

The intracellular markers of the cells were examined through flow cytometry by staining the cells with antibodies recognized T-bet (BioLegend), Tcf1 (Cell Signaling Technology), Foxo1 (Cell signaling Technology), and TOX (Invitrogen) using BD Pharmingen™ Transcription factor buffer set (Cat#562574). The intracellular staining was done following the recommended procedure by the manufacturer.

The surface markers of the human cells were examined by staining cells with antibodies against CD8a (BioLegend), CD62L (BD Biosciences) and CD45RA (BD Biosciences). The human CD19-BBz CAR-T cells were identified using CD19-Fc AF647 conjugated peptide, and analyzed through flow cytometer.

The sorting of different subsets of human CD8 cells was conducted by staining the PBMCs from the healthy donor with antibodies against CD8 (BioLegend), CD4 (BioLegend), CD45RA (BioLegend), CD45RO (BioLegend), and CCR7 (BioLegend). Tem cells were sorted as CCR7-/CD45RA-/CD45RO+, Temra cells were sorted as CCR7-/CD45RA+/CD45RO-, naïve T cells were identified as CCR7+/CD45RA+/CD45RO-.

The gating strategies of flow plots are shown in Supplementary Fig. 23. The detailed information of antibodies is listed in Supplementary Table 1.

## CD107a degranulation assay

The E2a GFP+ or Nalm6 GFP+ cells were co-cultured with CD8 CAR-T cells for 4 h with the presence of monensin (Invitrogen, Ref# 00−4505) and fluorescent anti-CD107a antibody (BioLegend). The CD8 CAR-T cells were identified with an anti-CD8 antibody (BioLegend). The CD107a expression on CD8 CAR-T cells was examined by flow cytometry. The detailed information of antibodies is listed in Supplementary Table 1.

## In vitro killing assay

The B16-OVA cells were plated in 24-well plate at $10^5$ cells and grown for 2 days before adding the IL-2 expanded OT-I cells for co-culture. The presence of B16-OVA cells was examined by counting. The anti-IFNγ antibody (BioLegend, Cat#505812) (5 μg/ml) were used to block IFNγ during the co-culture. The concanamycin A (CMA) was used as the pretreatment to OT-I cells expanded in IL-2 for 2 expansions. OT-I cells were treated with CMA (Santa Cruz Biotechnology, Cat# sc-202111) in 100 nM for 2 h, washed, and added to the B16-OVA cells for co-culture.

The E2a GFP+ cells were co-cultured with IL-2-expanded murine CD8 CAR-T cells for 15 h with the presence of flow counting beads (Accucheck Counting beads, Invitrogen, Cat# PCB100). The CD8 CAR-T cells were identified with anti-CD8 staining antibody (BioLegend, Cat#100725, RRID: AB_493425). The presence of E2a cells was examined by flow cytometry. The cell count was normalized to the number of beads within each sample. The percentage of E2a killing is normalized to the E2a cell count in the well without CAR-T cells. The E2a GFP+ cells were co-cultured with murine medium-resting IL-2-expanded CD8 CAR-T cells for 5 h. The presence of E2a cells was examined by counting. The blocking anti-IFNγ antibody (5 μg/ml) was utilized to block IFNγ during the co-culture. EGTA (3 mM) and MgCl₂ (2 mM) (Sigma) were used to block cytotoxicity during the co-culture. CMA (100 nM) was used to do pre-treatment with CAR-T cells for 2 h. The CAR-T cells treated with CMA were washed and plated to co-culture with E2a cells. Oligomycin (Torcis Bioscience, Cat# 4110) (5 μM) was used as pre-treatment to inhibit ATP synthase in the CAR-T cells. After 3 h of oligomycin treatment, the CAR-T cells were washed and plated to co-culture with E2a cells. The detailed information of antibodies is listed in Supplementary Table 1.

The Nalm6 cells were co-cultured with IL-2-expanded human CD8 CAR-T cells for 20 h. The presence of Nalm6 cells was examined by counting.

## Bulk RNAseq

Total RNA is isolated using RNeasy Micro Kit (Qiagen, Cat# 74004). The genomic DNA was removed using DNase I treatment as recommended by the manufacturer (Qiagen). RNA purity, quantity and integrity were determined with NanoDrop (ThermoFisher Scientific) and TapeStation 4200 (Agilent, CA, USA) analysis prior to RNAseq library preparation. The Tecan Universal Plus mRNA-Seq Library Kit was used (Tecan) with an input of 200 ng of total RNA to generate RNA-Seq libraries. Paired-end sequencing reads of 150nt was generated on NovaSeq X (illumina, Inc., CA, USA) sequencer at a target depth of 80 million paired-end reads per sample. Raw sequencing reads were de-multiplexed using bcl2fastq.

Quality of reads was assessed pre- and post-trimming using FastQC v0.11.9. Illumina universal adapters were removed, bases were trimmed if the Phred Score was less than 20, and any reads after trimming that were fewer than 10 base pairs in length were discarded using Cutadapt v4.4 under Python version 3.10. Reads were aligned and quantified to the mm10 reference genome using STAR v2.7.10b. The quality of the alignments was assessed using PicardTools CollectRnaSeqMetrics v2.27.5-13-g04e9b2, and the summarization of log outputs generated by STAR. For differential expression analysis, any genes with less than 10 counts total summed across all 6 samples were removed from the analysis. Counts were normalized using DESeq2 v1.36.0 and 2 RUVs were added as covariates to the generalized linear model using RUVSeq v1.30.0. GSEA was performed using the R package fgsea v1.22.0 and ranks were based upon the Wald statistic extracted from the differential expression matrix generated by DESeq2. The Hallmark gene set from the Molecular Signatures Database (MSigDB) was used for enrichment analysis and pathways were considered statistically significant if they had an adjusted $p < 0.05$. Heatmaps were generated using pheatmap v1.0.12 and visualized

based upon the covariate corrected counts matrix generated from limma v3.52.2 and row-normalized by z-score.

### Real-time RT-PCR analysis

Total RNA is isolated using RNeasy Micro Kit (Qiagen, Cat# 74004) or RNAqueous Phenol-free total RNA isolation (Invitrogen, Cat# AM1912) followed by the recommended protocol provided from the manufacturers. cDNA was synthesized using M-MLV Reverse Transcriptase (Invitrogen) as previously described[51]. Level of human MCJ (*DnaJC15*) mRNA was determined by real-time RT-PCR using the designated primer/probes previously described[111]: probe, 5′-CCTTGCCAGCAGATGG GCTTACACCTAAA-3′; sense primer, 5′-CAGAAAATGAGTAGGCGAGAA GC-3′; and antisense primer, 5′-TGAC TCTCCTATGAGCTGTTCTAATC-3′. Values of gene of interest were normalized to β2-microglobulin (Thermo Fisher, Hs00187842_m1) or HPRT (Thermo Fisher; Hs02800695_m1) by the comparative delta CT method.

### Analysis of MCJ/DnaJC15 in human CD8 cells using DICE database

The human MCJ expression data analyzed in this study were obtained from the published database, DICE (database of immune cell expression, expression quantitative trait loci [eQTLs], and epigenomics)[96].

### Western blot analysis

Whole-cell extracts and specific antibodies were used to perform western blot analysis, as previously described[105], using anti-human MCJ A12[111], anti-mouse MCJ Abs[52], anti-GAPDH (Cell Signaling Technology) and anti-β-actin (Cell Signaling Technology). The detailed information of antibodies is listed in Supplementary Table 1.

### Mass spectrometry metabolomics analysis

CAR-T cells were isolated as described above either from in vitro culture or from bone marrow harvested in an in vivo study. The cells were washed in PBS and frozen at −80 °C until the assay is ready to run. Metabolites from cells were extracted at $2 \times 10^6$ cells/ml at 4 °C (30 min) in the presence of 5:3:2 MeOH:MeCN:water (v/v/v). The samples were spun down and the resulting supernatant was transferred to new tubes and dried under a vacuum. The resulting residue was reconstituted in 0.1% formic acid at a 3x concentration, then analyzed on a Thermo Vanquish UHPLC coupled to a Thermo Q Exactive MS as previously described in detail[115].

### Statistics and reproducibility

The statistical analysis used GraphPad Prism v. 8.0. The standard student's t-test was performed between 2 groups. The 2-way ANOVA analysis was performed for more than 2 groups. Kaplan-Meier survival curve was analyzed with a log-rank (Mantel−Cox) test. Metabolic pathway enrichment analysis, partial least squares-discriminant analysis, heat maps and hierarchical clustering were performed using the MetaboAnalyst 4.0 and 6.0 package. The Standard $p < 0.05$ was served as the cut-off for significance. To ensure the reproducibility, the experiments were repeated independently at different times, different mice, different healthy donors, different preparation of cells, and even some cases the experiments were repeated at different institutions. In vivo experiments were repeated at least twice, in vitro experiments were repeated 2–3 times.

### Reporting summary

Further information on research design is available in the Nature Portfolio Reporting Summary linked to this article.

## Data availability

The data generated in this study are available within the article and its supplementary data files. The results from the MCJ/*DnaJC15* expression in human CD8 cells using DICE database are publicly available at https://dice-database.org/genes/DNAJC15. The bulk RNAseq data are available from the GEO database under accession number GSE263259. The results from the metabolomics analyses are available at the NIH Common Fund's National Metabolomics Data Repository (NMDR) website, the Metabolomics Workbench, under project number PR001970 [https://doi.org/10.21228/M8314P]. Source data are provided with this paper.

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

## Acknowledgements

We thank Dr. Karen Fortner (who deceased from cancer during the course of these studies) for her contributions to TCR-specific CD8 cell experiments. We thank Dr. Shivani Srivastava and Dr. Stanley Riddell for providing the murine CD19-BBz CAR construct. We thank Dr. Tinalyn Kupfer for help with flow cytometry analysis from CU AMC ImmunoMicro Flow Cytometry Shared Resource (RRID:SCR_021321) in the Department of Immunology and Microbiology at the University of Colorado Anschutz Medical Campus. We thank Dr. Julie Haines for help with metabolomics analysis from Mass Spectrometry Metabolomics Shared Resource Facility (RRID: SCR_021988) in support by the Cancer Center Support Grant P30CA046934 at the University of Colorado Anschutz Medical Campus. We thank the Human Immune Monitoring Shared Resource (RRID:SCR_021985) within the University of Colorado Human Immunology and Immunotherapy Initiative and the University of Colorado Cancer Center (P30CA046934) for their expert assistance in the analysis of immunofluorescent staining of solid tumor. We thank Colin Larson for his expertise in RNA sequencing from Genomics Shared Resource (RRID: SCR_021984) in support by the Cancer Center Support Grant P30CA046934 at the University of Colorado Anschutz Medical Campus. This work was supported by NIH R01 CA260909 (M.R.), R21 CA223389 (M.R.), R21 AI149187 (M.R.), R56 AI148434 (M.R.), K12 CA086913 (M.K.) and Hyundai Hope on Wheels Young Investigator Grant, 634956 (M.K.). This study was also partly supported by the National Institutes of Health P30CA046934 by utilizing the Bioinformatics (RRID: SCR_021983) and Biostatistics (RRID: SCR_021981) Shared Resource. The metabolomics analysis was supported by NIH grant U2C-DK119886 and OT2-OD030544 grants.

## Author contributions

M.W., M.E.K. and M.R. designed the project. M.W., E.L.G., F.V.P., C.P., A.J.N. and M.C.Y. performed experiments. M.E.K. and M.C.Y. designed the shRNA CAR constructs. J.H. and R.A.F. generated T-cMCJ KO mouse model. F.C. and A.D. performed metabolomics experiments and analyses. T.M.B. and S.B.T. performed RNAseq analyses. M.W., F.V.P., E.L.G., R.A.F, M.E.K. and M.R. edited the manuscripts. M.W., M.E.K. and M.R. wrote the manuscript.

## Competing interests

M.R. declares to be a member of the scientific advisory board and co-founder of Mitotherapeutix, and is co-inventor in patents held by the University of Vermont. M.R., M.E.K., M.W. and M.C.Y. declare the filing of a patent that discloses findings described in this manuscript. R.A.F. is an advisor to Glaxo Smith Kline, Celsius, EvolveImmune, Ventus Therapeutics, and is the recipient of a grant from Genetech/Roche. The remaining authors declare no competing interests.

## Additional information

[1]Department of Immunology and Microbiology, University of Colorado, Anschutz Medical Campus, Aurora, CO, USA. [2]Department of Biochemistry and Molecular Genetics, University of Colorado Anschutz Medical Campus, Aurora, CO, USA. [3]Division of Immunobiology, Department of Medicine, Larner College of Medicine, University of Vermont, Burlington, VT, USA. [4]Department of Pediatric Hematology, Oncology and Bone Marrow Transplant, University of

Colorado, Anschutz Medical Campus, Aurora, CO, USA. [5]Department of Pathology and Laboratory Medicine, University of Pennsylvania, Philadelphia, PA, USA. [6]Institute for Immunology, Perelman School of Medicine, University of Pennsylvania, Philadelphia, PA, USA. [7]Division of Transplant Immunology, Department of Pathology and Laboratory Medicine, Children's Hospital of Philadelphia, University of Pennsylvania, Philadelphia, PA, USA. [8]Department of Immunobiology, School of Medicine, Yale University, New Haven, CT, USA. [9]Howard Hughes Medical Institute, Yale University School of Medicine, New Haven, CT, USA. [10]Center for Cancer and Blood Disorders, Children's Hospital Colorado, Aurora, CO, USA. ✉e-mail: Mark.Kohler@cuanschutz.edu; Mercedes.Rincon@cuanschutz.edu

