## [Peer Review File · Nature Communications]

Deleting the mitochondrial respiration negative regulator MCJ enhances the efficacy of CD8⁺ T cell- adaptive therapies in pre-clinical studiesREVIEWER COMMENTS

Reviewer #1 (expert in CAR T cells and haematological malignancies):

In this manuscript, Wu et al explore the consequence of MCJ/DnaJC15 ko / downregulation on CD8+ T cell effector function.

Notably, MCJ is naturally down-regulated in CD4+ T cells.

The authors first show that MJC KO mice are relatively resistant to B16-OVA melanoma cells compared with WT mice. MCJ T cell selective KO mice appeared immunologically normal. CD8+ T cells from these mice differed only with increased IFN-g following mitogenic stimulation from controls. Similar to the MJC KO mice, resistance to B16-OVA melanoma engraftment was observed.

Using TCR transgenic MCJ CD8+ T cells, the authors make a key observation: namely in vitro effector function is only enhanced compared with control following prolonged culture in IL2. This finding is recapitulated in an in vivo model of CAR activity using CD8+ mouse CD19 CAR T cells. Benefit of MCJ KO was lost when T cells were not exposed to prolonged culture in IL2.

Enhanced effector function is reduced by either blocking IFN-g or following concanamycin A exposure.

Some of the in vitro findings were reproduced in human CD8+ CD19 CAR T cells using siRNA for MCJ silencing.

GENERAL COMMENTS

The authors characterize a clear difference in CD8+ T cell function in mouse and human T cells following MCJ ko with detailed and well performed experiments. They then propose that MCJ silencing is a strategy to improve CAR T cell function. However caution must be exercised with this assertion:

1. On a practical level, current CAR T cell manufacture rarely uses prolonged culture in IL2. Most manufacturing processes have moved to use IL7/IL15 cultures with as brief an expansion period as possible. In fact, cutting edge manufacturing processes lack expansion completely. Demonstrating that MCJ KO/silencing has benefit in CAR T cells generated using IL7/IL17 would strengthen the proposal regarding therapeutic utility of MCJ KO/silencing.

2. The authors focus entirely on CD8+ T cells, which makes sense given that MCJ is more important in CD8+ T cell biology than CD4+ T cell biology. In the discussion, the authors assert that CD8+ CAR T cell therapy may have advantages over CD4/CD8 CAR T cell therapy since toxicity would be lower due to absence of IL6 secretion from CD4+ T cells. Caution must be exercised with this assertion: Adoptive immunotherapy with only CD8+ T cells has largely failed and mixed CD4/CD8 therapeutics are preferred; While IL6 is an important part of the biology of CAR T immunotoxicity, the main source of IL6 is macrophages in response to T cell secreted IFN-g. Demonstrating utility of MCJ silencing in adoptive immunotherapy models using both CD4/CD8 would strengthen the proposal regarding therapeutic utility of MCJ ko/silencing.

3. Continuing with this line of thought, if MCJ ko/silencing CAR T cells secrete more IFN-g, they may cause increased toxicity. This aspect should also be considered.

4. While the metabolic "re-wiring" and consequences on effector function caused by MCJ/silencing is well shown, there is not a clear explanation of key biologic observation in terms of broader T cell biology. More specifically, what is the link between observed functional benefit and IL2 exposure. In the discussion, the authors compare this strategy to that of pharmacologic AKT inhibition during CAR T cell manufacture. In that approach, CAR T cell stemness, Tn and Tscm subtypes are preserved along with polyfunctionality. Understanding the consequence of MCJ ko/silencing in a

similar fashion would be helpful in understanding therapeutic application.

I recommend providing some experimental detail addressing these points and/or being more guarded regards potential therapeutic utility.

5. It may be interesting to see if pharmacologic downregulation of MCJ using Sinapine has similar effects to genetic KO/silencing.

6. What are the implications of reduced Caspase 3 activity associated with MCJ ko/silencing in these experiments?

SPECIFIC COMMENTS

7. Regarding supplementary figure 1: Tumour size and survival is shown in the main figure of ko mice, but only survival was shown in the supplementary figure of T-cMCJ KO mice. Since there was a difference in survival, this must have been due to tumour size - please show this.

8. It would be helpful if histograms with superimposed scatter of individual data points was used.

9. In suppl Fig 3(b), could most of the difference in tumour size be ascribed to two mice in QT OT-1?

Reviewer #2 (expert in adoptive cell therapies and CAR T cells):

The manuscript by Wu and colleagues describes how MCJ/DnaJC15, a known regulator of metabolism, and of mitochondrial complex I can be targeted to increase anti-tumour efficacy of CD8 T cells. The CD8 T cells could be modified both with tumour-specific CARs and TCRs. The authors demonstrate reduced growth of B16-OVA tumours in OT-I MJC knock-out mice versus WT mice and also enhanced in vitro killing of B16-OVA tumours by OT-I cells from the KO mice. The enhanced in vitro killing is abrogated when neutralizing antibody against IFN-g is added or Concanamycin A (CMA) which blocks perforin-mediated killing.

When WT mice transplanted with B16-OVA tumours were treated with in vitro expanded OT-1 cells from either WT or MJC KO mice, the MJC KO OT-I showed improved tumour infiltration and controlled tumour growth more efficiently. No difference in cell surface markers like CD44 or PD-1 were detected between the OT-I cells from MJC KO versus WT mice, but cells from MJC KO mice produced higher levels of IFN-g. The same findings were shown in a CD19 CAR murine leukemia model comparing CAR CD8 T cells from WT versus MJC KO mice.

CD19CAR CD8 T cells from MJC KO mice killed leukemia cells more efficiently, produced higher levels of IFN-g even after withdrawal of IL-2 used for expansion, but seemed not to reply on IFN-g for killing. The enhanced killing in MJC KO T cells was however inhibited by the addition of CMA or EGTA, blocking pathways required for perforin-based cytotoxicity.

CD19CAR CD8 T cells from MJC KO and WT mice showed different metabolism, with the MJC KO cells having higher mitochondrial respiration. Finally, the authors also investigated the role of MCJ in human primary CD8 T cells, including CD19CAR CD8 T cells. The lentiviral CAR construct also contained either shRNA MCJ to knock down MCJ in CAR T cells or control shRNA.

In summary, the study is timely as several reports of the importance of CAR-T metabolism for the persistence and long-term effect in vivo have been published. The manuscript provides ample evidence for the improved anti-tumour efficacy of MCJ negative T cells.

The authors did not report any differences in phenotype between the MCJ knock-out or knock-down T cells and the WT T cells. As memory CD8 T cells rely on mitochondrial respiration for survival and efficacy, it would have been interesting, however, to test additional memory T cells markers as well as comparing MCJ function in IL-2 expanded T cells with IL-7/IL-15 expanded T cells as the latter is thought to induce more central memory T cells. Did the MCJ negative CAR/TCR transgenic cells show any differences in stem-cell memory markers such as the transcription factor TCF-1?

Whether removing MCJ regulation of mitochondrial respiration omits the need for CD4 T cells in the

formation of memory and long-term efficacy of CD8 CAR T cells may be too early to say as this may not be sufficiently demonstrated in the CAR-T models used. It would, however, be interesting to perform repeated in vitro re-challenges of the CAR T cells with leukemia cell lines to see if the MCJ negative CAR T cells maintain their function over time.

Specific comments:

1. In the abstract the authors claim that they here identify the protein as a target to enhance mitochondrial respiration. However, they do not cite e.g., the following paper until the discussion: Champagne et al., *Immunity* 2016 (PMID: 27234056). Please modify statement in the abstract as the authors already identified MCJ as a target in previous publications. The introduction to adoptive T cell therapies also needs some updated references for TCR- and CAR-based T-cell therapy. Some of the later clinical trials should also be cited and a recent overview of approved CAR-T therapy (PMID: 37256141).

2. Figure 1c: update to show how many days n= for expansion

3. Figure 4. Fig. 4b: CD19 CD8 CAR-T cells isolated from the BM of injected mice 30 days later and used for ex vivo killing assays of E2a leukemia cells. What were E:T ratios, % of CAR T cells? Please show E:T ratios and non-normalised values. Was tumour load measured at all? If the tumour cells already contained GFP, a construct containing luciferase and the use of IVIS could easily have been adapted for monitoring of tumour load.

4. Figure 5: The Seahorse MitoStress assay (Figure 5e-g) showed higher maximal respiration and spare respiratory capacity in MJC KO cells. How many parallels were performed? Please provide some more detail of the number of replicates and use standard deviations rather than SEM in graphs f-g.

The authors also performed metabolic profiling by mass spec, which strengthen the data.

5. Figure 6: For Western blots, uncropped versions of gels/blots presented in the figure should be shown in the supplementary data or source data file. The results showed very heterogenous expression in CD8 T cells from healthy donors (Figure 6d) which had no correlation with age or sex. Did the authors look at whether it correlated to any degree with phenotype/activation status/differentiation of the CD8 T cells? How important was the difference in MCJ expression in the two healthy donors relative to the difference in WT and MCJ KO murine cells?

6. In Figure 7, how many donors were tested for the siRNA MCJ Knock-down experiments? Did the donors have high or low MCJ expression to start with? In Figure 7c, CD8 T cells were stimulated with aCD3/aCD28 to measure IFN-g levels. The siMCJ cells produced around 2-fold the amount of IFN-g. aCD3/aCD28 stimulation is very strong. Could this difference in IFN-g production also be seen if antigen-specific stimulation was used, studying human CAR T cells as in Fig. 6? When human CD19 CAR CD8 T cells were produced, the lentiviral CAR construct also contained either shRNA MCJ to knock down MCJ in CAR T cells or control shRNA (Fig. 7 d-e). The level of MCJ did not look very high in the control, but was clearly reduced by the shRNA MCJ. The CAR expression levels shown in Extended Fig 9 were not very high. The killing assays performed by flow cytometry should be explained in more detail. The E:T ratio is mentioned in the figure legend and is low (e.g. 0.125). After 20h, the remaining leukemia cells which express GFP are counted by flow cytometry. Please explained how this was done to accurately quantify the cancer cells. Were flow count beads used? Example FACS plots need to be shown in extended data. SEM are shown for all figures, please show standard deviations instead.

7. What were the humane endpoints of in vivo experiments? Please clarify.

8. Please provide additional details for antibodies used in IHC staining in Fig.2b

Reviewer #3 (expert in immunometabolism):

Meng-Han Wu et al target MCJ/DnaJC15 to increase CAR T cell persistence and function. They found a valid pathway linking bioenergetics with anti tumor activity.

The application of this pathway to CAR T cell therapy is novel and compelling and this manuscript in this regard would provide substantial information for future analyses for both basic and translational research in the fields of immunology/ immunometabolism and cancer biology.

Part of the authors' conclusions are not fully supported by the data specifically regarding direct

linking of MCJ to respiration and respiration to antitumor cytotoxicity of CAR CD8 T cells. This aspect would require revision. Another additional evidence that is needed in my opinion is a full characterization of the metabolic phenotype of these cells and their metabolic requirements for in vivo activity and persistence. This is lacking at the moment in the manuscript and I believe it is fundamental especially as a future source of interrogation for the community of what metabolic pathways and aspects characterize a fully functional tumor killing CAR T cell in vitro and in vivo.

Major:

1. Metabolic re-programming of CD8 cells should be better explained and referenced. Authors hint to a complete shift towards glycolysis when it is instead the case of a global increase in basal metabolism during activation, where both glycolysis and OXPHOS play a role.
2. Only in the discussion authors talk about the metabolic limitations of the CAR T persistence in vivo. Authors should provide more examples starting from the introduction and talk more also about what are the current available data that show that CAR T (specifically CD8) with better bioenergetics (and less glycolytic phenotype) live longer in the host with antitumor capacities. On the same note, authors should discuss better the attempts regarding PGC1a and based on what data this has been speculated as an energetically high cost process that cannot be further pursued.
3. It would be very important that authors explain better the molecular mechanisms by which MCJ is an endogenous negative regulator of Complex I and how it inhibits supercomplex assembly. Moreover, authors should talk about other reported or possible functions of MCJ.
4. Do MCJ-deficient mouse CD8 cells have higher CI activity? What about the activity of the other ETC complexes?
5. Are there differences in CD4 T cells or ILCs activation in T-cMCJ KO?
6. The conclusion of a superior killing linked to ATP represents a bit of a stretch if authors are trying to link killing to OXPHOS. Authors don't really prove here that ATP is important in this process in this model. What are ATP levels in these cells? Can authors interfere with ATP specifically to modulate this phenotype?
7. Authors should show better how the withdrawal of IL-2 affects T cell metabolism specifically in WT vs MCJ deficiency. Moreover, it would be important that authors describe better the molecular mechanisms behind IL-2-mediated upregulation of MCJ and how mechanistically IL2 promotes glycolysis while compromising mitochondrial respiration in T cells.
8. Regarding Fig 3f, authors should speculate on the fact that the anti-IFN γ antibody doesn't work for E2a compared to the B16 tumor authors used previously in the results.
9. Please specify if the leukemia tumor also grows slower in the KO and conditional KO compared to WT.
10. Authors should draw more conclusions from Fig 5h. What metabolic pathways are affected? For Fig 5i and 7a-b, are these only the metabolites that change significantly? Some short chain carnitines seem to be accumulating in the KO. What are the causes-repercussions on fatty acid metabolism in these cells and is this an aspect that needs to be considered for increased anti-tumor activity? Regarding this point the manuscript sorts of lack a complete metabolic characterization of these cells. Can authors use an antibody against MCJ or if available, small molecule inhibitors and show higher OCR in WT? Otherwise can supercomplexes assembly or CI activity be blocked in the KO and see consequences in OCR and their metabolic flexibility? A full analysis on the complexes and their activity in these cells and also addressing fuel preferences is mandatory. Authors should manipulate fuels or complexes and see what MCJ deficient cells are relying on. A small in vivo piece of data would also be beneficial: what metabolic environment are these cells finding in vivo that the absence of MCJ would give an advantage? (Lack of glucose/ fatty acids types-levels, etc..?).
11. It would be important to comment/ provide information regarding transcriptional changes in the WT vs KO. Maybe one round of IL2 expansion may be sufficient to see some differences; although authors provide surface markers that do not change besides IFN γ , it may be interesting to look at genes/senescence, general fitness etc.. in these cells at the level of transcription, possibly investigating gene sets downstream of major transcription factors (es PPAR γ etc.).
12. Authors mention that ROS do not change but they do not show ROS after all the IL2 passages and in the experiments ex vivo from the tumor.
13. Regarding Fig 7c, have authors tried this in IL2 expansion?

14. It is not clear if (and where) authors show (western blot) if CD8 T cells maintain MCJ expression during the ex vivo expansion in mouse vs human.
15. It would be interesting to show that the efficacy of human CD19-BBz/shMCJ-1 CD8 CAR-T cells in killing human leukemia cells expressing CD19 is poor or absent in a donor that has low level of MCJ.
16. It would be very important to show that MCJ is directly regulating the respiration which in turn is the solely responsible for antitumor activities and not via activation of transcriptional /translation/ signaling pathways (this goes back to point n10 and 11). This point is raised since MCJ has been link to c-Jun pathway. There are compounds that promote supercomplex assembly (3,4-methylenedioxy- β -nitrostyrene (MNS) and SYK inhibitors) that authors could use to show that in the WT they ameliorate CAR T function at least in vitro, and maybe they have small or no effect in the KO. The idea behind this is forcing the respiratory supercomplexes in other ways and show the same results achieved with the deficiency of MCJ.
17. Authors do not address well the memory parameters throughout the manuscript: if authors take the CAR T cells out of the mouse after different periods of time can they show higher response to the antigen in vitro?

Minor:

1. Arise of CRS and ICANS should be better explained.
2. Explain molecular mechanisms by which memory CD8 cells protect against a lethal dose of influenza virus without the help of CD4 cells. Moreover, it is not clear in what aspect they address the presence of CD4 as irrelevant to the success of the therapy in authors' models.
3. For Fig 1e-f-g please indicate on the figure panel the number of expansions addressed.
4. The use of scatter plots for data representation is highly encouraged.
5. It is not entirely clear how WT or MCJ KO OT-I CD8 cells are sorted from the cultures and administered to mice.
6. There is a bit of confusion regarding why sometimes authors use 1, 2 or 3rd expansion in showing their results/phenotypes.
7. It is not clear why in Fig 3h vs 3i the killing of leukemia is ~25% in WT vs 50%.
8. It would be important to highlight if there is literature available regarding perforin as better killer than IFN γ in leukemia.
9. Indicate better where are authors gating exactly in all the expansions (Fig 5a-c).
10. Often, too much method information is inserted in the results.
11. Regarding Fig 7e, have authors done a western blot after 1 expansion?
12. It is not clear what shMCJ-2 is and how it differs from the other one mentioned/used.
13. Regarding references 34,35,60, please discuss more the mechanism regulating the source of ATP production during cell manufacturing and the in vivo function and persistence.
14. For CAR+ enrichment, are there differences in % of enriched cells for WT and MCJ KO? Authors should characterize these cells for markers other than the ones they show (es chemokine markers, TNF α , TBX21, IL-4, IL-5, CCR4, GATA3, IL-9, IL-10, IRF4, CCR6, KLRB1, IL-17, RORC, CCR7, CD95, TIM-3, LAG-3, CTLA-4, NKG2A, CD39 etc).

RESPONSE TO REVIEWERS' COMMENTS

REVIEWER 1

The authors characterize a clear difference in CD8+ T cell function in mouse and human T cells following MCJ ko with detailed and well performed experiments. They then propose that MCJ silencing is a strategy to improve CAR T cell function. However caution must be exercised with this assertion:

Point 1. On a practical level, current CAR T cell manufacture rarely uses prolonged culture in IL2. Most manufacturing processes have moved to use IL7/IL15 cultures with as brief an expansion period as possible. In fact, cutting edge manufacturing processes lack expansion completely. Demonstrating that MCJ KO/silencing has benefit in CAR T cells generated using IL7/IL17 would strengthen the proposal regarding therapeutic utility of MCJ KO/silencing.

We agree with the reviewer that the exploration of shorter manufacturing and use of cytokines other than IL-2 is an area of great interest in CAR T cell therapy and we have included this in the introduction. However, the majority of knowledge regarding the biology of CAR T cells in patients is based upon CAR T cell products that were manufactured with IL-2, and thus represented the starting point for these studies. Nevertheless, to address the reviewer's point, we have now performed experiments with WT and MCJ KO CAR-T cells expanded in IL7/IL-15. We now show that there is no difference in the cell surface markers and intracellular markers between WT and MCJ KO CAR-T cells expanded in IL7/IL-15 (**new Extended Data Fig. 6b and 6c**). However, similar to CAR-T cells expanded with IL-2, we also observed a superior killing of E2a leukemia cells by MCJ KO CD8 CAR-T cells expanded with IL-7/IL-15 relative to WT CD8 CAR-T cells also expanded with IL-7/IL-15 (**new Extended Data Fig. 6d**).

Point 2 and Point 3. The authors focus entirely on CD8+ T cells, which makes sense given that MCJ is more important in CD8+ T cell biology than CD4+ T cell biology. In the discussion, the authors assert that CD8+ CAR T cell therapy may have advantages over CD4/CD8 CAR T cell therapy since toxicity would be lower due to absence of IL6 secretion from CD4+ T cells. Caution must be exercised with this assertion: Adoptive immunotherapy with only CD8+ T cells has largely failed and mixed CD4/CD8 therapeutics are preferred; While IL6 is an important part of the biology of CAR T immunotoxicity, the main source of IL6 is macrophages in response to T cell secreted IFN-g. Demonstrating utility of MCJ silencing in adoptive immunotherapy models using both CD4/CD8 would strengthen the proposal regarding therapeutic utility of MCJ ko/silencing.

Continuing with this line of thought, if MCJ ko/silencing CAR T cells secrete more IFN-g, they may cause increased toxicity. This aspect should also be considered.

We fully agree with the reviewer's comments that CAR T cell approaches relying on mixtures of CD4 and CD8 CAR T cells have demonstrated advantages over CD8-only CAR T cell products, and this highlight the impact that MCJ-deficiency in CD8 CAR-T cells boosts their effector functions and in vivo efficacy. Regarding the question about the effect of MCJ loss on the mix of CD4 and CD8 CAR-T cells, following the reviewers' suggestion, we have performed experiments where CD8 CAR-T cells and CD4 CAR-T cells were independently generated from WT and MCJ KO mice and expanded with IL-2. We performed in vitro killing assay with CD8 CAR-T cells alone, or combining CD8 and CD4 CAR-T cells (1:1 CD4:CD8 ratio). Interestingly, the killing efficacy of E2a cells was higher for CD8 CAR-T cells than for the combination of CD8+CD4 CAR-T cells, for both WT and MCJ KO cells (**Reviewer Fig. 1a**). This is probably due to a reduced number of CD8 CAR-T cells in the mix of CD4/CD8 cells, since CD8 cells are the more cytotoxic

population. Nevertheless, we also found increased killing by MCJ KO CD4/CD8 CAR-T cells compared with the WT CD4/CD8 CAR-T cells (**Reviewer Fig. 1a**). We also performed an *in vivo* experiment with the CD4/CD8 CAR-T cells. Similar to the *in vitro* results, it seems that the efficacy of CD4/CD8 CAR-T cells is inferior to CD8 CAR-T cells since 100% of the mice that had received WT CD4/CD8 CAR-T cells died (**Reviewer Fig. 1b**), while our previous studies in the manuscript (Fig. 4 in the original version) showed around 20% survival for mice with WT CD8 CAR-T cells. Nevertheless, we also saw increased survival in mice that had received MCJ KO CD4/CD8 CAR-T cells (**Reviewer Fig. 1b**). Since our manuscript is focused on CD8 CAR-T cells, we have not included these data in the manuscript (it will disrupt the flow of the paper) and we provide these data for the reviewer. However, if the reviewer or editor feels that these data should also be included, we will be happy to do it.

Figure 1. (a) CD8 or CD4 cells were isolated from the spleen and lymph nodes of the WT and MCJ KO mice, activated and transduced separately to generate CAR-T cells as described in Fig. 3a of the manuscript. CD8 and CD4 CAR-T cells were separately cultured with IL-2 for 3 expansions, washed, and incubate in medium for 24h. 1:1 ratio of CD4 and CD8 CAR-T cells or only CD8 CAR-T cells were co-cultured with E2a cells (0.5 E:T ratio) to assess the killing activity. Error bars show Mean± SD. **(b)** WT

and MCJ KO CD4 and CD8 together were isolated from lymph nodes/spleen of the mice, were activated and transduced to express CD19-BBz CAR as described in Fig. 3a of the manuscript. WT hosts were administrated with 10^6 E2a cells. After 3 days, the leukemia-bearing mice were irradiated, and i.v. injected the following day with WT or MCJ KO CD4/CD8 CAR-T cells expanded with IL-2 (60 IU/ml) for 3 expansions (10^5 CAR positive cells total per mouse) (n=7 mice per group). The survival of the mice was followed over time. **p* < 0.05, as defined by 2-way ANOVA (a), and by log-rank (Mantel-Cox) test (b).

With regard to the reviewer's point about macrophages being the primary source of IL-6, thereby negating the potential benefit of a CD4-depleted product for the purpose of reducing toxicity, we kindly disagree. While monocytes and macrophages have been implicated as major sources of IL-6 in CRS, there is evidence that CD4 CAR T cells are still key drivers of this IL-6 production. Giavridis et al (Nature Medicine, 2018, PMID 29808005) demonstrated mouse monocytes/macrophages secreted higher amounts of mIL-6 when CAR-T cells are engineered to express the murine CD40L, implicating the importance of direct contact. More recently Bove et al (JITC, 2023, PMID36593069) demonstrated that CD4 and not CD8 CAR-T cells exacerbated CRS in a humanized mouse model. We also contend that CD4 CAR-T cells can be a source of IL-6, as our recent paper demonstrated conventional human CD4 T cells produce high levels of IL-6 upon activation (Valenca-Pereira et al. PNAS 2021, PMID 34507993) similar to findings reported by Norelli et al (Nature Medicine, 2018, PMID 29808007, Sup Fig 15). Within the introduction and discussion, we now discuss the role of CD4 CAR T cells in the models of CRS described above. We also provide new data showing how human CD4 cells, but not human CD8 cells, produce high levels of IL-6 upon activation (we had not previously shown that CD8 cells do not make IL-6) (**new Extended data Fig. 22a**). In addition, we also examined the production of IL-6 by human CD4 cells during the expansion with IL-2 (something we have not previously investigated/published). To our surprise, we found that CD4 cells produced IL-6 during the expansion with IL-2 in the absence of TCR signaling (**new Extended data Fig. 22b**). We refer to these data in the discussion section since in the manuscript it will be out of context, but if the editor or reviewer do not consider to be appropriate within the Discussion section, we can delete it and follow it up in future publications.

With regards to the concern that the increased IFN γ may lead to increased toxicity, we do not believe this will contribute to increased CRS. While elevated IFN γ is frequently seen in CRS, the recent publication by Boulch et al (Cell Reports Medicine, 2023, PMID 37595589) did not find IFN γ was necessary for CD4 CAR-T cell mediated CRS. Furthermore, we have included experiments testing human CD14 cells (monocytes) from a healthy donor for IL-6 production upon treatment with human IFN γ (or LPS as a positive control). While LPS induced high production of IL-6, IFN γ has no effect on the levels of IL-6 (**Reviewer Fig. 2a**). As a positive control for the activity of IFN γ , we also examined the expression of HLA class II (HLA-DR and HLA-A2), and the results show that IFN γ causes a clear upregulation of HLA class II (**Reviewer Fig. 2b**). We provide the data for the perusal of the reviewer. We do not think it will be required to include in the manuscript but if the reviewer or the editor believe these data should be included we will be happy to include it.

Figure 2. (a) CD14+ monocytes were isolated from a healthy donor PBMC, and cultured (n=3) for 24 hours with LPS (1 μ g/ml), IFN γ (200 IU/ml), or medium alone. Supernatants were collected after 24 hours to measure human IL-6 levels by ELISA. Error bars show mean \pm SD (a); *p < 0.05, as defined by Student's t-test.

(b) Purified CD14+ monocytes were activated for 48 hours in the absence or presence of IFN γ (200 IU/ml). HLA-DR and HLA-A2 expression was examined by flow cytometry analysis.

Point 4. While the metabolic "re-wiring" and consequences on effector function caused by MCJ/silencing is well shown, there is not a clear explanation of key biologic observation in terms of broader T cell biology. More specifically, what is the link between observed functional benefit and IL2 exposure. In the discussion, the authors compare this strategy to that of pharmacologic AKT inhibition during CAR T cell manufacture. In that approach, CAR T cell stemness, Tn and Tscm subtypes are preserved along with polyfunctionality. Understanding the consequence of MCJ ko/silencing in a similar fashion would be helpful in understanding therapeutic application. I recommend providing some experimental detail addressing these points and/or being more guarded regards potential therapeutic utility.

We apologize for the lack of clarity regarding how silencing MCJ expression in CAR-T cells could result in a functional benefit for CAR-T cells, primarily after expansion of CAR-T cells in IL-2. We have previously shown that increasing mitochondrial respiration enhances CD8 cell functions with high-ATP demand such as cytotoxic degranulation and cytokine secretion (e.g. IFN γ) (Champagne et al Immunity 2016). In this manuscript, we show the loss of MCJ function and subsequent increase in T cell effector functions also applies to CAR-T cells. We also show that CAR-T cell mitochondrial function diminishes with multiple rounds of expansion in IL-2 (as measured by mitochondrial membrane potential in Figure 5a vs 5c) and that this mitochondrial function is improved in CAR-T cells lacking MCJ (Figure 5a-f) resulting in improved ATP production (**new Extended Data Figure 11**). Both cytokine secretion and cytotoxic degranulation with their high-ATP demand are improved in MCJ KO CAR-T cells correlating with their enhanced metabolisms. This correlation is strengthened by our new data showing that inhibition of Complex V/ATP synthase (OXPHOS) with oligomycin negates the cytotoxic advantage of MCJ KO CAR-T cells over WT CAR-T cells (**new Figure 5k**), further demonstrating that cytotoxic activity is highly dependent on mitochondrial ATP.

We do not see major shifts in the expression of memory/effector markers in MCJ KO CAR-T cells relative to WT CAR-T cells after manufacturing (**new Extended Data Figures 6a-b, 8a**), although we found a slight increase of the expression of Foxo1 and TCF1 (markers associated with more stem cell memory phenotype) in MCJ KO CAR-T cells (**new Extended Data Figure 6c**). The increase in TCF1 expression was further supported by our new RNAseq data (**new Extended Data Fig. 13d**). We do not see major shifts in the expression of memory/effector markers in MCJ KO CAR-T cells relative to WT CAR-T cells in vivo (**new Extended Data Figures 8b-d**), however ex vivo killing activity of in vivo MCJ KO CAR-T cells is also increased (Fig. 4b). Our studies in human CAR-T cells show similar increases in mitochondrial membrane potential (Figure 7b), ATP production (**new Extended Data Figure 19**), and effector functions (Figure 7c, 7g-l and **new 7m**) with MCJ-silencing. Similar to our murine findings, we did not see major changes in the composition of human CAR-T cell products with MCJ-silencing (**new Extended Data Figure 20a-b**).

Furthermore, in the original submission, we showed the enhanced mitochondrial respiration in mouse MCJ KO CD8 CAR-T cells relative to WT CAR-T cells (Fig. 5). We now show that silencing MCJ (shMCJ) in human CD8 CAR-T cells also results in enhanced mitochondrial respiration (**new Fig. 7f and 7j**). Thus, while far from being developed as therapeutics, the strategy is silencing MCJ in CAR-T cells to sustain the mitochondrial respiration to enhance effector function. We have updated the discussion (**page # 27-28**) to clarify this interpretation of the role MCJ-deletion/silencing may impact CAR T cell therapy.

Point 5. It may be interesting to see if pharmacologic downregulation of MCJ using Sinapine has similar effects to genetic KO/silencing.

We believe that the reviewer is referring to the paper by Li et al. 2022 in Life Science. However, this paper does not show any specific effect of sinapine, a natural component of the rapeseed oil, on MCJ expression. There are several concerns with the paper. First, the Western blot analysis for MCJ levels in some cells (not specified in the figure legend but most likely THLE-2 cell line) treated with sinapine shows just a small effect on what they called MCJ protein. This Western blot does not show an important control, a mitochondrial protein such as CoxIV to discard that sinapine causes a reduction on total mitochondrial mass. More importantly, the authors claim (in the Methods) that the anti-MCJ antibody they used to examine MCJ by Western is from R&D. However, we have verified that R&D does not provide anti-MCJ Abs. The University of Vermont has a patent on anti-MCJ antibodies made against the N-terminal of MCJ, the only unique region of MCJ since the C-terminal is conserved with other DnaJ proteins, and R&D has not licensed it. Thus, we question the specificity of the anti-MCJ Ab and the validity of a single panel in the manuscript examining MCJ. The manuscript also presents JC1 staining as a readout for ROS in cells, but JC1 staining is used for mitochondrial membrane potential. Based on our assessment of the referenced paper we do not feel there is sufficient evidence to conclude that sinapine will specifically decrease the expression of MCJ. Unfortunately, other than siRNA/shRNA we do not have a mechanism to specifically decrease MCJ expression.

Point 6. What are the implications of reduced Caspase 3 activity associated with MCJ ko/silencing in these experiments?

We think the reviewer is referring to the study by our collaborators (Secinaro et al. Front. Cell Dev. Biol. 2019) examining activation of caspase 3 by IL-2-induced glycolysis during expansion of WT and MCJ KO CD8 cells. In our studies with mouse or human CD8 CAR-T cells we have not observed differences in proliferation or survival during the expansion with IL-2. Nevertheless, we have additionally examined the presence of active caspase 3 (cleaved caspase 3) in human CD19-BBz/shMCJ-2 CD8 CAR-T cells and CD19-BBz/c-shRNA CD8 CAR-T cells.

After 3 expansions with IL-2 we could not detect differences in the levels of cleaved caspase 3 between shMCJ CD8 CAR-T cells and c-shRNA CD8 CAR-T cells. Thus, we do not think impaired activation of caspase 3 plays a role in the increased killing activity of CAR-T cells lacking MCJ. We provide the data here for the perusal of the reviewer (**Reviewer Fig. 3**), we do not think is essential for the manuscript, but we will be happy to also include it in the manuscript if the reviewers feel this is necessary.

Figure 3. CD8 cells from donor with high MCJ expression were activated and transduced to generate CD19-BBz CAR-T cells as described in Fig. 7e. The CD19-BBz CAR-T cells were isolated from the cultures after 3 expansions in IL-2. The expression of pro-caspase 3 and cleaved-caspase 3 were determined by Western blot analysis. Anti-Caspase 3 Ab (Cell Signaling Technologies, Cat#9662) were used to detect both pro- and cleaved- form of caspase 3.

SPECIFIC COMMENTS

Point 7. Regarding supplementary figure 1: Tumour size and survival is shown in the main figure of ko mice, but only survival was shown in the supplementary figure of T-cMCJ KO mice. Since there was a difference in survival, this must have been due to tumour size - please show this.

We now provide those data (**new Extended Data Fig. 1k**).

Point 8. It would be helpful if histograms with superimposed scatter of individual data points was used.

Following the reviewer recommendation, we have now revised all the bar graphs figures to show individual data points.

Point 9. In suppl Fig 3(b), could most of the difference in tumour size be ascribed to two mice in QT OT-1?

We now verified that this is not the case. Even if we eliminate the data from those two mice we still have almost statistical significance ($p=0.06$) despite the reduce "n" number (only $n=4$ for WT group after eliminating the two mice that the reviewer was concerned). In addition, the results from this experiment were reproduced in another earlier independent experiment that was performed in a different institution. Thus, we feel highly confident in the results and conclusions from these experiments, and we show individual mice (Extended Data Fig. 4b) for transparency.

REVIEWER 2

The manuscript by Wu and colleagues describes how MCJ/DnaJC15, a known regulator of metabolism, and of mitochondrial complex I can be targeted to increase anti-tumour efficacy of CD8 T cells. The CD8 T cells could be modified both with tumour-specific CARs and TCRs. The authors demonstrate reduced growth of B16-OVA tumours in OT-I MJC knock-out mice versus WT mice and also enhanced in vitro killing of B16-OVA tumours by OT-I cells from the KO mice. The enhanced in vitro killing is abrogated when neutralizing antibody against IFN-g is added or Concanamycin A (CMA) which blocks perforin-mediated killing.

When WT mice transplanted with B16-OVA tumours were treated with in vitro expanded OT-1 cells from either WT or MJC KO mice, the MJC KO OT-I showed improved tumour infiltration and controlled tumour growth more efficiently. No difference in cell surface markers like CD44 or PD-1 were detected between the OT-I cells from MJC KO versus WT mice, but cells from MJC KO

mice produced higher levels of IFN-g. The same findings were shown in a CD19 CAR murine leukemia model comparing CAR CD8 T cells from WT versus MJC KO mice.

CD19CAR CD8 T cells from MJC KO mice killed leukemia cells more efficiently, produced higher levels of IFN-g even after withdrawal of IL-2 used for expansion, but seemed not to rely on IFN-g for killing. The enhanced killing in MJC KO T cells was however inhibited by the addition of CMA or EGTA, blocking pathways required for perforin-based cytotoxicity.

CD19CAR CD8 T cells from MJC KO and WT mice showed different metabolism, with the MJC KO cells having higher mitochondrial respiration. Finally, the authors also investigated the role of MCJ in human primary CD8 T cells, including CD19CAR CD8 T cells. The lentiviral CAR construct also contained either shRNA MCJ to knock down MCJ in CAR T cells or control shRNA. In summary, the study is timely as several reports of the importance of CAR-T metabolism for the persistence and long-term effect in vivo have been published. The manuscript provides ample evidence for the improved anti-tumour efficacy of MCJ negative T cells.

Point 1. The authors did not report any differences in phenotype between the MCJ knock-out or knock-down T cells and the WT T cells. As memory CD8 T cells rely on mitochondrial respiration for survival and efficacy, it would have been interesting, however, to test additional memory T cells markers as well as comparing MCJ function in IL-2 expanded T cells with IL-7/IL-15 expanded T cells as the latter is thought to induce more central memory T cells. Did the MCJ negative CAR/TCR transgenic cells show any differences in stem-cell memory markers such as the transcription factor TCF-1?

Following the recommendations from the reviewers, in addition to the cell surface markers (e.g. CD44, CD62L, etc) we have now examined the expression of transcriptional regulators like Tcf1, Foxo1, Tbet and Tox to further defined MCJ-mediated differences in CAR-T cell differentiation. Moreover, we have examined the activity of WT and MCJ KO CD8 CAR-T cells generated in the presence of IL-7/IL-15. As previously published, we observed a difference in the expression of Foxo1 and Tcf1 between CAR-T cells generated in the presence of IL-2 versus IL-7/IL-15 (**new Extended Data Fig. 6a, 6b, 6c**). We did not observe drastic differences in the expression of Tcf1, Foxo1, Tbet and Tox between WT and MCJ KO CD8 CAR-T cells expanded in IL-7/IL15, but noted a slight increase in Tcf1 and Foxo1 in MCJ-KO CAR-T cells expanded in IL-2 (**new Extended Data Figure 6c**). We have now performed RNA-seq analyses in WT and MCJ KO CD8 CAR-T cells generated with IL-2 and, interestingly, the expression of *Tcf7* (gene encoding TCF1) was also slightly upregulated (**new Extended Data Fig. 13d**). While these data suggest disruption of MCJ may alter the differentiation of CAR-T cells, we did not see a difference in the phenotypes of MCJ-KO CAR-T cells relative to WT CAR-T cells in vivo at 1, 3, or 4 weeks after transfer into leukemia-bearing mice (**new Extended Figure 8b-d**). Based on the lack of impact on in vivo differentiation, we continue to favor that the impact of deleting MCJ is on the increased mitochondrial production of ATP which is essential for effector processes like cytokine secretion or cytotoxicity.

In addition, to address the reviewer's questions, we now show that if we inhibit mitochondrial ATP production with oligomycin, we block the killing activity of CD8 CAR-T cells, showing the essential role of mitochondrial ATP in the killing activity of CAR-T cells (**new Fig. 5k**). Furthermore, we show now that, similar to mouse CD8 CAR-T cells, silencing MCJ in human CD8 CAR-T cells also results in increased mitochondrial respiration (**new Fig. 7f and 7j**). Silencing MCJ however has minimal effect on phenotype (Tcm, Tscm, Tem) of human CD8 CAR-T cells (**new Extended Data Fig. 20a and 20b**).

Following the reviewer's suggestion, we have also examined whether the loss of MCJ could improve the killing activity of CD8 CAR-T cells expanded with IL-7/IL-15 instead of L-2. The

data show enhanced killing activity by MCJ KO CD8 CAR-T cells relative to WT CAR-T cells expanded with IL7/IL-15 (**new Extended Data Fig. 6d**).

Point 2. Whether removing MCJ regulation of mitochondrial respiration omits the need for CD4 T cells in the formation of memory and long-term efficacy of CD8 CAR T cells may be too early to say as this may not be sufficiently demonstrated in the CAR-T models used. It would, however, be interesting to perform repeated in vitro re-challenges of the CAR T cells with leukemia cell lines to see if the MCJ negative CAR T cells maintain their function over time.

Since Reviewer 1 also have questions regarding whether loss of MCJ could have an effect on combination of CD4/CD8 CAR-T cells, we have now performed additional experiments to address this question. Please, see **Reviewer 1 Point 2 and Point 3** for extended description of the experiments, the results and the data. In summary, loss of MCJ also enhance killing activity of a combination of CD4/CD8 CAR-T cells. As we mentioned to Reviewer 1, we provide the data for the reviewers' purpose (since the manuscript is focused on CD8 CAR-T cells), but we will be happy to include these data in the manuscript if the reviewers or the editor recommend.

Specific comments:

Point 1. In the abstract the authors claim that they here identify the protein as a target to enhance mitochondrial respiration. However, they do not cite e.g., the following paper until the discussion: Champagne et al., *Immunity* 2016 (PMID: 27234056). Please modify statement in the abstract as the authors already identified MCJ as a target in previous publications. The introduction to adoptive T cell therapies also needs some updated references for TCR- and CAR-based T-cell therapy. Some of the later clinical trials should also be cited and a recent overview of approved CAR-T therapy (PMID: 37256141).

We apologize, but we kindly disagree with the reviewer on "*they do not cite e.g., the following paper until the discussion: Champagne et al., Immunity 2016 (PMID: 27234056)*". We cite 5 times this paper (our paper) describing MCJ in CD8 cells, mitochondrial respiration and influenza infection in the Introduction section (**page #5, reference #51**). Indeed, this paper was the basis for the current manuscript.

Regarding the abstract, we apologize if our statement may have been misleading. We did not say that "*here identify the protein as a target to enhance mitochondrial respiration.*" We have already shown that MCJ is a regulator of mitochondrial respiration. In the abstract we say that we identify **MCJ as a metabolic target in CD8 CAR-T cells** (*Here we identify MCJ/DnaJC15, an endogenous negative regulator of mitochondrial Complex I, as a metabolic target to enhance mitochondrial respiration in CD8 CAR-T cells*). We have previously investigated MCJ in CD8 cells in the context of influenza virus infection. This manuscript is the first study investigated the role of MCJ in CAR-T cells and CD8 cell anti-tumor response. If the reviewer feels we still have to modify the statement, we can try to rewrite it.

We agree with the reviewer with the need of updated references in the Introduction and the Discussion. We have added the recommended review to the references (**reference #4**) and other references, including clinical trials.

Point 2. Figure 1c: update to show how many days n= for expansion

We had the drawing with an "n" only because it was a schematic representation for the experiments in the figure where n is 2, 4 or 6, depending on the panel. However, we agree with

the reviewer that the "n" by itself is confusing so we have now **revised Fig 1c** to display n= 2, 4 or 6 days.

Point 3. Figure 4. Fig. 4b: CD19 CD8 CAR-T cells isolated from the BM of injected mice 30 days later and used for ex vivo killing assays of E2a leukemia cells. What were E:T ratios, % of CAR T cells? Please show E:T ratios and non-normalised values. Was tumour load measured at all? If the tumour cells already contained GFP, a construct containing luciferase and the use of IVIS could easily have been adapted for monitoring of tumour load.

We apologize for the lack of clarity and missed experimental information. We have now added the E:T ratio for CAR-T cells to E2a cells to figure legend (**revised Figure 4b legend**) For in vivo studies we did not use GFP/luciferase-tagged E2a cells. The experiments were performed in immunocompetent mice and in prior attempts to develop such a model, the expression of these two foreign proteins caused spontaneous rejection by endogenous T cells (unpublished data). This is in contrast to the parental E2a cell line used in these studies in which mice uniformly succumb to leukemia with as little as 10,000 cells per mouse and ~80% will succumb to leukemia at a dose of 1,000 cell per mouse (Qin et. al. Blood, 2018).

Point 4. Figure 5: The Seahorse MitoStress assay (Figure 5e-g) showed higher maximal respiration and spare respiratory capacity in MCJ KO cells. How many parallels were performed? Please provide some more detail of the number of replicates and use standard deviations rather than SEM in graphs f-g. The authors also performed metabolic profiling by mass spec, which strengthen the data.

We have now added the "n" number in the figure legend for all the Seahorse Mitostress figures, including the two new figures (new Fig. 7f and 7j).

In addition, following the recommendations from this reviewer as well as Reviewer 2 and Reviewer 3 we have now revised all the graphs in the manuscript (not only Fig. 5) to show SD instead of SEM.

Point 5. Figure 6: For Western blots, uncropped versions of gels/blots presented in the figure should be shown in the supplementary data or source data file. The results showed very heterogenous expression in CD8 T cells from healthy donors (Figure 6d) which had no correlation with age or sex. Did the authors look at whether it correlated to any degree with phenotype/activation status/differentiation of the CD8 T cells? How important was the difference in MCJ expression in the two healthy donors relative to the difference in WT and MCJ KO murine cells?

Regarding the uncropped versions of the Westerns, following the journal policies we will provide the uncropped versions, as well as raw data as the Source Data file (instead of supplementary figures) prior to the publication.

The reviewer makes a good point regarding the expression of MCJ in human CD8 cells according to their phenotype. This was also a question when we started these studies. We therefore examined MCJ expression in human naive, effector memory (Tem) and effector memory re-expressing CD45RA (Temra) CD8 cells and we found similar expression in all subsets, which is presented in **new Extended Data Fig. 16**.

Regarding "how important was the difference in MCJ expression in the two healthy donors relative to the difference in WT and MCJ KO murine cells", we think the reviewer is referring to Fig. 6 where we show a MCJhigh and a MCJlow healthy volunteers. The difference in the

expression in those is about 2-fold in the MCJ^{high} (Fig. 6e). Based on the distribution on the expression of MCJ in a large healthy population it seems to vary up to 3-fold (Fig. 6c and 6d).

In MCJ KO CD8 cells the expression of MCJ is abolished as expected. In our studies in human CD8 CAR-T cells expressing shMCJ, the expression of MCJ is not totally abolished as expected, we get a reduction of MCJ levels by about 2-3-fold (Fig. 7e and 7i)). However, we show an enhanced anti-tumor activity (Fig. 7g and 7k). In addition, we have now performed Mitostress Assay in those human shMCJ CAR-T cells and the results show a clear increase in mitochondrial respiration (**new Fig. 7f and 7j and new Extended Data Fig. 19a and 19b**). Thus, we now demonstrate that reduction on MCJ levels is sufficient to increase mitochondrial respiration. Moreover, as part of an independent project, we have shown that a reduction of MCJ levels (not a complete abrogation) in the liver in vivo using an siRNA causes a major reduction in liver steatosis and fibrosis in mouse models of fatty liver disease, and we are currently in talks with FDA for an IND to treat NASH patients (there are 5 siRNA-based drugs FDA approved for liver diseases).

Point 6. In Figure 7, how many donors were tested for the siRNA MCJ Knock-down experiments? Did the donors have high or low MCJ expression to start with? In Figure 7c, CD8 T cells were stimulated with aCD3/aCD28 to measure IFN- γ levels. The siMCJ cells produced around 2-fold the amount of IFN- γ . aCD3/aCD28 stimulation is very strong. Could this difference in IFN- γ production also be seen if antigen-specific stimulation was used, studying human CAR T cells as in Fig. 6?

The shMCJ CAR-T cells studies in Fig. 7 were done with three donors, as represented in the figures by the labels H16 and H19 for killing assays and CD107a assays (**Fig. 7g, 7h, 7k, 7l**). Seahorse experiments were done with another independent donor (**new Fig. 7f and 7j and new Extended Data Fig. 19a and 19b**). All three donors were defined as MCJ^{high}. The experiments transfecting siRNA were performed in 2 independent donors (one representative shown). In addition, following the recommendations of Reviewer 3, we now provide data on shMCJ CAR-T cells from another independent donor defined as MCJ^{low} (**new Extended Data Fig. 21a and 21b**).

Regarding whether IFN γ production could be examined in antigen specific stimulation of CD8 cells, this is almost impossible in human CD8 cells because the low frequency of TCR specific CD8 cells. However, to address the reviewer point we have included experiments examining IFN γ production by human CD19-BBz/shMCJ CD8 CAR-T cells and CD19-BBz/c-shRNA CD8 CAR-T cells activated with Nalm16 leukemia cells. The production of IFN γ by shMCJ CD8 CAR-T cells is higher than the production by c-shRNA CD8 CAR-T cells (**new Fig. 7m**). In addition, as a negative control we used CD19 negative Nalm16 cells, which did not induce IFN γ production (**new Fig. 7m**).

When human CD19 CAR CD8 T cells were produced, the lentiviral CAR construct also contained either shRNA MCJ to knock down MCJ in CAR T cells or control shRNA (Fig. 7 d-e). The level of MCJ did not look very high in the control, but was clearly reduced by the shRNA MCJ. The CAR expression levels shown in Extended Fig 9 were not very high. The killing assays performed by flow cytometry should be explained in more detail. The E:T ratio is mentioned in the figure legend and is low (e.g. 0.125). After 20h, the remaining leukemia cells which express GFP are counted by flow cytometry. Please explained how this was done to accurately quantify the cancer cells. Were flow count beads used? Example FACS plots need to be shown in extended data. SEM are shown for all figures, please show standard deviations instead.

We apologize, we now provide additional experimental details in the Methods section and the figure legend. In addition, following the reviewer's recommendation we provide examples of

FACS plots (**new Extended Data Fig. 5c**). Moreover, following the recommendations from the reviewer we have now revised all bar graphs in the manuscript to show SD instead of SEM.

Point 7. What were the humane endpoints of in vivo experiments? Please clarify.

We apologize for not including this important information in the Methods section. We have now added to the revised manuscript. We thank the reviewer for pointing out this error.

Point 8. Please provide additional details for antibodies used in IHC staining in Fig.2b

We apologize for this missing information. We have now incorporated this information in the Methods section (**page #35**).

REVIEWER 3

Meng-Han Wu et al target MCJ/DnaJC15 to increase CAR T cell persistence and function. They found a valid pathway linking bioenergetics with anti tumor activity. The application of this pathway to CAR T cell therapy is novel and compelling and this manuscript in this regard would provide substantial information for future analyses for both basic and translational research in the fields of immunology/ immunometabolism and cancer biology.

Part of the authors' conclusions are not fully supported by the data specifically regarding direct linking of MCJ to respiration and respiration to antitumor cytotoxicity of CAR CD8 T cells. This aspect would require revision. Another additional evidence that is needed in my opinion is a full characterization of the metabolic phenotype of these cells and their metabolic requirements for in vivo activity and persistence. This is lacking at the moment in the manuscript and I believe it is fundamental especially as a future source of interrogation for the community of what metabolic pathways and aspects characterize a fully functional tumor killing CAR T cell in vitro and in vivo.

We are pleased to know that the reviewer considers our manuscript novel and compelling. We have now tried to address the concerns that the reviewer has regarding the characterization of the metabolic profile of CD8 CAR-T cells in the absence of MCJ as well as providing experimental evidence to connect the effect of MCJ on mitochondrial respiration with the effect of MCJ in antitumor cytotoxicity of CD8 CAR-T cells.

In our original version of the manuscript, performing unbiased metabolomics analysis we showed differences in the metabolic profile of MCJ KO and WT CD8 CAR-T cells after 3 expansions with IL-2. The results support an enhanced mitochondrial metabolism (current Fig. 5g). In addition, we have now performed RNAseq analysis in WT and MCJ KO CD8 CAR-T cells after 3 expansions with IL-2 as well. The results from these analyses further show that the main component that distinguishes WT and MCJ KO CD8 CAR-T cells is metabolism, with most of the ten top enrichment pathways being metabolic pathways primarily associated with mitochondria such as OXPHOS and fatty acid oxidation (**new Fig. 5i and new Extended Data Fig. 13a, 13b, 13c and 13d**).

Moreover, since we propose that increased mitochondrial respiration in MCJ KO CD8 CAR-T cells can improve the fitness of these cells when transferred to patients in an environment without high levels of exogenous IL-2, we have now performed metabolomic analysis of WT and MCJ KO CD8 CAR-T cells after three expansions with IL-2 followed by incubation in cytokine-free

medium. The results further support an enhanced mitochondrial metabolism in MCJ KO CAR-T cells even under these conditions of cytokine withdrawal (**new Fig. 5j and new Extended Data Fig. 14a and 14b**). Interestingly, these results also suggest a potential adaptation of the metabolism towards enhanced protein catabolism and the use of mitochondrial fatty acid beta-oxidation as a source of energy (**new Fig. 5j and new Extended Data Fig. 14a and 14b**).

Following the reviewer's suggestion, we have also performed metabolomics in WT and MCJ KO CD8 CAR-T cells isolated from bone marrow 4 days after *in vivo* transfer to mice bearing E2a leukemia. This is a challenging experiment in terms of adequate T cell recovery which is why we chose a time point of CAR-T cell expansion for this evaluation. The results also show a selective difference in the metabolic profile (**new Extended Data Fig. 15a**). We found an accumulation of short/medium-chain fatty acids in MCJ KO CD8 CAR-T cells relative to WT CAR-T cells (**new Extended Data Fig. 15b**). Pathway enrichment analysis indicated likely alterations in fatty acid metabolism and beta oxidation in MCJ KO CAR-T cells (**Extended Data Fig. 15c**). Interestingly, fatty acid oxidation is a key pathway for memory CD8 cell and the presence of short chain fatty acids has been associated with promoting the generation of memory CD8 cells (Bachem et al. Immunity, 2019).

We also provide additional data showing that reducing MCJ levels in human CD8 CAR-T cells (shMCJ) results in a marked increase in mitochondrial respiration (**new Fig. 7f and 7j, and new Extended Data Fig. 19a and 19b**).

To further show that the increased mitochondrial respiration found in the absence of MCJ is responsible for the enhanced anti-tumor killing activity of MCJ KO CD8 CAR-T cells (the question raised by the reviewer), we have now performed experiments using oligomycin, the inhibitor of Complex V/ATP-synthase. The results show that a short pre-treatment with oligomycin blocks the anti-tumor killing activity of CD8 CAR-T cells (**new Fig. 5k**), demonstrating that mitochondrial respiration is essential for CAR-T cells anti-tumor activity.

Major Points:

Point 1. Metabolic re-programming of CD8 cells should be better explained and referenced. Authors hint to a complete shift towards glycolysis when it is instead the case of a global increase in basal metabolism during activation, where both glycolysis and OXPHOS play a role.

We apologize for the lack of clarity and description about the metabolomics studies. We do not propose that the lack of MCJ causes a complete shift towards glycolysis. Instead, our data indicate a "shift" towards enhancing mitochondrial respiration. We also agree with the reviewer that it is not a "complete shift" since glycolysis is not compromised. Please, see the new experiments and additions that have been done now in the revised manuscript to better support the effect of MCJ on metabolism of CD8 CAR-T cells.

Point 2. Only in the discussion authors talk about the metabolic limitations of the CAR T persistence *in vivo*. Authors should provide more examples starting from the introduction and talk more also about what are the current available data that show that CAR T (specifically CD8) with better bioenergetics (and less glycolytic phenotype) live longer in the host with antitumor capacities. On the same note, authors should discuss better the attempts regarding PGC1a and based on what data this has been speculated as an energetically high cost process that cannot be further pursued.

We have now revised the Introduction to provide more specific examples and corresponding reference regarding the effect of inhibiting glycolysis in the differentiation stage of CAR-T cells (**page #5**). We apologize, we did not imply that promoting mitochondrial biogenesis cannot be further pursued, we try to state that promoting mitochondrial function without the need

of increasing mitochondrial mass (mitochondrial biogenesis) could be more beneficial. The "energetic cost" for mitochondrial biogenesis is the same that the biogenesis of any other organelles such the ER, Golgi etc. Mitochondrial biogenesis requires a large amount of lipids for the biosynthesis of the outer and inner membrane. The synthesis of lipids is an energetic cost. We have revised our description now in the Introduction (**page #5**).

Point 3 and Point 4. It would be very important that authors explain better the molecular mechanisms by which MCJ is an endogenous negative regulator of Complex I and how it inhibits supercomplex assembly. Moreover, authors should talk about other reported or possible functions of MCJ.

Do MCJ-deficient mouse CD8 cells have higher CI activity? What about the activity of the other ETC complexes?

We apologize for the lack of sufficient background regarding MCJ and how MCJ regulates mitochondrial respiration. We have now revised the introduction (**page #5**) to provide the readers with better background. We have shown that MCJ interacts with Complex I and that the absence of MCJ increases Complex I activity. This has been shown in CD8 cells, heart, liver, cell lines by different studies (we have now included more references in the Introduction). We have also shown that the absence of MCJ facilitates the formation of supercomplexes (Complex I, III and IV) and thus increases mitochondrial membrane potential without increasing ROS production. We now provided a better description in the Introduction and additional references. The molecular mechanisms by which MCJ affects the formation of supercomplexes remains unclear but we think that interaction of MCJ with complex I affects the conformation and interaction with Complex III and Complex IV. Although it remains unclear, it is believed that Complex I has an "active" and "inactive" conformation of Complex I (Babot et al. *Biochim. Biophys. Acta.* 2014, Maklashina et al. *Biochim. Biophys. Acta.* 2003).

Point 5. Are there differences in CD4 T cells or ILCs activation in T-cMCJ KO?

We now include more references (**page #8**) regarding the selective expression of MCJ in CD8 cells relative to CD4 cells and the fact that lack of MCJ does not affect the mitochondrial membrane potential in this population (Hatle et al. *Mol. Cell. Bio* 2013). In addition, we now provide data showing no difference in activation markers (**new Extended Data Fig. 1g**) and IFN γ production (**new Extended Data Fig. 1i**) in CD4 cells from T-cMCJ KO mice after activation

Point 6. The conclusion of a superior killing linked to ATP represents a bit of a stretch if authors are trying to link killing to OXPHOS. Authors don't really prove here that ATP is important in this process in this model. What are ATP levels in these cells? Can authors interfere with ATP specifically to modulate this phenotype?

We agree with the reviewer that we were missing this key experiment in the original version, and we thank the reviewer for bringing this point. We now provide data showing that the ATP levels are higher in MCJ KO CD8 CAR-T cells (**new Extended Data Fig. 11**). In addition, following the reviewer's suggestion we now provide data showing that a short pre-treatment with oligomycin, an inhibitor of Complex V/ATP-synthase that mediates mitochondrial ATP production, abolishes the enhanced killing activity of MCJ KO CD8 CAR-T cells, showing the cytotoxic activity of CAR-T cells is highly dependent on mitochondrial ATP (**new Fig. 5k**). Exocytosis of the granules containing granzyme and perforin at the cytoplasmic membrane is a process highly dependent on ATP. We have previously shown the presence of ATP-rich microdomains (high ATP concentrations) in CD8 cells even closed to the membrane and the need of mitochondrial ATP (Champagne et al. *Immunity* 2016).

Point 7. Authors should show better how the withdrawal of IL-2 affects T cell metabolism specifically in WT vs MCJ deficiency. Moreover, it would be important that authors describe better the molecular mechanisms behind IL-2-mediated upregulation of MCJ and how mechanistically IL2 promotes glycolysis while compromising mitochondrial respiration in T cells.

The first studies showing that IL-2 promotes glycolysis over mitochondrial respiration, in contrast to IL-15, in CD8 cells came from Erika Pearce's group (Immunity, 2012). We have now clarified it and provided appropriate references in the results section (**page #16**). It has been shown that IL-2 promotes glycolysis in part by increasing Glut1. It has also been shown that the total mitochondrial mass and the morphology of mitochondria is different between CD8 cells expanded with IL-2 (more effector) versus cells expanded with IL-15 (more like memory cells) (van der Windt et al. Immunity 2012, Buck et al. Cell 2016).

To address how the withdrawal of IL-2 affects T cell metabolism in WT vs MCJ deficiency, we have now performed metabolomic analysis of WT and MCJ KO CD8 CAR-T cells after three expansions with IL-2 followed by incubation in cytokine-free medium. As mentioned above, the results further support an enhanced mitochondrial metabolism in MCJ KO CAR-T cells even under these stress conditions (**new Fig. 5j and new Extended Data Fig. 14a and 14b**). Interestingly, these results also suggest a potential adaptation of the metabolism towards enhanced protein catabolism and the use of fatty acid beta-oxidation (also in mitochondria) as a source of energy (**new Fig. 5j and new Extended Data Fig. 14a and 14b**).

Regarding the regulation of MCJ by IL-2, our collaborators (Secinaro et al. Front. Cell. Dev. Biol. 2019) have shown that inhibiting glycolysis with 2-DG reduces substantially the levels of MCJ in IL-2 expanded CD8 cells. In addition, they show that IL-15 reduces MCJ levels and this is due to increases DNA methylation (it is known that MCJ expression is regulated by gene methylation). We already cite these studies, but if the reviewer considers we should extend our discussion to further mention these studies we will be happy to do it.

Point 8. Regarding Fig 3f, authors should speculate on the fact that the anti-IFN γ antibody doesn't work for E2a compared to the B16 tumor authors used previously in the results.

We have now added in the Results section after the presentation of Fig. 3f a potential explanation for the different IFN γ dependency between killing B16 by TCR-specific CD8 cells and the killing of E2a cells by CD8 CAR-T cells (**page #14**). IFN γ upregulates MHC class I and this will lead to increase presentation of antigen by tumor cells to the TCR-specific CD8 cells. In contrast, CAR-T cell-mediated killing does not require MHC antigen presentation. Additionally, a recent study has suggested that IFN γ plays a greater role in direct cytotoxicity of solid tumors as opposed to leukemias (Larson et. al. Nature, 2022).

Point 9. Please specify if the leukemia tumor also grows slower in the KO and conditional KO compared to WT.

The E2a leukemia cell line is poorly immunogenic (Qin et al. Blood 2018). The targeted CD19 antigen is murine CD19 and thus a self-antigen. For the studies using B16 cells in the conditional MCJ KO mice, we used B16-OVA melanoma cells that express chicken ovalbumin as non-self antigen. Nevertheless, to address the question from the reviewer, we injected E2a cells into WT mice and T-cMCJ KO mice (not irradiated). As expected, E2a was uniformly fatal in both WT and T-cMCJ KO mice. There was no statistically significant difference in the survival of WT or T-cMCJ KO mice and no convincing trend towards a survival advantage of T-cMCJ-KO mice

(Reviewer Fig. 4). We provide the results to the reviewer, but we do not think they are needed for our manuscript.

Figure 4. No enhanced protection against E2a B-ALL leukemia cells in T-cMCJ KO mice. WT or T-cMCJ KO hosts were administrated with 10^6 E2a cells through I.v. injection (n=5). The survival of the mice was followed over time. *, denotes $p < 0.05$ as determined by log-rank (Mantel-Cox) test

Point 10. Authors should draw more conclusions from Fig 5h. What metabolic pathways are affected? For Fig 5i and 7a-b, are these only the metabolites that change significantly? Some short chain carnitines seem to be accumulating in the KO. What are the causes-repercussions on fatty acid metabolism in these cells and is this an aspect that needs to be considered for increased anti-tumor activity? Regarding this point the manuscript sorts of lack a complete metabolic characterization of these cells. Can authors use an antibody against MCJ or if available, small molecule inhibitors and show higher OCR in WT? Otherwise can supercomplexes assembly or CI activity be blocked in the KO and see consequences in OCR and their metabolic flexibility? A full analysis on the complexes and their activity in these cells and also addressing fuel preferences is mandatory. Authors should manipulate fuels or complexes and see what MCJ deficient cells are relying on. A small *in vivo* piece of data would also be beneficial: what metabolic environment are these cells finding *in vivo* that the absence of MCJ would give an advantage? (Lack of glucose/fatty acids types-levels, etc..?).

We agree with the reviewer that the metabolic aspect of the manuscript needed to be further support. As described in detail above, in the introduction for this reviewer, we have now performed a significant amount of new experiments to address this weakness. We now provide: **1)** RNAseq analysis in WT and MCJ KO CD8 CAR-T cells after 3 expansions with IL-2, further supporting enhanced mitochondrial metabolism (OXPHOS and fatty acid oxidation) in MCJ KO CAR-T cells (**new Fig. 5i and new Extended Data Fig. 13a, 13b, 13c and 13d**), **2)** metabolomics analysis of CAR-T cells after 3 expansions with IL-2 followed by incubation in media to mimic the cytokine withdrawal these cells undergo after transfusion into the *in vivo* environment, supporting a potential adaptation of the metabolism towards enhanced protein catabolism and the use of fatty acid β -oxidation (also in mitochondria) as a source of energy in MCJ KO CAR-T cells (**new Fig. 5j and new Extended Data Fig. 14a and 14b**), **3)** metabolomics in WT and MCJ KO CAR-T cells isolated from bone marrow 4 days after *in vivo* transfer to mice bearing E2a leukemia, showing an accumulation of short/medium-chain fatty acids in MCJ KO CD8 CAR-T cells (**new Extended Data Fig. 15b and 15c**), **4)** data showing that reducing MCJ levels in human CD8 CAR-T cells (shMCJ) results in a marked increase of mitochondrial respiration (**new Fig. 7f and 7j, and new Extended Data Fig. 19a and 19b**), **5)** data showing that pre-treatment with oligomycin blocks the enhanced anti-tumor killing activity of MCJ KO CD8 CAR-T cells (**new Fig. 5k**), demonstrating that mitochondrial respiration is essential for CAR-T cells anti-tumor activity.

We hope that with all these new experiments, analyses and conclusions the reviewer also considers that our manuscript now provides a sufficient metabolic characterization of the CAR-T cells.

Unfortunately, we do not currently have a small molecule inhibitor for MCJ. However, the shMCJ strategy works well for CAR-T cell therapy and should be adaptable to TCR-driven T cell therapy since the genetic manipulation required to inhibit MCJ expression is analogous to those used in our CAR-T cells. We now provide additional data showing that human shMCJ CD8 CAR-T cells have increased mitochondrial respiration (**new Fig. 7f and 7j and new Extended Data**

Fig. 19a and 19b). We think this strategy has the potential to be moved forward to clinic testing. As we mentioned in the response to Reviewer 2, as part of an independent project, we have shown that a reduction of MCJ levels (not a complete abrogation) in the liver in vivo using an siRNA causes a major reduction in liver steatosis and fibrosis in mouse models of fatty liver disease, and we are currently in talks with FDA for an IND to treat NASH patients (there are 5 siRNA-based drugs FDA approved for liver diseases).

Point 11. It would be important to comment/ provide information regarding transcriptional changes in the WT vs KO. Maybe one round of IL2 expansion may be sufficient to see some differences; although authors provide surface markers that do not change besides IFN γ , it may be interesting to look at genes/senescence, general fitness etc.. in these cells at the level of transcription, possibly investigating gene sets downstream of major transcription factors (es PPAR γ etc.).

As mentioned above, following the suggestion from the reviewer we have now performed RNAseq analysis in WT and MCJ KO CAR-T cells after the 3rd expansion with IL-2. The results show that the top 10 differentially regulated pathways between WT and MCJ KO CD8 CAR-T cells are metabolic pathways (myc-targets, OXPHOS, fatty acid oxidation etc), as well as IFN γ pathway (**new Fig. 5i, and new Extended Fig. 13a, 13b, 13c**). In addition, the results also show increased expression of genes associated with generation of survival of memory CD8 cells (e.g. Tcf1, Ifitm3) (**new Extended Fig. 13d**).

Point 12. Authors mention that ROS do not change but they do not show ROS after all the IL2 passages and in the experiments ex vivo from the tumor.

The reviewer is correct, we have shown that the ROS is not increased in activated CD8 cells (and other cell types) but we did not examined ROS in IL-2 expanded CAR-T cells. We have now included those experiments by mitoSox staining (mitochondrial ROS). The results actually show a decrease in ROS levels in MCJ KO CD8 CAR-T cells relative to WT CD8 CAR-T cells after 3 expansions with IL-2 (**new Extended Data Fig. 10a and 10b**)

Point 13. Regarding Fig 7c, have authors tried this in IL2 expansion?

IL-2 expansion of non-transduced T cells (as in Figure 7c) was not done, however, the increased IFN γ secretion seen with siRNA knock-down of MCJ in Figure 7c was replicated in human CAR T cells containing an effective shRNA against MCJ (**new Figure 7m**). These CAR-T cells were expanded in IL-2 and the MCJ-related effect on IFN γ was observed.

Point 14. It is not clear if (and where) authors show (western blot) if CD8 T cells maintain MCJ expression during the ex vivo expansion in mouse vs human.

In our initial submission, we showed the expression of MCJ in human CD8 CAR-T cells after 3 expansions with IL-2 (and the reduction of MCJ levels in shMCJ CD8 CAR-T cells) (Fig. 7e and 7i). However, the reviewer is correct, we did not provide data showing MCJ expression in mouse CD8 cells after expansion. We now provide Western blot analysis showing MCJ expression in CD8 cells after expansion with IL-2 (**new Extended Data Fig. 2**).

Point 15. It would be interesting to show that the efficacy of human CD19-BBz/shMCJ-1 CD8 CAR-T cells in killing human leukemia cells expressing CD19 is poor or absent in a donor that has low level of MCJ.

We apologize if we misinterpreted this comment, but our interpretation is that the reviewer is asking whether silencing MCJ by shMCJ in CD8 CAR-T cells from an MCJ-low donor would have little or no effect on their killing activity relative to control c-shRNA CD8 CAR-T cells considering the levels of MCJ are already low. We have now included experiments using CD19-BBz/shMCJ-2 CD8 CAR-T cells and CD19-BBz/c-shRNA CD8 CAR-T cells from an MCJ-low donor. Consistent with our hypothesis, we found no difference in cytotoxic degranulation between CAR-T cells with MCJ silencing versus those transduced with control shRNA (**new Extended Data Fig. 21a and 21b**). Again, we apologize if this was not the question that the reviewer asked.

Point 16. It would be very important to show that MCJ is directly regulating the respiration which in turn is the solely responsible for antitumor activities and not via activation of transcriptional /translation/ signaling pathways (this goes back to point n10 and 11). This point is raised since MCJ has been link to c-Jun pathway. There are compounds that promote supercomplex assembly (3,4-methylenedioxy- β -nitrostyrene (MNS) and SYK inhibitors) that authors could use to show that in the WT they ameliorate CAR T function at least in vitro, and maybe they have small or no effect in the KO. The idea behind this is forcing the respiratory supercomplexes in other ways and show the same results achieved with the deficiency of MCJ.

We thank the reviewer to point out the paper by Kobayashi et al 2023 identified Syk inhibitors as compounds that can promote the supercomplex assembly. Those studies were performed in the C2C12 myoblastic cell line. Unfortunately, Syk inhibitors (MNS is also identified in the paper as a Syk inhibitor) will not work for our studies with CD8 cells. Syk (spleen tyrosine kinase) is a cytoplasmic tyrosine kinase abundantly expressed in CD8 and CD4 cells, as well as B cells, and it is from the same family of Zap-70 kinase (essential for T cell activation and T cell development). Syk also plays a role in T cell development and activation (Hauck et al. 2018 Clinical Immunology; Cheng et al. 1997 PNAS; Colucci et al. JI 2010; etc). Thus, the well-established role of Syk in T cell activation will be a confounding factor for testing the Syk inhibitors as a way to promote supercomplexes formation in the mitochondria.

Nevertheless, as described above (Point 12), we now provide new experiments showing MCJ KO CAR-T cells after the expansion with IL-2 do not have increased ROS, but decreased ROS levels, despite the increased mitochondrial membrane potential relative to WT CAR-T cells (**new Extended Data Fig. 10a and 10b**). These data further support that loss of MCJ in CD8 CAR-T cells promotes mitochondrial membrane potential by facilitating supercomplex formation. In addition, we now show that a short pretreatment with oligomycin (inhibitor of mitochondrial ATP synthase) is sufficient to abrogate the enhanced killing activity by MCJ KO CD8 CAR-T cells (**new Fig. 5k**).

Point 17. Authors do not address well the memory parameters throughout the manuscript: if authors take the CAR T cells out of the mouse after different periods of time can they show higher response to the antigen in vitro?

We now provide cell surface markers (CD44, CD62L, PD1 and Tim3) expression in WT and MCJ KO CD8 CAR-T cells at different times post-administration into mice bearing E2a leukemia. As expected, we did not see significant differences in the expression of these markers (**new Extended Data Fig. 8b, 8c, 8d**). We did not think that increased mitochondrial metabolism will affect the expression of these markers,

Minor Points.

Point 1. Arise of CRS and ICANS should be better explained.

The development of CRS and ICANS in CAR-T cell therapy is well established and, as we mentioned in the manuscript, it is one of the limiting factors of CAR-T cell therapy. Administration of anti-IL-6R blocking antibody to suppress IL-6 activity is commonly used as part of the clinical protocol for CAR-T cell therapy. We now expanded the Introduction section regarding mechanisms for ICANS (**page #4**). Please, if this is not what the reviewer was requesting, the reviewer can let us know the specific aspects of the toxicity that he/she would like for us to expand.

Point 2. Explain molecular mechanisms by which memory CD8 cells protect against a lethal dose of influenza virus without the help of CD4 cells. Moreover, it is not clear in what aspect they address the presence of CD4 as irrelevant to the success of the therapy in authors' models.

In our previous study (Champagne et al Immunity 2016) we show that WT memory CD8 cells in the absence of CD4 cells were not able to protect mice from a lethal dose of influenza virus. In contrast, MCJ KO memory CD8 cells were able to provide some protection. It is well known that CD8 cells require CD4 cells to help for maximum activity, and it is thought to be through cytokines produced by CD4 cells. We think that CD4 cells help to maintain the metabolism of CD8 cells through those cytokines, and that the increased mitochondrial respiration in MCJ KO CD8 cells is sufficient to improve cell fitness of CD8 cells in the absence of CD4 cells.

Our manuscript focuses on CD8 CAR-T cells. However, following Reviewer 1's suggestions, we have now performed experiments to test whether the absence of MCJ could also improve anti-tumor killing activity of a combination of CD4 and CD8 CAR-T cells. Please, see the data provide for Reviewer 1 (**Reviewer Fig. 2**). Interestingly, the combination of CD4/CD8 CAR-T cells do worse than just CD8 CAR-T cells *in vitro* (probably because the frequency of CD8 CAR-T cells is lower in the combination and they are the best in killing), but MCJ KO CD4/CD8 CAR-T cells still do better than WT CD4/CD8 CAR-T cells.

Point 3. For Fig 1e-f-g please indicate on the figure panel the number of expansions addressed.

The number of expansions for each panel was already defined in the figure legend for each figure panel. Following the reviewer's request, we have now also added it to the actual figure panel (on the top) (**Revised Fig. 1d, 1e, 1f and 1g**).

Point 4. The use of scatter plots for data representation is highly encouraged.

Following the reviewer recommendation, we have now revised all the bar graphs figures to show individual points (scatter plots).

Point 5. It is not entirely clear how WT or MCJ KO OT-I CD8 cells are sorted from the cultures and administered to mice.

We apologize but we are a bit confused about the question. We did not perform sorting of OT-I CD8 cells from cultures for their administration into mice for *in vivo* experiments. OT-I CD8 cells were isolated from WT and MCJ KO OT-I mice, activated with anti-CD3/CD28 Abs, washed after two days, expanded with IL-2 for two expansions, extensively washed and injected i.v. into the mice. No sorting was used for these studies. We have now clarified the procedure in the Methods sections.

Point 6. There is a bit of confusion regarding why sometimes authors use 1, 2 or 3rd expansion in showing their results/phenotypes.

We investigate the activity of CAR-T cells after different expansions with IL-2 because it is well known that CD8 cells after activation with anti-CD3/CD28 Abs (2 days) behave as effector cells and have the capacity of producing high levels of cytokines and have high killing activity, but when these cells are expanded over time with IL-2 they start losing their effector activity. In our studies we show that indeed mitochondrial membrane potential (MMP) of WT CAR-T cells after the 1st expansion with IL-2 (following activation with anti-CD3/CD28) is high, but these cells lose their MMP over time with IL-2 and after the 3rd expansion their MMP is quite low (Fig. 5a, 5b, 5c, 5d). We slightly revised the section of the results to make it clearer in the conclusions (**page #17**).

Point 7. It is not clear why in Fig 3h vs 3i the killing of leukemia is ~25% in WT vs 50%.

The results shown in Fig. 3h and 3i are from two independent experiments performed at a different time (this is why they are in two independent panels). The E:T ratio was slightly different between the two independent panels. We have now added the E:T ratio to the figure legend for each of the individual panels.

Point 8. It would be important to highlight if there is literature available regarding perforin as better killer than IFN γ in leukemia.

The contribution of perforin as a mechanism for CAR-T cell-mediated killing of leukemia cells has been shown (Ishii et al. J. Clin. Invest. 2020). In addition, it has also been shown that CAR-T cell killing does not require IFN γ pathways in liquid tumors (Larson et al. Nature 2022). We have now added these papers to our manuscript to further support our conclusions (**page #14 and page #13**).

Point 9. Indicate better where are authors gating exactly in all the expansions (Fig 5a-c).

We apologize for missing this information. We always gate in the live population of CD8 cells based on FCS and SSC by flow cytometry. We have now added to the Methods section (**page #36**). We thank you the reviewer for pointing this out.

Point 10. Often, too much method information is inserted in the results.

The reviewer may be correct, but we wanted to be sure that the reviewers (and the readers) could understand the experimental procedures and how the experiments were performed as they read the manuscript to help the interpretation of the results (instead of having to read the Methods section or figure legend to understand). For now, we leave the manuscript as it is. However, if the editor and/or the other reviewers also feel that we should shorten the method information in the Results, we will be happy to do it.

Point 11. Regarding Fig 7e, have authors done a western blot after 1 expansion?

These Western blots were performed with purified CAR-T cells (separating them from those cells that were not expressing CAR). As expected, after one expansion with IL-2 we only have limited numbers of CAR-T cells. This is the reason we examined MCJ expression after the 3rd expansion when we can obtain larger number of CAR-T cells.

Point 12. It is not clear what shMCJ-2 is and how it differs from the other one mentioned/used.

shMCJ-1 and shMCJ-2 are two different and independent sequences of siRNA (different locations in the gene) for human MCJ gene. We designed two constructs with two different siRNAs

sequence to further show the specificity of the effect of silencing MCJ. In the methods section we provide the two sequences (Methods section).

Point 13. Regarding references 34,35,60, please discuss more the mechanism regulating the source of ATP production during cell manufacturing and the *in vivo* function and persistence.

In the revised version, we have now extended the Discussion where these references were cited to provide more information about glycolysis, mitochondrial respiration as well as fatty acid oxidation (**page #27**).

Point 14. For CAR+ enrichment, are there differences in % of enriched cells for WT and MCJ KO? Authors should characterize these cells for markers other than the ones they show (es chemokine markers, TNF α , TBX21, IL-4, IL-5, CCR4, GATA3, IL-9, IL-10, IRF4, CCR6, KLRB1, IL-17, RORC, CCR7, CD95, TIM-3, LAG-3, CTLA-4, NKG2A, CD39 etc).

After CAR+ enrichment, there were no differences in the % of enriched cells for WT and MCJ KO CAR-T cells, in both groups the enrichment was about 100%. We have added a sentence in the methods with this information (**page #32**).

Following the suggestions from the reviewer we have now examined cell surface markers expression on WT and MCJ KO CD8 CAR-T cells *in vitro* and *in vivo* by flow cytometry, and as predicted, we do not observe major differences in the surface markers, but we found slight differences in the intracellular markers such as Tcf1 (**new Extended Data Figures 6a, 6c, 8a, 8b, 8c, 8d, and 20a and 20b**).

In addition, as mentioned above we have now performed RNAseq analysis in WT and MCJ KO CAR-T cells after the 3rd expansion with IL-2. The results show that the top 10 differentially regulated pathways between WT and MCJ KO CD8 CAR-T cells are metabolic pathways (myc-targets, OXPHOS, fatty acid oxidation etc), as well as the IFN γ pathway (**new Fig. 5i, and new Extended Fig. 13a, 13b, 13c**). Moreover, the results also show an increased expression of genes associated with generation or survival of memory CD8 cells such as *Tcf7* (encoding Tcf1), *Ifitm3*, *Sell* (**new Extended Fig. 13d**).

REVIEWERS' COMMENTS

Reviewer #1 (Remarks to the Author):

Dear authors, thank you for addressing my points carefully and providing the additional data. I have no further comments.

Reviewer #2 (Remarks to the Author):

The authors have added a substantial amount of results to address the questions from the reviewers and the manuscript which is commendable.

I apologize for not having detected the reference to Chamagne et al. Earlier in the manuscript.

There are a few points remaining :

Point 1 : Regarding the additional transcription factors investigated by the authors (Tcf1, Foxo1, Tbet and Tox) that the authors show in New Extended data figure 6. A slight increase in Tcf1 and Foxo1 in MCJ-KO CAR-T cells expanded in IL-2 is shown, but the levels are higher in T cells expanded in IL7/IL-15. The authors also saw upregulation of Tcf7 in WT and MCJ KO CD8 CAR-T cells generated with IL-2 in RNA-seq analyses (new Extended Data Fig. 13d).

How do the authors explain that despite the high expression of Tcf1 whereas all the T cells also seem to express high levels of Tox? Others have described Tox to be predominantly expressed in TEM and TEMRA cells, whereas Tcf1 is predominantly expressed in TN and TCM cells (PMID: 32620560, PMID: 37625402).

In new Extended Data Fig. 20a and 20b the authors show subtypes of memory/effector cell in human CD19-BBz-shMCJ. Here the CD62L axis of the plots should be adjusted to avoid having too many cells on the axis. If 70-80% of these cells are classified as TSCM the levels of Tox found in New Extended data figure 6 should be lower.

Regarding Point 2 and the need for CD4 T cells in MCJ negative CD8 CAR T cells and the Figure 1 provided for the reviewers, the effort to clarify is appreciated. However, this would probably require more long term experiments than in vitro killing as the authors also mentions, killing is reduced with the CD4:CD8 ratio of 1:1 as CD4 T cells have slower killing kinetics. It is not recommended to include the figure in the manuscript which focuses on CD8 CAR T cells.

In Fig. 1i the authors show that IFN γ is necessary for the killing of tumour cells by TCR-specific CD8 T cells. This is in line with several recent publications demonstrating that CAR T cell killing is dependent of IFN γ in solid tumours PMID: 35418687 (which the authors cite), very important for the CD4 CAR T cell contribution PMID: 37248395 and essential for sustaining CD8 CAR T cell cytotoxicity (PMID: 33771887), not only due to the effects of IFN γ on the TME.

In Extended Data Fig. 3, the authors show that IFN γ blockage abolishes the superior killing activity of MCJ-KO OT-I CD8 CAR-T cells. The authors say on page 14 that « These results indicated that the killing of leukemia cells by CD8 CAR-T cells is independent of IFN γ in contrast to killing of tumor cells by TCR-specific CD8 cells (Fig. 1i). This is mostly because the ability of CAR-T cells to kill is independent on MHC antigen presentation (IFN γ -mediated MCH class I expression) ».

This could be the difference between killing of solid tumours and liquid tumolurs by CAR T cells as indicated in the paper already cited by Larson et Al. Nature 2022 (PMID: 35418687) where they found that IFN γ R signalling was required in solid tumour for sufficient adhesion of CAR T cells to mediate productive cytotoxicity. ACR T cells have also been shown to upregulate ICAM-1 in and IFN γ -dependent manner (PMID: 34686489). Please comment.

Reviewer #3 (Remarks to the Author):

Authors have thoroughly addressed almost all the concerns raised during the initial review process. The revisions made significantly enhance the manuscript's clarity, rigor, and contribution to the field.

After a careful and comprehensive evaluation of the updated submission, I am quite satisfied with

the modifications and responses provided by the authors. They have effectively incorporated the suggested changes and provided clear explanations for each point raised, which, in my opinion, resolve the previously identified issues.

Given the improvements made and the quality of work presented, I have no further comments or suggestions. I believe the manuscript now meets the high standards of your journal and will make a valuable contribution to the scientific community.

RESPONSE TO REVIEWERS' COMMENTS

Reviewer #1 (Remarks to the Author):

Dear authors, thank you for addressing my points carefully and providing the additional data. I have no further comments.

Reviewer #2 (Remarks to the Author):

The authors have added a substantial amount of results to address the questions from the reviewers and the manuscript which is commendable. I apologize for not having detected the reference to Chamagne et al. Earlier in the manuscript. There are a few points remaining :

Point 1 : Regarding the additional transcription factors investigated by the authors (Tcf1, Foxo1, Tbet and Tox) that the authors show in New Extended data figure 6. A slight increase in Tcf1 and Foxo1 in MCJ-KO CAR-T cells expanded in IL-2 is shown, but the levels are higher in T cells expanded in IL7/IL-15. The authors also saw upregulation of Tcf7 in WT and MCJ KO CD8 CAR-T cells generated with IL-2 in RNA-seq analyses (new Extended Data Fig. 13d). How do the authors explain that despite the high expression of Tcf1 whereas all the T cells also seem to express high levels of Tox? Others have described Tox to be predominantly expressed in TEM and TEMRA cells, whereas Tcf1 is predominantly expressed in TN and TCM cells (PMID: 32620560, PMID: 37625402).

Tox is normally the canonical marker for exhausted T cells (Tex) based on the initial studies by John Wherry's group (*Nature* volume 571, pages 211–218 (2019)). Most of the studies have examined the expression of Tcf1 and Tox in the context of chronic infection and, more recently within a tumor, with continued exposure to antigen. In our case, after the initial activation, effector CD8 cells are just simply expanded with IL-2 in the absence of ongoing TCR or CAR stimulation. These cells should not be considered exhausted as those from chronic infections, despite their expression of Tox. The reviewer brought a good point, other studies (the papers provided by the reviewer) have shown that Tox is expressed in effector memory cells in human PBMC. However, our cells should not be considered true memory cells either since they are still proliferating with IL-2. We described MCJ KO CD8 CAR-T cells as having "memory-like phenotype" without defined them as "memory cells". Little characterization about the expression of these markers during the expansion of effector CD8 cells with IL-2 has been done. Based on the reviewer's comments, we have now added the references provided by the reviewer to the revised version (page 13)

In new Extended Data Fig. 20a and 20b the authors show subtypes of memory/effector cell in human CD19-BBz-shMCJ. Here the CD62L axis of the plots should be adjusted to avoid having too many cells on the axis. If 70-80% of these cells are classified as TSCM the levels of Tox found in New Extended data figure 6 should be lower.

Extended Data Fig. 20 shows the phenotype for human CAR-T cell, but Extended Data Fig. 6 shows the phenotype of mouse CAR-T cells. Clear differences have been found between human

and mouse CAR-T cells phenotype, in part due to the phenotype of T cells in human before the activation. The markers used to defined T cell subsets (e.g. effector, memory, stem cell memory etc) are also different between mouse and human. We don't think we can compare the phenotype of mouse versus human CAR-T cells

Regarding Point 2 and the need for CD4 T cells in MCH negative CD8 CAR T cells and the Figure 1 provided for the reviewers, the effort to clarify is appreciated. However, this would probably require more long term experiments than in vitro killing as the authors also mentions, killing is reduced with the CD4:CD8 ratio of 1:1 as CD4 T cells have slower killing kinetics. It is not recommended to include the figure in the manuscript which focuses on CD8 CAR T cells.

We agree with the reviewer and we will not incorporate those figures into the manuscript.

In Fig. 1i the authors show that IFN γ is necessary for the killing of tumour cells by TCR-specific CD8 T cells. This is in line with several recent publications demonstrating that CAR T cell killing is dependent of IFN γ in solid tumours PMID: 35418687 (which the authors cite), very important for the CD4 CAR T cell contribution PMID: 37248395 and essential for sustaining CD8 CAR T cell cytotoxicity (PMID: 33771887), not only due to the effects of IFN γ on the TME. In Extended Data Fig. 3, the authors show that IFN γ blockage abolishes the superior killing activity of MCH-KO OT-I CD8 CAR-T cells. The authors say on page 14 that « These results indicated that the killing of leukemia cells by CD8 CAR-T cells is independent of IFN γ in contrast to killing of tumor cells by TCR-specific CD8 cells (Fig. 1i). This is mostly because the ability of CAR-T cells to kill is independent on MHC antigen presentation (IFN γ -mediated MCH class I expression) ». This could be the difference between killing of solid tumours and liquid tumours by CAR T cells as indicated in the paper already cited by Larson et Al. Nature 2022 (PMID: 35418687) where they found that IFN γ R signalling was required in solid tumour for sufficient adhesion of CAR T cells to mediate productive cytotoxicity. CAR T cells have also been shown to upregulate ICAM-1 in and IFN γ -dependent manner (PMID: 34686489). Please comment.

We thank the reviewer to bring this point. We agree that exposure of malignant cells (both solid and hematologic) to IFN γ can have consequences beyond MHC I upregulation, such as changes to adhesion molecules or direct cytotoxicity that could impact the effectiveness of both TCR-driven and CAR T cells, and that we cannot make a general statement about the fact that CAR-T cells do not require IFN γ solely because they do not need MHC presentation. We have now modified our description (page 14) to specifically state that IFN γ may not be required for CAR-T cell activity in all cancers (as it is our case), acknowledging that it can contribute to CAR-T cell activity in some solid and hematologic tumors. We also provide the appropriate references.

Reviewer #3 (Remarks to the Author):

Authors have thoroughly addressed almost all the concerns raised during the initial review process. The revisions made significantly enhance the manuscript's clarity, rigor, and contribution to the field.

After a careful and comprehensive evaluation of the updated submission, I am quite satisfied with the modifications and responses provided by the authors. They have effectively incorporated the suggested changes and provided clear explanations for each point raised, which, in my opinion, resolve the previously identified issues.

Given the improvements made and the quality of work presented, I have no further comments or suggestions. I believe the manuscript now meets the high standards of your journal and will make a valuable contribution to the scientific community.